# Neutralizing antibody immune correlates in COVAIL trial recipients of an mRNA second COVID-19 vaccine boost

Neutralizing antibody titer has been a surrogate endpoint for guiding COVID-19 vaccine approval and use, although the pandemic's evolution and the introduction of variant-adapted vaccine boosters raise questions as to this surrogate's contemporary performance. For 985 recipients of an mRNA second bivalent or monovalent booster containing various Spike inserts [Prototype (Ancestral), Beta, Delta, and/or Omicron BA.1 or BA.4/5] in the COVAIL trial (NCT05289037), titers against 5 strains were assessed as correlates of risk of symptomatic COVID-19 ("COVID-19") and as correlates of relative (Pfizer-BioNTech Omicron vs. Prototype) booster protection against COVID-19 over 6 months of follow-up during the BA.2-BA.5 Omicron-dominant period. Consistently across the Moderna and Pfizer-BioNTech vaccine platforms and across all variant Spike inserts assessed, both peak and exposure-proximal ("predicted-at-exposure") titers correlated with lower Omicron COVID-19 risk in individuals previously infected with SARS-CoV-2, albeit significantly less so in naïve individuals [e.g., exposure-proximal hazard ratio per 10-fold increase in BA.1 titer 0.74 (95% CI 0.59, 0.94) for naïve vs. 0.41 (95% CI 0.23, 0.64) for non-naïve; interaction p = 0.013]. Neutralizing antibody titer was a strong inverse correlate of Omicron COVID-19 in non-naïve individuals and a weaker correlate in naïve individuals, posing questions about how prior infection alters the neutralization correlate.

Multiple COVID-19 vaccines spanning various platforms have been approved or authorized for emergency use by the US Food and Drug Administration (FDA)[1] on the basis of having demonstrated safety and efficacy against symptomatic COVID-19 in randomized, placebo-controlled phase 3 vaccine trials[2–4]. As these trials were conducted early in the COVID-19 pandemic, efficacy was demonstrated against primarily Prototype (Ancestral) COVID-19 or variants with only minor genetic drift. At approximately one year into the pandemic, highly genetically divergent SARS-CoV-2 variants began to emerge and dominated in sequential waves, each outcompeting the previously dominant variant[5]. These 'variants of concern' (VOCs) included Alpha/B.1.1.7, Gamma/P.1, Beta/B.1.351, Delta/B.1.617.2, and Omicron/B.1.1.529, as well as Omicron subvariants BA.1, BA.4, and BA.5, and

demonstrated varying degrees of immune escape[6]. With the introduction of booster doses, vaccines based on the Prototype strain maintained high protection against severe disease and death[7–12], although protection was reduced against any infection, including mild or asymptomatic infections[11,13,14].

In response, variant-adapted vaccines began to be deployed during the Omicron era[15], and the FDA recommended BA.4/BA.5 inclusion in COVID-19 booster vaccines in Fall 2022[16]. This recommendation was later updated to XBB.1.5 in Fall 2023 and then again to JN.1 in Fall 2024[17]. COVID-19 boosters will likely need to continue being updated as SARS-CoV-2 evolves, raising questions as to booster immunogenicity and protection against previous and current variants. The COVID-19 Variant Immunologic Landscape (COVAIL) trial was a

✉ e-mail: dfollmann@niaid.nih.gov

randomized, open-label clinical trial that evaluated the safety and immunogenicity of second COVID-19 variant vaccine boosters and was conducted in four stages at 22 sites in the United States[18,19]. COVAIL enrolled participants aged 18 years and older who had previously received an initial primary vaccination series as well as a first Prototype booster dose at least 16 weeks prior to enrollment and randomized them to a second homologous or heterologous boost, administered on Day 1 (D1). These booster vaccines included mRNA and recombinant protein vaccines with one or two Spike insert proteins from among Prototype, Beta, Delta, and Omicron (BA.1 or BA.4/BA.5).

In this work, we studied how antibody level correlates with symptomatic COVID-19 (hereafter, "COVID-19") using correlate of risk (CoR) and correlate of protection (CoP) analyses[20–22]. Formally, a CoR seeks to understand how the immune marker is associated with the clinical endpoint in a defined cohort such as a particular vaccine arm. CoP analyses[23] seek to understand how the immune marker potentially modifies and explains the vaccine efficacy of a vaccine arm vs placebo, or the relative protective efficacy of one vaccine arm vs a second vaccine arm.

We previously reported that 50% inhibitory dilution (ID50) neutralizing antibody (nAb) titers against D614G ("Reference", corresponding to Prototype harboring the D614G mutation), Delta, Beta, BA.1, and BA.4/BA.5, as well as their maximum diversity-weighted[24] geometric mean, measured on 14 days post-boost (D15, 'peak'), were inverse CoRs of COVID-19 through ~6 months in COVAIL trial Stage 3 participants[25]. These participants received a boost containing recombinant Spike protein of Prototype, Beta, or Beta + Prototype variants. We also assessed predicted-at-exposure levels of the same nAb titer markers, with similar results supporting each as an inverse CoR. In contrast, markers corresponding to the fold-change of antibody levels between D1 and D15 were not inverse CoRs.

The first objective of the present study was to assess the same six D15, fold-rise, and exposure-proximal[26] (i.e., the predicted level at the time of exposure leading to a COVID-19 endpoint) nAb titer markers as CoRs of COVID-19 in recipients of a one-dose mRNA second vaccine boost, including separate analyses in SARS-CoV-2 naïve and non-naïve participants. The second objective was to assess, for groups of booster study arms (e.g., Omicron-containing vaccines versus Prototype-only vaccines), the same six antibody markers in the same population as CoPs against COVID-19. Given that the majority of COVID-19 endpoints were caused by BA.4 or BA.5, the third objective was to assess BA.4/BA.5 and D614G nAb titers as CoRs of BA.4/BA.5 COVID-19. In the present analysis, the COVID-19 endpoint was the symptomatic subset of self-reported or study-conducted positive SARS-CoV-2 tests (details below). As more than 200 breakthrough COVID-19 endpoints among the mRNA vaccine boost recipients were available for the correlates analyses, substantial precision could be achieved in the analyses. Moreover, this large number of breakthrough endpoints enabled additional questions to be answered compared to the previous COVAIL correlates analysis of the recombinant protein vaccines (with 22 breakthrough COVID-19 endpoints) and compared to other previous COVID-19 vaccine correlates analyses (e.g., 36 breakthrough COVID-19 endpoints in the first correlates analysis of the COVE trial of the mRNA-1273 COVID-19 vaccine[27]).

## Results

### Trial schema and participant demographics
The COVAIL trial enrolled healthy adults (≥18 years old), irrespective of prior SARS-CoV-2 infection status, who had received a primary vaccination series and a single boost (homologous or heterologous) with an approved or emergency use authorized COVID-19 vaccine[18,19]. In Stage 1 of the trial (March 30 to May 6, 2022), a total of 602 participants were enrolled and randomized to one of six Moderna mRNA-1273 vaccine regimens as a second vaccine boost, 597 of whom received the assigned dose(s). Of those 597, 512 (85.8%) received one dose of an mRNA-1273 vaccine, while the other 85 (14.2%) received a two-dose Beta + Omicron mRNA-1273 regimen. In Stage 2 of the trial (May 9 to 23, 2022), a total of 313 participants were enrolled and randomized to one of six Pfizer-BioNTech BNT162b2 mRNA vaccines as a second vaccine boost, 312 (99.7%) of whom received the single dose. In Stage 4 of the trial (October 4 to 28, 2022), a total of 202 participants were enrolled and randomized to one of two Pfizer-BioNTech mRNA bivalent vaccines as a second vaccine boost, 201 (99.5%) of whom received the single dose. The dose of any Moderna arm was 50 mcg and the dose of any Pfizer-BioNTech arm was 30 mcg.

Supplementary Fig. 1 shows the trial stages and details of the vaccine candidate(s) in each study arm. Supplementary Fig. 2 details participant flow from the 512 (Stage 1), 312 (Stage 2), and 201 (Stage 4) participants who were enrolled, randomized, and received a one-dose mRNA vaccine (total n = 1025) through to eligibility for COVID-19 cumulative incidence analysis (n = 1006) and further through to inclusion in the correlates analysis per-protocol cohort (n = 985). Participants who had eligibility deviations or were missing D1 or D15 BA.1 ID50 titers were excluded from cumulative incidence and correlates analyses. Further, participants with a COVID-19 endpoint prior to 7 days post-D15 were excluded from the correlates analysis per-protocol cohort for the reason that the D15 antibody marker measurements in these individuals might have been influenced by the preceding SARS-CoV-2 infection causing this COVID-19 endpoint.

A participant was determined to be SARS-CoV-2 non-naïve (hereafter, "non-naïve") if they self-reported a previous SARS-CoV-2 infection or had detectable anti-N antibodies [defined as Elecsys Anti-SARS-CoV-2 assay (Roche) cutoff index ≥1.0] at D1. Otherwise, a participant was considered SARS-CoV-2 naïve (hereafter, "naïve"). Of the 1006 participants eligible for cumulative incidence analyses (Supplementary Fig. 2), 648 (64.4%) were naïve (N = 341 female, mean age 50.4 years, and N = 307 male, mean age 51.8 years). The remaining 358 (35.6%) were non-naïve (N = 201 female, mean age 40.4 years and 157 male, mean age 41.2 years), almost all of whom (342/358, 95.5%) had a positive anti-N test result at baseline. The remaining 16 non-naïve participants (4.5%) tested anti-N negative at baseline and were classified as non-naïve due to self-reporting a previous SARS-CoV-2 infection. Supplementary Tables 1–3 provide demographic and clinical information for all enrolled participants, naïve participants, and non-naïve participants by stage and boost arm.

### Available neutralizing antibody titer data
Serum titers were assayed against pseudoviruses expressing Spike-D614G, B.1.617.2 (Delta), B.1.351 (Beta), B.1.1.529 (Omicron BA.1), or Omicron BA.4/BA.5 via a central lab (Monogram, San Francisco, California). Titers were measured for all one-dose mRNA arm participants using samples from baseline (D1) and 14 days (D15), 28 days (D29), 90 days (D91), and 180 days (D181) after the boost, with the exception that BA.4/BA.5 titers at D15 were only measured from a random subset (Supplementary Fig. 1 shows the serum sampling time points). All data analyzed in the present work have been previously studied in immunogenicity analyses[18,19]. D15 BA.4/BA.5 titers for participants with missing values were imputed based on D29 BA.4/BA.5 ID50 and D15 BA.1 ID50 [see the Statistical Analysis Plan (SAP), provided in the Supplementary Material], leveraging the high correlation of these markers with D15 BA.4/BA.5 ID50. The same maximum diversity-weighted geometric mean ID50 marker as in our previous work[25], hereafter referred to as weighted average titer, was also assessed.

### COVID-19 endpoint definition, case, and non-case definitions
The COVID-19 endpoint was a self-reported positive SARS-CoV-2 test (RT-PCR or antigen test) or study-conducted positive SARS-CoV-2 test (nasal swab and subsequent nucleic acid amplification test at an unscheduled illness visit) with onset date the earliest positive test date. Almost all (208/213) COVID-19 endpoints met the CDC clinical criteria

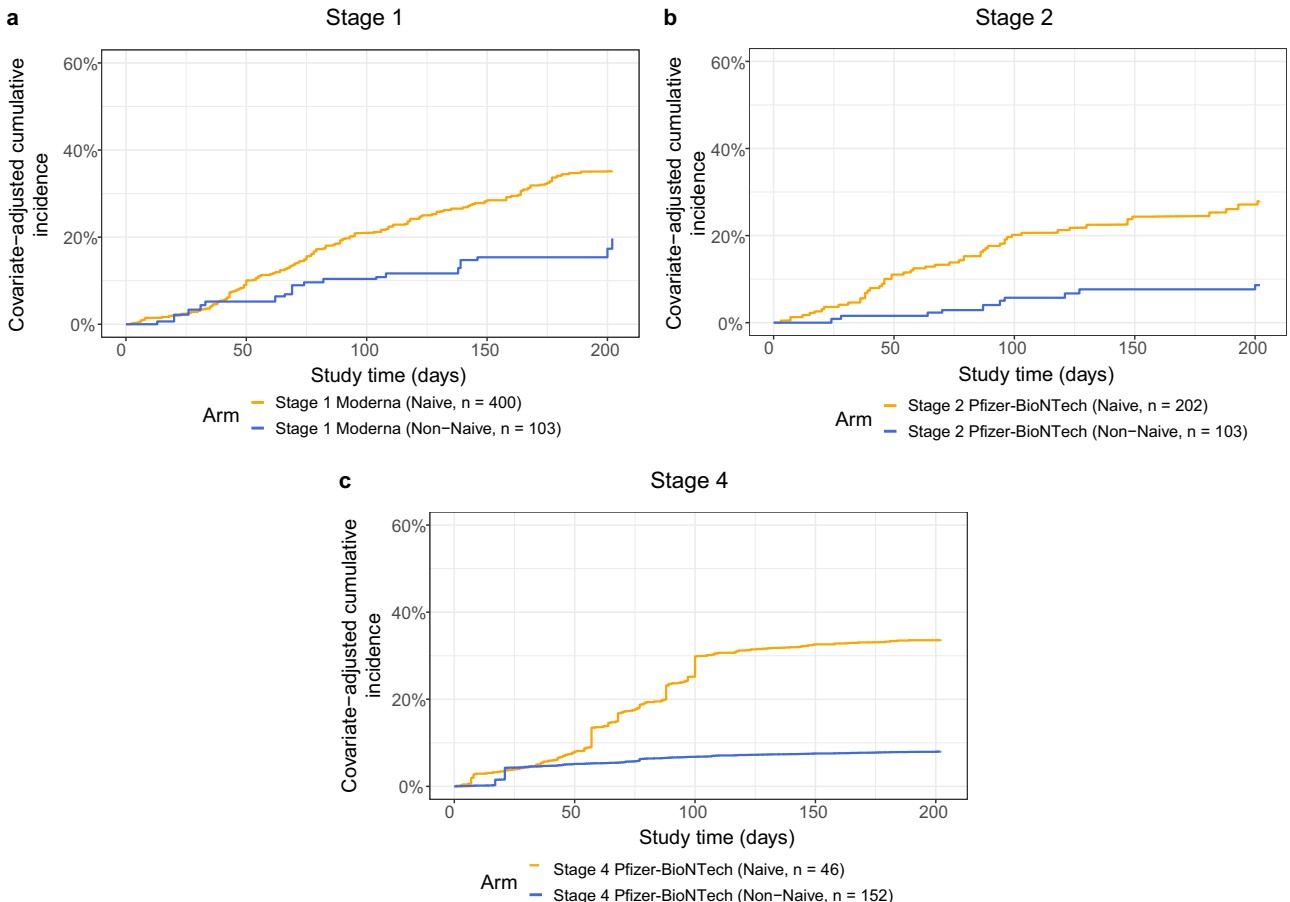

**Fig. 1 | Covariate-adjusted cumulative incidence of COVID-19 over time from the date of boost receipt (D1) through 188 days post-D15, shown separately for naïve (golden lines) and non-naïve participants (blue lines), in each of the three COVAIL trial stages 1, 2, and 4.** Cumulative incidence of COVID-19 is shown for **a** Stage 1 Moderna, **b** Stage 2 Pfizer-BioNTech, and **c** Stage 4 Pfizer-BioNTech boost recipients. Analyses adjusted for the force of infection score and risk score. Source data are provided as a Source Data file.

and supportive laboratory criteria for a COVID-19 surveillance case[28] (Supplementary Table 4). Periods of COVID-19 endpoint occurrence for correlates assessment were booster-proximal (endpoint occurring 7–91 days after D15), booster-distal (endpoint occurring 92–188 days after D15), and overall (endpoint occurring 7–188 days after D15).

Cases were endpoints occurring in participants in the correlates analysis per-protocol cohort in the booster-proximal, the booster-distal, or the overall time period as defined above. Non-cases were participants in the correlates analysis per-protocol cohort with no evidence of SARS-CoV-2 infection at 7 days post-D15 through to 188 days post-D15.

**Cumulative incidence of COVID-19**
Supplementary Table 5 shows the numbers of COVID-19 endpoints in each study stage, by participant naïve/non-naïve status, according to the time period of endpoint occurrence (early = prior to 7 days post-D15 visit, booster-proximal, and booster-distal). In Stage 1, among 400 naïve participants, there were 138 (34.5%) COVID-19 endpoints, including 11 early (prior to 7 days post-D15 visit), 78 booster-proximal, and 49 booster-distal. Among 103 non-naïve participants, there were 17 (16.5%) COVID-19 endpoints, including 1 early, 10 booster-proximal, and 6 booster-distal. In Stage 2, among 202 naïve participants, there were 51 (25.2%) COVID-19 endpoints, including 6 early, 35 booster-proximal, and 10 booster-distal. Among 103 non-naïve participants, there were 10 (9.7%) COVID-19 endpoints, including 7 booster-proximal and 3 booster-distal. In Stage 4, among 46 naïve partici-pants, there were 11 (23.9%) COVID-19 endpoints, including 2 early and

9 booster-proximal. Among 152 non-naïve participants, there were 7 (4.6%) COVID-19 endpoints, including 1 early, 5 booster-proximal, and 1 booster-distal. Supplementary Table 6 provides a breakdown of the numbers of COVID-19 endpoints, by participant naïve/non-naïve sta-tus, with each of 22 different symptoms. There was no statistical dif-ference in the distribution of symptoms between naïve and non-naïve participants (chi-squared p = 0.12).

Because the three stages enrolled participants in different calen-dar periods, a force of infection (FOI) score was calculated with data from the Coronavirus Resource Center's database hosted by Johns Hopkins University (JHU) (see the SAP); Supplementary Fig. 3 shows the distribution of FOI scores across calendar time and trial stage. In addition, a risk score was constructed for each participant based on baseline demographics and previous vaccination history using an ensemble machine learning method (see the SAP). Fig. 1 shows the cumulative incidence curves adjusting for the FOI score and risk score among naïve and non-naïve participants in Stages 1, 2, and 4. In all three stages, non-naïve participants had lower COVID-19 incidence than naïve participants (calendar-time covariate-adjusted Cox model p-values < 0.001).

**Neutralizing antibody titers in cases and in non-cases**
Immunogenicity characterization of nAb titers by randomization arm and vaccine platform were previously reported[18,19]; in the present work, we describe the titers in naïve and non-naïve participants by COVID-19 case/non-case status with our correlates scope. The six titer markers were highly correlated at D1 (Spearman rank correlation ρ between

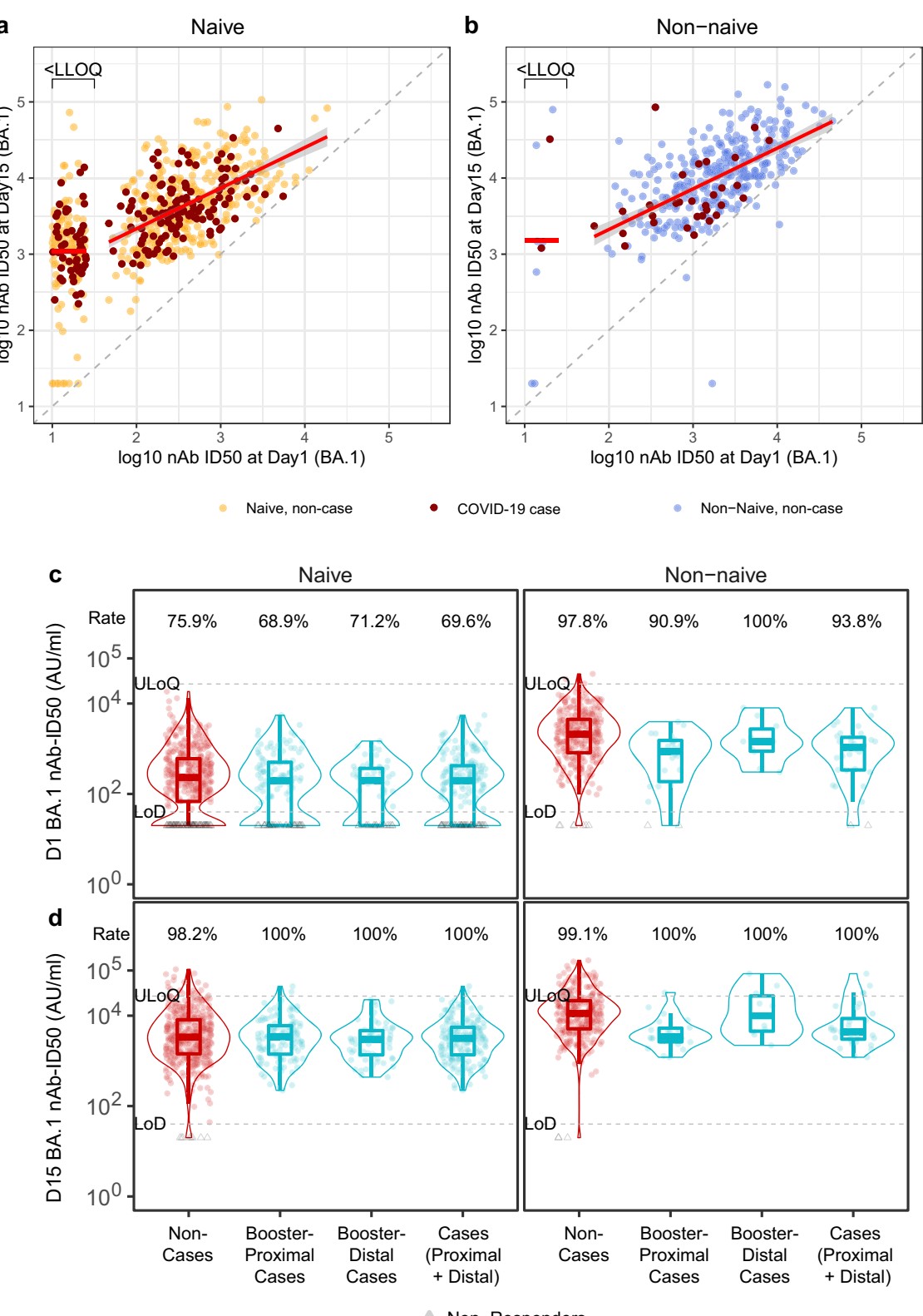

0.84 and 0.97; Supplementary Fig. 4) and at D15 (ρ between 0.76 and 0.96; Supplementary Fig. 5). We focus on BA.1 ID50 titer as it was measured for all participants at D15 and is relevant to the circulating Omicron strains. Fig. 2 shows scatterplots of D15 vs D1 BA.1 titers in naïve (Fig. 2a) and non-naïve (Fig. 2b) participants, identifying cases and non-cases. Naïve participants had lower D1 BA.1 titers as well as lower D1 BA.1 response rates compared to non-naïve participants, with

26% (N = 163 of 629) of naïve participants vs. 3% (N = 9 of 356) of non-naïve participants having undetectable BA.1 titer [defined as D1 BA.1 titer ≤limit of detection (LoD), 40 AU/ml]. For participants with detectable D1 titer, the linear regression lines showing a positive correlation between D1 and D15 titers were virtually identical for naïve and non-naïve participants, i.e. leading to the same predicted D15 titer for a given D1 titer whether naïve or non-naïve. D1 BA.1 titers of 100 and

**Fig. 2 | D1 and D15 BA.1 titers in SARS-CoV-2 naïve and non-naïve participants, and in non-cases and COVID-19 cases. a, b** Scatterplots of D1 and D15 log10 BA.1 titers in **a** naïve participants (N = 629; 181 cases and 448 non-cases) and **b** non-naïve participants (N = 356; 32 cases and 324 non-cases). Titers are in units AU/ml. The short horizontal red line represents the mean of log10 nAb-ID50 values that are <LLOQ at Day 15. The diagonal red line represents the fitted linear regression line between Day 1 and Day 15 log10 nAb-ID50 values that are above LLOQ. LLOQ, a lower limit of quantification. The gray area around the diagonal red line indicates the 95% confidence interval of the fitted line. Red dots identify COVID-19 cases; yellow dots identify naïve non-cases; blue dots identify non-naïve non-cases. **c, d** Violin box plots of **c** D1 and **d** D15 BA.1 ID50 titers, shown by non-cases (red dots) and COVID-19 cases (turquoise dots) (stratified by booster-proximal cases, booster-distal cases, and proximal + distal cases). In (**c**), D1 titers are shown from N = 448 naïve and 324 non-naïve non-cases, N = 122 naïve and 22 non-naïve booster-proximal cases, N = 59 naïve and 10 non-naïve booster-distal cases, N = 181 naïve and 32 non-naïve proximal+distal cases; in (**d**), D15 titers are shown from N = 448 naïve and 324 non-naïve non-cases, N = 122 naïve and 22 non-naïve booster-proximal cases, N = 59 naïve and 10 non-naïve booster-distal cases, N = 181 naïve

and 32 non-naïve proximal+distal cases. Violin plots contain interior box plots with upper and lower horizontal edges at the 25th and 75th percentiles of antibody level and middle line at the 50th percentile, and vertical bars the distance from the 25th (or 75th) percentile of antibody level and the minimum (or maximum) antibody level within the 25th (or 75th) percentile of antibody level minus (or plus) 1.5 times the interquartile range. Each side shows a rotated probability density (estimated by a kernel density estimator with a default Gaussian kernel) of the data. Non-cases: No evidence of SARS-CoV-2 infection at 7 days post-D15 through to 188 days post-D15. Booster-proximal cases: COVID-19 endpoint between 7 and 91 days post-D15 visit; booster-distal cases: COVID-19 endpoint between 92 and 188 days post-D15 visit; cases (proximal + distal): COVID-19 endpoint between 7 and 188 days post-D15 visit. Rate is the percentage of participants with positive responses, with positive responses defined as titer above the limit of detection (LoD). Gray triangles identify non-responders/undetectable. The upper dashed horizontal line in each plot shows the antigen-specific upper limit of quantification (ULoQ) and the lower dashed horizontal line in each plot shows the LoD. nAb-ID50, 50% inhibitory dilution neutralizing antibody titer. Source data are provided as a Source Data file.

1000 AU/ml for both naïve and non-naïve participants are predicted to be boosted 22-fold and 7-fold, respectively, by D15. Supplementary Fig. 6 shows the parallel results, which were similar, for the weighted average titer.

Fig. 2c, d, Tables 1 and 2 present BA.1 titers by case/non-case status and proximal/distal to booster. Among naïve participants, BA.1 titers at D1 were slightly higher in non-cases than cases, with geometric mean (GM) ID50 in non-cases, booster-proximal cases, and booster-distal cases 203 (95% CI: 174, 236), 155 (115, 207), 134 (93.3, 191) AU/ml, respectively, with Geometric Mean Ratio (GMR) 1.31 (95% CI 0.95, 1.82) of non-cases vs. booster-proximal cases and 1.52 (0.98, 2.35) of non-cases vs. booster-distal cases (Fig. 2c, Table 1). At D15, BA.1 titers were similar or slightly higher in non-cases vs. cases, with GM ID50 in non-cases, booster-proximal cases, and booster-distal cases 3240 (95% CI: 2844, 3690), 3006 (2482, 3641), 2685 (2096, 3440) AU/ml, respectively, with GMR 1.08 (0.82, 1.41) of non-cases vs. booster-proximal cases and 1.21 (0.83, 1.74) of non-cases vs. booster-distal cases (Fig. 2d, Table 1).

In contrast, among non-naïve participants, BA.1 titers at both D1 and D15 were markedly higher in non-cases than in booster-proximal cases (D1: 1848 AU/ml vs. 544 AU/ml; GMR = 3.40 [1.88, 6.14]; D15: 10,257 AU/ml vs. 4060 AU/ml; GMR = 2.53 [1.52, 4.21]) (Table 2). This marked difference was not observed when comparing non-cases and booster-distal cases (D1: 1848 AU/ml vs. 1465 AU/ml; GMR = 1.26 [0.54, 2.96]; D15: 10,257 AU/ml vs. 11520 AU/ml; GMR = 0.89 [0.42, 1.90]) (Fig. 2c, d, Table 2).

Supplementary Fig. 7 shows the same violin box plots as in Fig. 2c, d, i.e. D1 and D15 BA.1 titers by case/non-case status and proximal/distal to booster, except additionally stratified by sex assigned at birth (male or female). Supplementary Figs. 8–10 show complete information for all six nAb titer markers for D1, D15, and fold-rise, respectively.

### D15 (peak) titer inverse correlates with risk of COVID-19

The inverse correlations of D15 titers with COVID-19 risk were generally stronger in non-naïve than naïve participants (Cox model interaction p-values ranging from 0.002 to 0.009 for the six markers for boost-proximal cases and p-values ranging from 0.014 to 0.071 for the entire follow-up; Supplementary Table 7). In addition, we repeated the interaction tests restricted to the intersection of the middle 90% of titer values (Supplementary Table 8) and to participants with detectable D1 titers (Supplementary Table 9) in two post hoc sensitivity analyses. The results also supported interactions (Supplementary Tables 8 and 9). Given the evidence of CoR modification by status of prior COVID-19, the correlate analyses were conducted separately for naïve and non-naïve participants.

Low, Medium, and High D15 BA.1 titer subgroups were defined by tertile cut-points calculated pooling over naïve and non-naïve participants. Fig. 3 shows the covariate-adjusted cumulative incidence of COVID-19 through 188 days post-D15 for these three subgroups for naïve (n = 629) (Fig. 3a) and non-naïve (n = 356) (Fig. 3b) participants. For both the naïve and non-naïve groups, the covariate-adjusted COVID-19 risk was similar for the Low vs. Medium subgroups (Naïve: 32.3% vs. 35.1%; Non-naïve: 16.0% vs. 17.7%), and lower for the High subgroup (Naïve: 20.4%; Non-naïve: 4.0%). Fig. 4a, c show these results across the full range of D15 BA.1 titers, showing drops in risk at the highest titers. In naïve participants, the estimated probability of COVID-19 acquisition was 35.6% (32.0%–38.9%), 31.2% (23.8%–38.9%) and 23.7% (16.6%–32.7%) at D15 BA.1 titer of 1000, 5000, and 10,000 AU/ml, respectively (nonparametric model) (Fig. 4a). In non-naïve participants, the estimated probability of COVID-19 acquisition was 41.6% (28.7%–56.0%) at BA.1 titer of 1000 AU/ml and decreased to 11.3% (6.8%–42.5%) and 9.2% (6.8%–13.9%) at D15 BA.1 titer of 5000 and 10,000 AU/ml, respectively (nonparametric model) (Fig. 4c). Supplementary Figs. 11–15 plot the same risk curves for the other five nAb titer markers.

Fig. 5a, c show the covariate-adjusted COVID-19 hazard ratios per 10-fold increase in each of the six D15 titer markers. For naïve participants the results supported weak inverse correlates (HRs 0.78–0.85 per 10-fold increase, p-values = 0.059 to 0.21), with the D15 BA.1 ID50 HR (95% CI) = 0.85 (0.67, 1.09); p = 0.21 (Fig. 5a). In the analyses shown in Fig. 5, adjustment was done for FOI score and risk score. We further conducted multiple sensitivity analyses to explore the influence of three alternative covariate adjustment strategies [adjustment for: (1) only the FOI score and the ≥65 age indicator; (2) adjustment only for the ≥65 age indicator; (3) no covariate adjustment] on the results; hazard ratios of each D15 ID50 marker remained virtually unchanged across all three strategies (Supplementary Fig. 16). In a post hoc analysis, we also estimated the covariate-adjusted HR of COVID-19 per 10-fold increase in each of the six D15 titer markers in naïve participants among individuals assigned male or assigned female sex at birth (Supplementary Fig. 18a).

Fig. 4b shows the estimated covariate-adjusted HR of COVID-19 in naïve participants per 10-fold increase in D15 BA.1 titer among six subgroups (Moderna, Pfizer-BioNTech, Prototype, Omicron-Containing, Bivalent Omicron, and Monovalent Omicron); the subgroup analyses were reported only when there were at least 20 endpoints to ensure reasonable precision. The HR point estimates were largely consistent across the subgroups, with that among Prototype insert boost recipients closest to unity [1.00 (0.56, 1.78); p = 0.999] and that among Monovalent Omicron insert boost recipients the smallest [0.57 (0.30, 1.07); p = 0.081], where adjustment was done for FOI score and

**Table 1 | Detectable neutralization titer frequencies and geometric mean (GM) titers of the six nAb markers (D614G, Delta, Beta, BA.1, BA.4/BA.5, weighted average) in SARS-CoV-2 naïve participants, shown separately by non-cases and COVID-19 endpoint cases (stratified by booster-proximal cases and booster-distal cases)**

| Visit | nAb Titer Marker | Booster-Proximal COVID-19 Cases | | | Booster-Distal COVID-19 Cases | | | Non-cases | | | Comparison: Non-cases to Booster-Proximal COVID-19 Cases | | Comparison: Non-cases to Booster-Distal COVID-19 Cases | |
|---|---|---|---|---|---|---|---|---|---|---|---|---|---|---|
| | | N | Pos Resp Freq (95% CI) | GM (95% CI) | N | Pos Resp Freq (95% CI) | GM (95% CI) | N | Pos Resp Freq (95% CI) | GM (95% CI) | Pos Resp Freq Diff (Non-cases – Cases) (95% CI) | Ratio of GM (Non-cases/Cases) (95% CI) | Pos Resp Freq Diff (Non-cases – Cases) (95% CI) | Ratio of GM (Non-cases/Cases) (95% CI) |
| D1 | D614G | 122 | 99.2% (95.5%, 100.0%) | 2509 (1998, 3151) | 59 | 100.0% (93.9%, 100.0%) | 2368 (1909, 2938) | 448 | 98.4% (96.8%, 99.4%) | 2775 (2446, 3148) | -0.7% (-2.6%, 3.0%) | 1.11 (0.85, 1.45) | -1.6% (-3.2%, 4.6%) | 1.17 (0.82, 1.67) |
| D1 | Delta | 122 | 96.7% (91.8%, 99.1%) | 1127 (883, 1439) | 59 | 100.0% (93.9%, 100.0%) | 1067 (857, 1326) | 448 | 97.1% (95.1%, 98.4%) | 1311 (1149, 1496) | 0.4% (-2.7%, 5.5%) | 1.16 (0.88, 1.54) | -2.9% (-4.9%, 3.3%) | 1.23 (0.85, 1.78) |
| D1 | Beta | 122 | 93.4% (87.5%, 97.1%) | 567 (439, 733) | 59 | 91.5% (81.3%, 97.2%) | 413 (293, 581) | 448 | 92.0% (89.0%, 94.3%) | 647 (558, 751) | -1.5% (-6.2%, 4.9%) | 1.14 (0.83, 1.56) | 0.4% (-5.9%, 10.9%) | 1.57 (1.03, 2.40) |
| D1 | BA.1 | 122 | 68.9% (59.8%, 76.9%) | 155 (115, 207) | 59 | 71.2% (57.9%, 82.2%) | 134 (93.3, 191) | 448 | 75.9% (71.7%, 79.8%) | 203 (174, 236) | 7.0% (-2.1%, 16.9%) | 1.31 (0.95, 1.82) | 4.7% (-7.1%, 18.5%) | 1.52 (0.98, 2.35) |
| D1 | BA.4/5 | 122 | 50.8% (41.6%, 60.0%) | 80.4 (61.5, 105) | 59 | 45.8% (32.7%, 59.2%) | 65.7 (45.8, 94.0) | 448 | 59.4% (54.7%, 64.0%) | 110 (95.1, 128) | 8.6% (-1.7%, 18.8%) | 1.37 (1.00, 1.88) | 13.6% (-0.7%, 27.4%) | 1.68 (1.10, 2.58) |
| D1 | Wt Avg | 122 | - | 331 (260, 422) | 59 | - | 280 (211, 370) | 448 | - | 410 (359, 469) | - | 1.24 (0.93, 1.65) | - | 1.47 (1.00, 2.15) |
| D15 | D614G | 122 | 100.0% (97.0%, 100.0%) | 17,833 (15,271, 20,826) | 59 | 100.0% (93.9%, 100.0%) | 13,638 (11,246, 16,539) | 448 | 99.8% (98.8%, 100.0%) | 18,485 (16,797, 20,342) | -0.2% (-1.2%, 2.8%) | 1.04 (0.85, 1.27) | -0.2% (-1.2%, 5.8%) | 1.36 (1.03, 1.78) |
| D15 | Delta | 122 | 100.0% (97.0%, 100.0%) | 9557 (8146, 11,213) | 59 | 100.0% (93.9%, 100.0%) | 7684 (6311, 9356) | 448 | 99.8% (98.8%, 100.0%) | 10,090 (9131, 11,150) | -0.2% (-1.2%, 2.8%) | 1.06 (0.86, 1.30) | -0.2% (-1.2%, 5.8%) | 1.31 (0.99, 1.74) |
| D15 | Beta | 122 | 100.0% (97.0%, 100.0%) | 7462 (6215, 8959) | 59 | 100.0% (93.9%, 100.0%) | 5932 (4586, 7672) | 448 | 99.6% (98.4%, 99.9%) | 8208 (7305, 9223) | -0.4% (-1.6%, 2.6%) | 1.10 (0.86, 1.40) | -0.4% (-1.6%, 5.6%) | 1.38 (0.99, 1.93) |
| D15 | BA.1 | 122 | 98.4% (94.2%, 99.8%) | 3006 (2482, 3641) | 59 | 100.0% (93.9%, 100.0%) | 2685 (2096, 3440) | 448 | 98.2% (96.5%, 99.2%) | 3240 (2844, 3690) | -1.8% (-3.5%, 1.4%) | 1.08 (0.82, 1.41) | -1.8% (-3.5%, 4.4%) | 1.21 (0.83, 1.74) |
| D15 | BA.4/5 | 122 | 98.4% (94.2%, 99.8%) | 1150 (923, 1433) | 59 | 98.3% (90.9%, 100.0%) | 963 (736, 1260) | 448 | 95.8% (93.5%, 97.4%) | 1315 (1156, 1496) | -2.6% (-5.3%, 1.9%) | 1.14 (0.87, 1.50) | -2.5% (-5.4%, 5.0%) | 1.37 (0.95, 1.97) |
| D15 | Wt Avg | 122 | - | 4247 (3576, 5043) | 59 | - | 3513 (2808, 4395) | 448 | - | 4628 (4149, 5162) | - | 1.09 (0.87, 1.37) | - | 1.32 (0.97, 1.80) |

Wt Avg = Maximum diversity-weighted geometric mean of the five nAb titers D614G reference, Beta, Delta, Omicron BA.1, and Omicron BA.4/BA.5.

Non-cases: No evidence of SARS-CoV-2 infection at 7 days post-D15 through to 188 days post-D15.

Booster-proximal cases: COVID-19 endpoint between 7 and 91 days post-D15 visit; booster-distal cases: COVID-19 endpoint between 92 and 188 days post-D15 visit.

Pos Resp Freq: Percent with nAb titer above the limit of detection (LoD) = 40 AU/ml. Diff = Difference.

AU arbitrary units, GMT geometric mean titer.

**Table 2 | Detectable neutralization titer frequencies and geometric mean (GM) titers of the six nAb markers (D614G, Delta, Beta, BA.1, BA.4/BA.5, weighted average) in SARS-CoV-2 non-naïve participants, shown separately by non-cases and COVID-19 endpoint cases (stratified by booster-proximal cases and booster-distal cases)**

| Visit | nAb Titer Marker | Booster-Proximal COVID-19 Cases | | | Booster-Distal COVID-19 Cases | | | Non-cases | | | Comparison: Non-cases to Booster-Proximal COVID-19 Cases | | Comparison: Non-cases to Booster-Distal COVID-19 Cases | |
|---|---|---|---|---|---|---|---|---|---|---|---|---|---|---|
| | | N | Pos Resp (95% CI) | GM (95% CI) | N | Pos Resp Freq (95% CI) | GM (95% CI) | N | Pos Resp Freq (95% CI) | GM (95% CI) | Pos Resp Freq Diff (Non-cases – Cases) (95% CI) | Ratio of GM (Non-cases/Cases) (95% CI) | Pos Resp Freq Diff (Non-cases – Cases) (95% CI) | Ratio of GM (Non-cases to Cases) (95% CI) |
| D1 | D614G | 22 | 95.5% (77.2%, 99.9%) | 4935 (2310, 10,539) | 10 | 100.0% (69.2%, 100.0%) | 8428 (5666, 12,536) | 324 | 99.1% (97.3%, 99.8%) | 11,944 (10,513, 13,571) | 3.6% (-1.1%, 21.9%) | 2.42 (1.44, 4.08) | -0.9% (-2.7%, 29.9%) | 1.42 (0.68, 2.93) |
| D1 | Delta | 22 | 90.9% (70.8%, 98.9%) | 1926 (872, 4257) | 10 | 100.0% (69.2%, 100.0%) | 4945 (3329, 7346) | 324 | 99.1% (97.3%, 99.8%) | 6040 (5324, 6853) | 8.2% (0.0%, 28.2%) | 3.14 (1.86, 5.27) | -0.9% (-2.7%, 29.9%) | 1.22 (0.60, 2.51) |
| D1 | Beta | 22 | 95.5% (77.2%, 99.9%) | 1925 (920, 4025) | 10 | 100.0% (69.2%, 100.0%) | 3262 (1847, 5761) | 324 | 99.1% (97.3%, 99.8%) | 4643 (4034, 5344) | 3.6% (-1.1%, 21.9%) | 2.41 (1.37, 4.25) | -0.9% (-2.7%, 29.9%) | 1.42 (0.64, 3.18) |
| D1 | BA.1 | 22 | 90.9% (70.8%, 98.9%) | 544 (277, 1070) | 10 | 100.0% (69.2%, 100.0%) | 1465 (686, 3132) | 324 | 97.8% (95.6%, 99.1%) | 1848 (1593, 2145) | 6.9% (-1.3%, 27%) | 3.40 (1.88, 6.14) | -2.2% (-4.4%, 28.7%) | 1.26 (0.54, 2.96) |
| D1 | BA.4/5 | 22 | 81.8% (59.7%, 94.8%) | 309 (155, 617) | 10 | 100.0% (69.2%, 100.0%) | 849 (561, 1283) | 324 | 97.5% (95.2%, 98.9%) | 1158 (1012, 1324) | 15.7% (2.5%, 37.9%) | 3.75 (2.18, 6.44) | -2.5% (-4.8%, 28.4%) | 1.36 (0.63, 2.94) |
| D1 | Wt Avg | 22 | - | 985 (514, 1889) | 10 | - | 2265 (1393, 3682) | 324 | - | 3033 (2665, 3452) | - | 3.08 (1.83, 5.18) | - | 1.34 (0.64, 2.80) |
| D15 | D614G | 22 | 100.0% (84.6%, 100.0%) | 18,578 (13,116, 26,315) | 10 | 100.0% (69.2%, 100.0%) | 37,681 (21,529, 65,949) | 324 | 99.7% (98.3%, 100.0%) | 37,075 (33,597, 40,912) | -0.3% (-1.7%, 15.1%) | 2.00 (1.36, 2.94) | -0.3% (-1.7%, 30.5%) | 0.98 (0.56, 1.73) |
| D15 | Delta | 22 | 100.0% (84.6%, 100.0%) | 10,827 (7266, 16,132) | 10 | 100.0% (69.2%, 100.0%) | 18,938 (9654, 37,149) | 324 | 99.7% (98.3%, 100.0%) | 21,669 (19,574, 23,987) | -0.3% (-1.7%, 15.1%) | 2.00 (1.34, 2.99) | -0.3% (-1.7%, 30.5%) | 1.14 (0.64, 2.05) |
| D15 | Beta | 22 | 100.0% (84.6%, 100.0%) | 10,765 (6887, 16,828) | 10 | 100.0% (69.2%, 100.0%) | 22,257 (10,306, 48,065) | 324 | 99.7% (98.3%, 100.0%) | 22,483 (20,059, 25,200) | -0.3% (-1.7%, 15.1%) | 2.09 (1.33, 3.27) | -0.3% (-1.7%, 30.5%) | 1.01 (0.52, 1.95) |
| D15 | BA.1 | 22 | 100.0% (84.6%, 100.0%) | 4060 (2863, 5759) | 10 | 100.0% (69.2%, 100.0%) | 11,520 (4833, 27,460) | 324 | 99.1% (97.3%, 99.9%) | 10,257 (8995, 11,697) | -0.9% (-2.7%, 14.5%) | 2.53 (1.52, 4.21) | -0.9% (-2.7%, 29.9%) | 0.89 (0.42, 1.90) |
| D15 | BA.4/5 | 22 | 100.0% (84.6%, 100.0%) | 1822 (1168, 2842) | 10 | 100.0% (69.2%, 100.0%) | 4210 (2136, 8299) | 324 | 98.8% (96.9%, 99.7%) | 4934 (4336, 5615) | -1.2% (-3.1%, 14.2%) | 2.71 (1.63, 4.49) | -1.2% (-3.1%, 29.6%) | 1.17 (0.56, 2.46) |
| D15 | Wt Avg | 22 | - | 5697 (3961, 8193) | 10 | - | 12,819 (6396, 25,693) | 324 | - | 13,207 (11,841, 14,731) | - | 2.32 (1.51, 3.55) | - | 1.03 (0.55, 1.93) |

Wt Avg =Maximum diversity-weighted geometric mean of the five nAb titers D614G reference, Beta, Delta, Omicron BA.1, and Omicron BA.4/BA.5.
Non-cases: No evidence of SARS-CoV-2 infection at 7 days post-D15 through to 188 days post-D15.
Booster-proximal cases: COVID-19 endpoint between 7 and 91 days post-D15 visit; booster-distal cases: COVID-19 endpoint between 92 and 188 days post-D15 visit.
Pos Resp Freq: Percent with nAb titer above the limit of detection (LoD) = 40 AU/ml. Diff = Difference.
AU arbitrary units, GMT geometric mean titer.

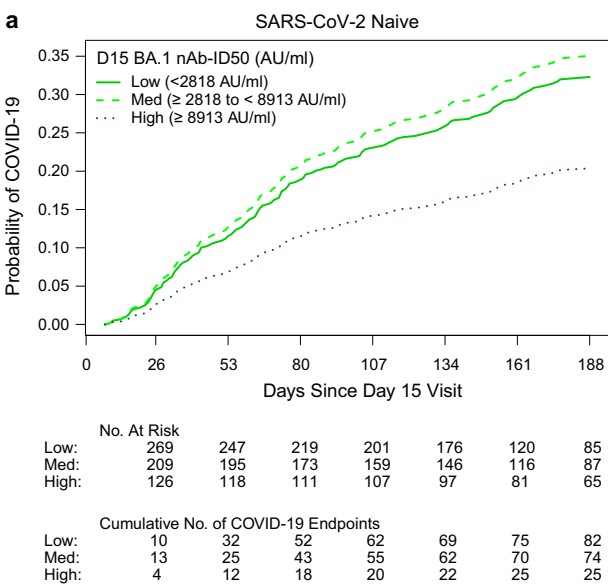

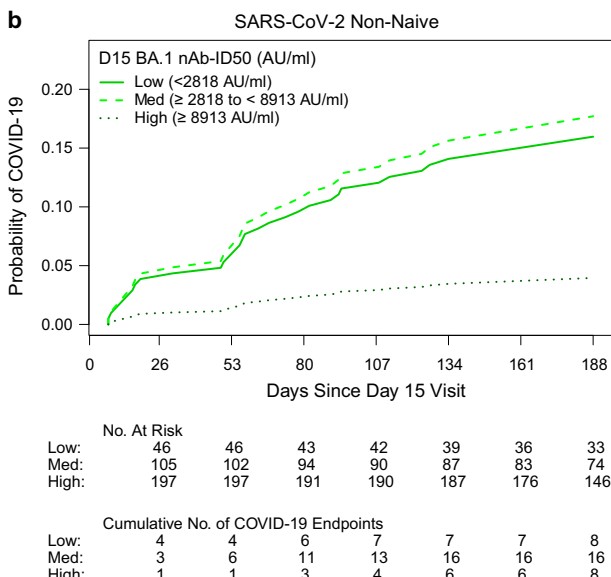

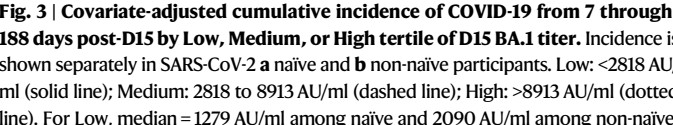

**Fig. 3 | Covariate-adjusted cumulative incidence of COVID-19 from 7 through 188 days post-D15 by Low, Medium, or High tertile of D15 BA.1 titer.** Incidence is shown separately in SARS-CoV-2 **a** naïve and **b** non-naïve participants. Low: <2818 AU/ ml (solid line); Medium: 2818 to 8913 AU/ml (dashed line); High: >8913 AU/ml (dotted line). For Low, median = 1279 AU/ml among naïve and 2090 AU/ml among non-naïve; for Medium, median = 4530 AU/ml among naïve and 5446 AU/ml among non-naïve; for High, median = 15,503 AU/ml among naïve and 18768 AU/ml among non-naïve. Analyses adjusted for the force of infection score and risk score. AU, arbitrary units; nAb-ID50, 50% inhibitory dilution neutralizing antibody titer. Source data are provided as a Source Data file.

risk score. Results were largely unchanged (Supplementary Fig. 17a, c, e) in sensitivity analyses exploring the influence of the same three alternative covariate adjustment strategies as described above.

For non-naïve participants, all six D15 titer markers were inverse correlates of risk (HRs 0.39–0.52 per 10-fold increase, p < 0.001–0.005), with the D15 BA.1 ID50 HR = 0.52 (0.33, 0.82); p = 0.005 (Fig. 5c). Among the six pre-specified subgroups, only Omicron-containing (N = 285) insert boost recipients had more than 20 endpoints: the HR per 10-fold increase in D15 BA.1 titer was nearly identical to that among all one-dose mRNA boost recipients (Fig. 4d). Results were also nearly identical in the sensitivity analysis to explore the influence of different covariate adjustment approaches (Supplementary Fig. 17b, d, f).

The correlate analyses were repeated for the fold-rise markers (Supplementary Figs. 19 and 20), including by subgroups where applicable based on the 20-endpoint criterion (Supplementary Figs. 21 and 22). Among naïve participants a larger BA.1 ID50 fold-rise was associated with higher COVID-19 risk with an estimated HR of 1.34 (1.04, 1.73) (Supplementary Fig. 19), while for non-naïve participants the estimated HR was 1.59 (0.89, 2.86) (Supplementary Fig. 20). These fold-rise correlates are challenging to interpret compared to D15 absolute-level correlates, given that larger post-boost fold-rises in antibody titer were seen in the data analyzed here (as shown in Supplementary Fig. 23 for BA.1).

To enable comparison to the results of the immune correlates analysis of a third dose of mRNA-1273 in the COVE trial[29], we defined a second endpoint ("COVE COVID-19"), based on the clinical criteria for the COVID-19 endpoint used in the COVE primary efficacy analyses[2,30] as well as previous COVE immune correlates analyses[27,29,31–33]. Of the 213 COVID-19 endpoints analyzed above, 191 (89.7%) met the COVE clinical criteria and supportive laboratory criteria for a COVE endpoint. We repeated the D15 correlates analyses above using the COVE endpoint and found that the results (Supplementary Fig. 24) were virtually the same as for the COVID-19 endpoint.

### Exposure-proximal correlates of risk
Titer dynamics were modeled using a mixed effects model based on observed titer readouts in each booster arm (see the SAP).

Supplementary Table 10 summarizes the estimated half-life of BA.1 titer among naïve and non-naïve participants in each boost arm. Averaged over the arms, half-life was estimated to be 66 days (95% CI: 58 to 78) for naïve participants and 80 days (95% CI: 67 to 101 days) for non-naïve participants. Supplementary Fig. 25 graphs the measured BA.1 log10 ID50 titers at D1, D29, D91, and D181 for 10 randomly selected individuals (4 were naïve and 6 were non-naïve) along with their predicted BA.1 log10 ID50 titer over time, based on a biphasic antibody decay model. Predicted titers aligned closely with the observed BA.1 ID50 readouts, providing support for the antibody decay model. Supplementary Fig. 26 additionally plots measured versus predicted (based on the biphasic antibody decay model) log10 ID50 titers at D15, D29, D91, and D181, separately for BA.1, Beta, Delta, BA.4/BA.5, D614G, and the weighted average marker. High concordance was seen between the observed and predicted values for each variant/marker, with Pearson correlations ranging from 0.95 to 0.98 and reliability ratios ranging from 90% to 95%. The predicted-at-exposure titer against each antigen was then correlated with COVID-19 based on a calendar-time Cox regression model (see the SAP).

Fig. 6a, c show the estimated HR relative to the minimum predicted titer through 188 days post-D15 across a range of predicted-at-exposure BA.1 titers among naïve (n = 629) and non-naïve (n = 356) participants. The HR decreased as predicted-at-exposure BA.1 titer increased. Fig. 6b, d show the HR of COVID-19 per 10-fold increase in predicted-at-exposure BA.1 ID50 in naïve and non-naïve participants, respectively, as well as in pre-specified subgroups. Compared to D15 (~peak) BA.1 ID50, predicted-at-exposure BA.1 titer showed a stronger inverse association with the COVID-19 endpoint in terms of HRs farther from 1, particularly in non-naïve participants [naïve participants: 0.74 (0.59, 0.94), p = 0.016; non-naïve participants: 0.41 (0.23, 0.64), p < 0.001].

We conducted a sensitivity analysis to explore the influence of an alternative covariate adjustment strategy (no covariate adjustment) on the results; HRs of each predicted-at-exposure ID50 marker remained virtually unchanged (Supplementary Fig. 27). We also repeated the predicted-at-exposure correlates analyses using the COVE endpoint and found that the results (Supplementary Fig. 28) were virtually the same as for the COVID-19 endpoint.

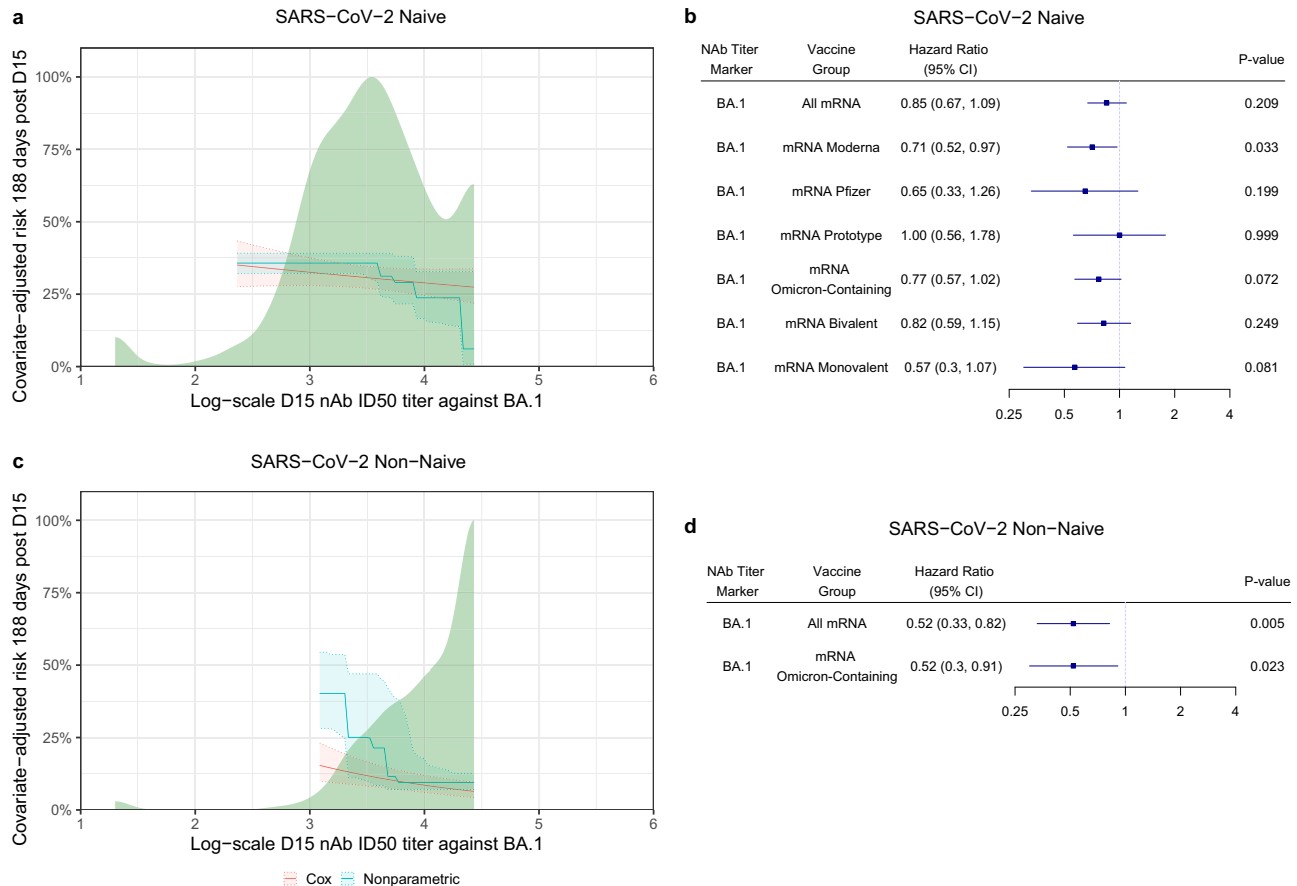

**Fig. 4 | Covariate-adjusted cumulative incidence of COVID-19 by D15 BA.1 titer and covariate-adjusted hazard ratios of COVID-19 per 10-fold increase in D15 BA.1 titer. a, c** Covariate-adjusted cumulative incidence of COVID-19 by D15 BA.1 titer at 188 days post-D15, estimated using a Cox model (orange line) or a non-parametric method (turquoise line), in SARS-CoV-2 **a** naïve and **c** non-naïve participants. Both curves were restricted to the middle 95% of the marker distribution. Shaded regions represent 95% confidence intervals. The green shaded region is a smoothed histogram of log10 D15 BA.1 titer (AU/ml). Analyses adjusted for the force of infection score and risk score. **b, d** Cox model covariate-adjusted hazard ratios of COVID-19 per 10-fold increase in D15 BA.1 titer (AU/ml), in **b** naïve participants (All mRNA N = 629) and in the designated subgroups (mRNA Moderna N = 306, mRNA Pfizer-BioNTech N = 131, mRNA Prototype N = 105, mRNA Omicron-containing N = 459, mRNA Bivalent N = 352, mRNA Monovalent N = 107), or **d** non-naïve participants (all mRNA N = 356) and in the designated subgroup (mRNA Omicron-containing N = 285). Point estimates, 95% confidence intervals (CIs), and 2-sided Wald p-values are shown. P-values were not adjusted for multiple comparisons. Subgroup analyses were only conducted when the number of COVID-19 endpoints was equal to or exceeded 20, to ensure reasonable precision. nAb-ID50, 50% inhibitory dilution neutralizing antibody titer. Source data are provided as a Source Data file.

## Exploratory analysis of pre and post-COVID-19 titer

Because of the substantial difference in the relationship between neutralization titer and COVID-19 risk by prior infection, we contrasted the log10 BA.1 titer before and after acquisition of COVID-19 in COVAIL in naïve cases compared to non-naïve cases. Supplementary Fig. 29a shows the log10 titers closest in time before (average: 34 days before) and after (average: 39 days after) the COVID-19 onset date for naïve and non-naïve cases. Naïve cases tended to have an increase following COVID-19 while non-naïve cases did not. Supplementary Fig. 29b provides a violin plot of the slopes by prior infection status. Adjusting for pre-COVID-19 titer being above or below the median in the non-naïve arm, the estimated change in BA.1 titer over the average time interval shown in Supplementary Fig. 29a was a 394% increase (95% CI: 296%, 515%) for naïve cases and a 149% increase (95% CI: 18% decrease, 272% increase) for non-naïve cases. Results were similar using the weighted average of neutralization titers (Supplementary Fig. 30).

## Lineage-specific correlates of risk

Fig. 7a shows the distribution of SARS-CoV-2 lineages over calendar-time among COVID-19 endpoint cases during the study follow-up, with Supplementary Tables 11–23 further detailing the lineage distribution in each of the 13 single-dose mRNA arms. Of the 213 COVID-19 endpoints in the per-protocol correlates cohort, sequences were successfully obtained and lineages were identified for 154 (72.3%). The proportion of viruses that were successfully sequenced was similar across the 3 stages: 74.8% (107/143) for Stage 1, 67.3% (37/55) for Stage 2, and 66.7% (10/15) for Stage 4. The lineages of the other 59 endpoints were imputed based on the modal circulating lineage on the date of COVID-19 onset according to the GISAID database (matched by date and state or District of Columbia). BA.4 and BA.5 were the most prevalent lineages causing COVID-19 endpoints, together accounting for 65.7% (94/143) endpoints in Stage 1, 89.1% (49/55) in Stage 2, and 80% (12/15) in Stage 4. Thus, our lineage-specific correlates analyses focused on BA.4/BA.5 COVID-19, combining BA.4 and BA.5 into a single genotype as justified by their Spike proteins having identical amino acid sequences[34], with "lineage-matched" correlates analysis studying BA.5 titer as a correlate of BA.4/BA.5 COVID-19.

Fig. 7b, d report the estimated HRs of BA.4/BA.5 COVID-19 per 10-fold increase of D15 BA.4/BA.5 titer in naïve and non-naïve participants, respectively, in all mRNA arms pooled and in each subgroup with at least 15 BA.4/BA.5 COVID-19 endpoints. Among naïve participants, the HR (95% CI) of lineage-matched COVID-19 was 0.79 (0.59, 1.05; p = 0.102) and a significant inverse correlation was seen in non-naïve participants with HR = 0.45 (0.26, 0.77; p = 0.004). To investigate

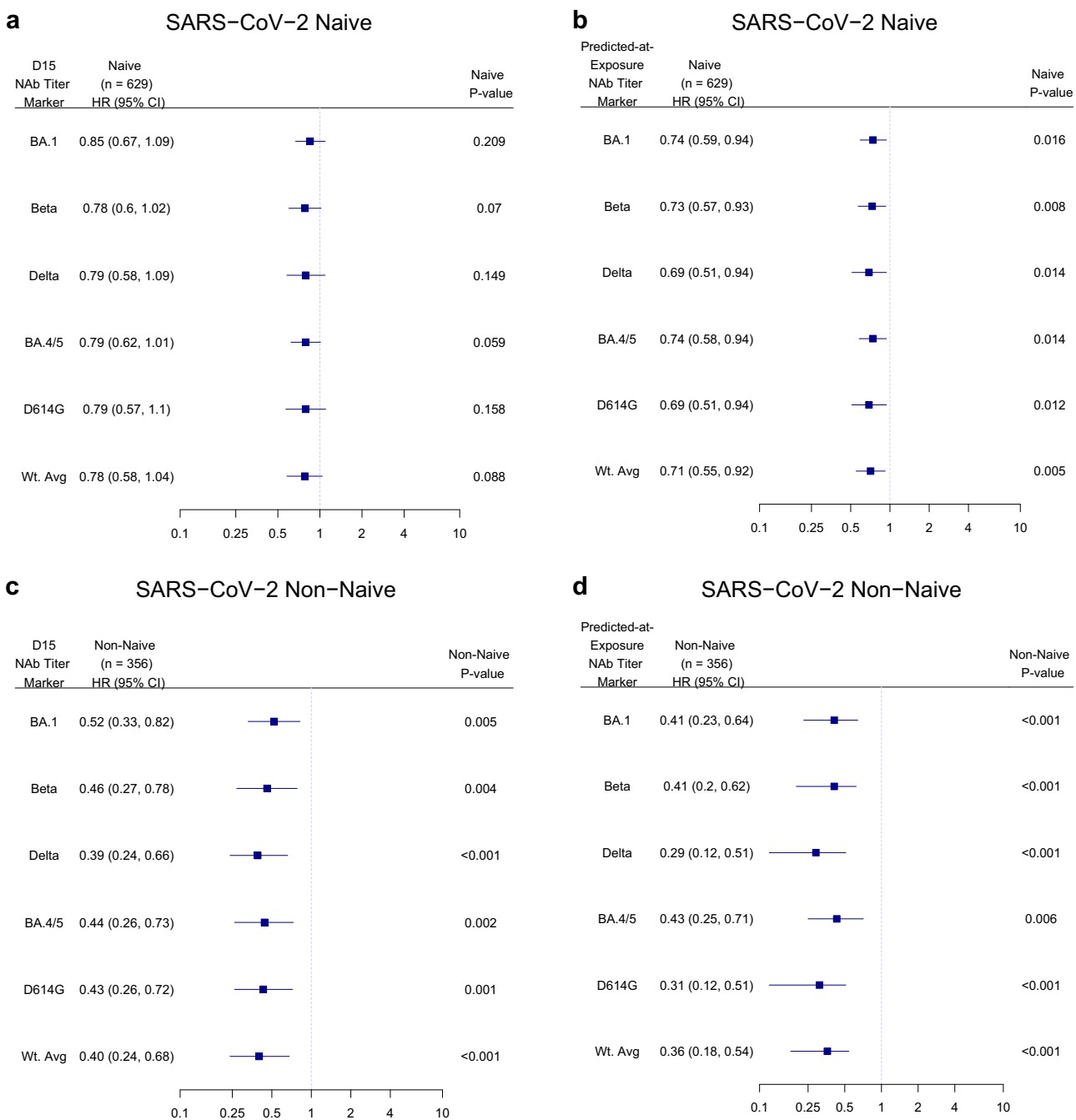

**Fig. 5 | Cox model covariate-adjusted hazard ratios of COVID-19 per 10-fold increase in D15 or predicted-at-exposure neutralizing antibody titer, shown separately in SARS-CoV-2 naïve and non-naïve participants. a, c** D15 titer (AU/ml) for each marker BA.1, Beta, Delta, BA.4/BA.5, D614G, or weighted average (Wt. Avg.); **b, d** predicted-at-exposure titer (AU/ml) for each marker BA.1, Beta, Delta, BA.4.BA.5, D614G, or Wt. Avg for follow-up 7–188 days post-D15. Results are shown in **a, b** naïve participants: **a** BA.1 (N = 629), Beta (N = 629), Delta (N = 629), BA.4/BA.5 (N = 629), D614G (N = 629), or weighted average (Wt. Avg.) (N = 629); **b** BA.1 (N = 629), Beta (N = 629), Delta (N = 629), BA.4/BA.5 (N = 605), D614G (N = 629), or weighted average (Wt. Avg.) (N = 629) or **c, d** non-naïve participants: **c** BA.1 (N = 356), Beta (N = 356), Delta (N = 356), BA.4/BA.5 (N = 356), D614G (N = 356), or weighted average (Wt. Avg.) (N = 356); **d** BA.1 (N = 356), Beta (N = 356), Delta (N = 356), BA.4/BA.5 (N = 351), D614G (N = 356), or weighted average (Wt. Avg.) (N = 356). D15 analyses adjusted for the force of infection score and risk score. Exposure-proximal analyses are calendar-time-based and adjusted for the risk score. Point estimates, 95% confidence intervals (CIs), and 2-sided Wald p-values are shown for the D15 titer analyses, and bootstrap percentile 95% CIs and 2-sided p-values are shown for the predicted-at-exposure analyses. P-values were not adjusted for multiple comparisons. Wt. Avg. = Maximum diversity-weighted geometric mean of the five nAb titers D614G reference, Beta, Delta, Omicron BA.1, and Omicron BA.4/BA.5. AU, arbitrary units; nAb-ID50, 50% inhibitory dilution neutralizing antibody titer.

whether lineage-matched nAb titer is a stronger correlate than lineage-unmatched nAb titer, we performed the same analyses using D15 D614G titer. As shown in Fig. 7c, e and b, d, point estimates and 95% CIs for the BA.4/BA.5 COVID-19 HR per 10-fold increase in D15 D614G (lineage-unmatched) titer [naïve: HR = 0.77 (0.53, 1.13; p = 0.184); non-naïve: HR = 0.45 (0.25, 0.82; p = 0.009)] were similar or even identical to those obtained in the lineage-matched analysis. Consistent results were obtained for the subgroup analyses. However, the lineage-

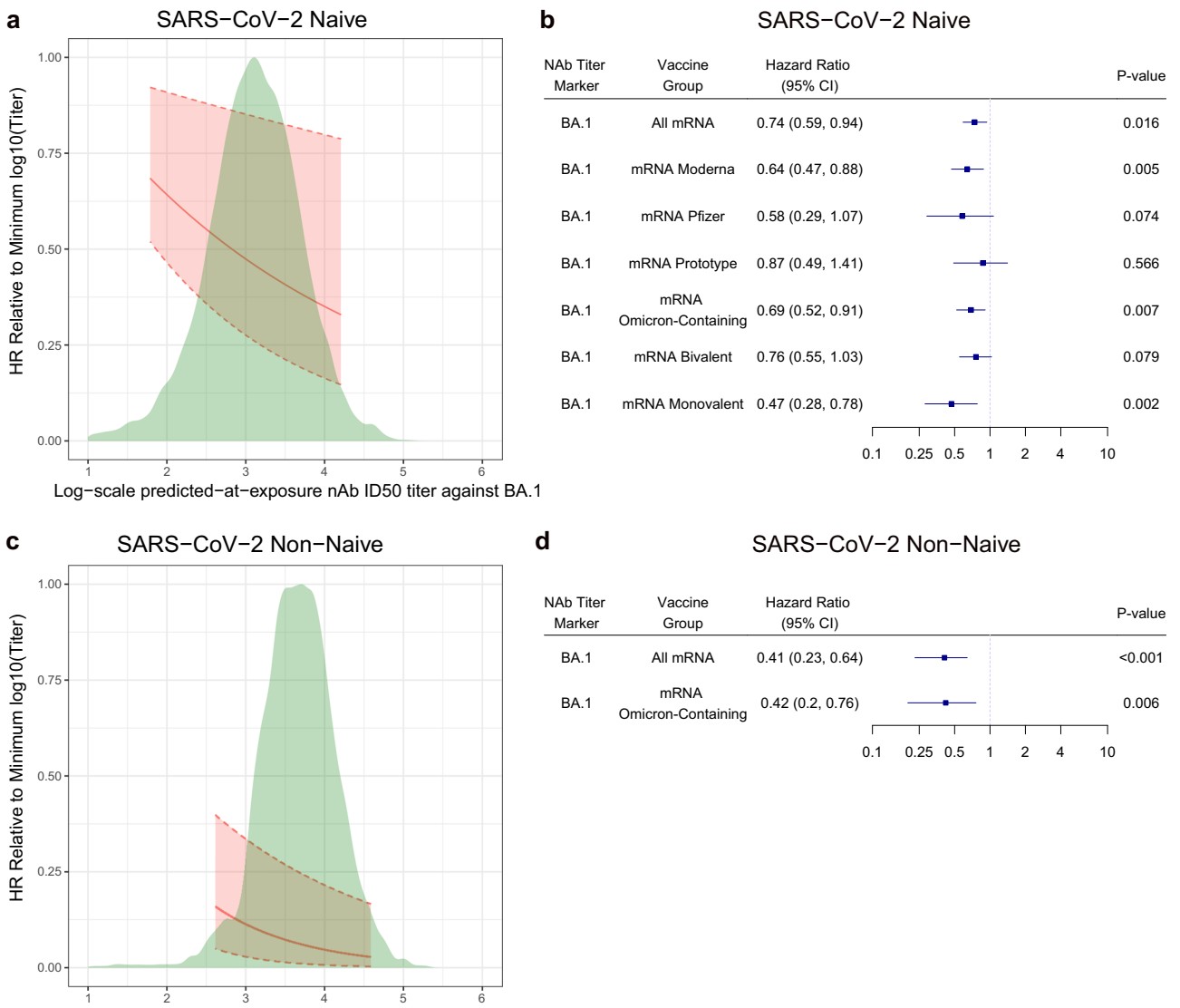

**Fig. 6 | Hazard ratio of COVID-19 relative to minimum predicted-at-exposure BA.1 titer by predicted-at-exposure BA.1 titer and calendar-time-based hazard ratios of COVID-19 per 10-fold increase in predicted-at-exposure BA.1 titer, shown separately in SARS-CoV-2 naïve and non-naïve participants. a, c** Hazard ratio of COVID-19 relative to minimum log10 predicted-at-exposure BA.1 titer (AU/ml) by predicted-at-exposure BA.1 titer (AU/ml), estimated using a Cox model, in **a** naïve participants and **c** non-naïve participants for follow-up 7–188 days post-D15. The curve was restricted to the middle 95% of the marker distribution. Shaded regions represent 95% confidence intervals. The green shaded region is a kernel density estimate of log10 predicted-at-exposure BA.1 titer (AU/ml). Analyses adjusted for risk score. **b, d** Calendar-time-based Cox model covariate-adjusted

hazard ratios of COVID-19 per 10-fold increase in predicted-at-exposure BA.1 titer (AU/ml), in **b** naïve participants (All mRNA N = 629) and in the designated subgroups (mRNA Moderna N = 306, mRNA Pfizer-BioNTech N = 131, mRNA Prototype N = 105, mRNA Omicron-containing N = 459, mRNA Bivalent N = 352, mRNA Monovalent N = 107), or **d** non-naïve participants (All mRNA N = 356) and in the designated subgroup (mRNA Omicron-containing N = 285). Point estimates, 95% bootstrap percentile confidence intervals (CIs), and 2-sided bootstrap p-values are shown. P-values were not adjusted for multiple comparisons. Subgroup analyses were only conducted when the number of endpoints was equal to or exceeded 20, to ensure sufficient precision. AU arbitrary units nAb-ID50 50% inhibitory dilution neutralizing antibody titer.

matched analyses displayed marginally improved precision as the lineage-matched-based CIs were in general narrower than those obtained based on D614G titer.

In the exposure-proximal analyses, the inverse correlations of predicted-at-exposure BA.4/BA.5 titer with BA.4/BA.5 COVID-19 were consistent with those of lineage-matched D15 (peak) BA.4/BA.5 titer, with highly similar HR point estimates (95% CI) and recapitulated the result of stronger inverse correlations in non-naïve participants [HR: 0.40 (0.22, 0.67; p = 0.0043)] compared to naïve participants [HR: 0.73 (0.56, 0.94; p = 0.009)] (Supplementary Fig. 31). Similar results were also seen in the subgroup analyses. When predicted-at-exposure titer against D614G was assessed as a correlate of BA.4/BA.5 COVID-19 in a lineage-unmatched analysis, similar to the D15 (peak) results, lineage matching

did not improve the strength of the correlate [non-naïve: HR = 0.33 (0.14, 0.55; p < 0.001); naïve: HR = 0.69 (0.49, 0.97; p = 0.026)].

**Correlates of protection analysis in Stage 2 participants**
Among Stage 2 Pfizer-BioNTech recipients, Omicron-containing vaccines (Beta + Omicron, Omicron, and Omicron + Wildtype/Prototype) resulted in lower COVID-19 incidence compared to the Prototype vaccine (0.36 per 100 person-years vs. 0.92 per 100 person-years; calendar-time Cox model p = 0.001). A correlation of protection analysis was conducted to assess the D15 ID50 markers as controlled relative vaccine efficacy CoPs and as mediator CoPs[23] of relative vaccine efficacy (Omicron-Containing vs. Prototype). Fig. 8 displays the controlled relative vaccine efficacy curves that contrast the COVID-19

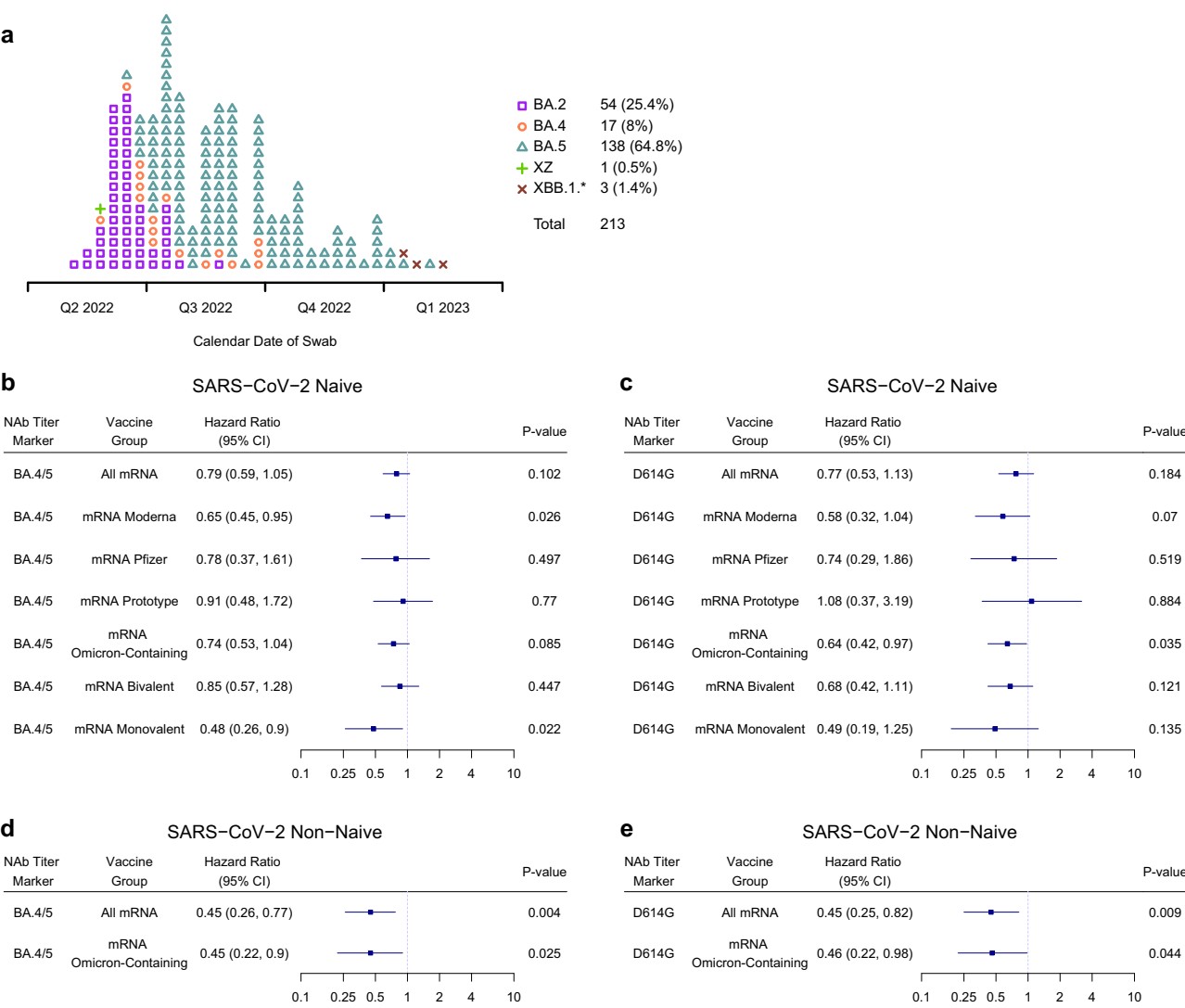

**Fig. 7 | SARS-CoV-2 lineages of COVID-19 endpoints and hazard ratios of BA.4/BA.5 COVID-19 per 10-fold increase in neutralizing antibody titer, shown separately in SARS-CoV-2 naïve and non-naïve participants. a** SARS-CoV-2 lineages among the 213 COVID-19 endpoints during follow-up 7–188 days post-D15, shown by calendar date of COVID-19 onset swab collection. Square: BA.2, circle: BA.4, triangle: BA.5, plus sign: XZ, rotated plus sign, XBB.1*. The asterisk designates the inclusion of any descendent lineages of XBB.1 (e.g. XBB.1.1). **b**–**e** Cox model covariate-adjusted hazard ratios of BA.4/BA.5 COVID-19 per 10-fold increase in **b**, **d** D15 BA.4/BA.5 titer (AU/ml) or **c**, **e** D15 D614G titer (AU/ml), in **b**, **c** naïve participants: **b** (All mRNA N = 629) and in the designated subgroups (mRNA Moderna N = 306, mRNA Pfizer-BioNTech N = 131, mRNA Prototype N = 105, mRNA Omicron-

containing N = 459, mRNA Bivalent N = 352, mRNA Monovalent N = 107); **c** (All mRNA N = 629) and in the designated subgroups (mRNA Moderna N = 306, mRNA Pfizer-BioNTech N = 131, mRNA Prototype N = 105, mRNA Omicron-containing N = 459, mRNA Bivalent N = 352, mRNA Monovalent N = 107)or **d**, **e** non-naïve participants: **d** (All mRNA N = 356) and in the designated subgroup (mRNA Omicron-containing N = 285); **e** (All mRNA N = 356) and in the designated subgroup (mRNA Omicron-containing N = 285). Point estimates, 95% confidence intervals (CIs), and 2-sided Wald p-values are shown. P-values were not adjusted for multiple comparisons. Subgroup analyses were only conducted when the number of BA.4/BA.5 COVID-19 endpoints were equal to or exceeded 15, to ensure reasonable precision. AU arbitrary units.

cumulative incidence through 188 days post-D15 for the hypothetical assignment of all participants to an Omicron-containing vaccine and the D15 titer to a given fixed value with the overall cumulative incidence of Prototype vaccine recipients. The Cox-model-based controlled relative vaccine efficacy (95% CI) was estimated to be 51.7% (8.3%, 74.6%), 60.3% (30.3%, 77.3%), and 67.4% (38.5%, 82.7%) when D15 titer against BA.1 equals 1000, 3160, and 10,000 AU/ml, respectively; the nonparametric analysis yielded controlled relative vaccine efficacy estimates spanning from 26% at BA.1 ID50 = 1000 AU/ml to 100% at BA.1 ID50 = 10,000 AU/ml (Fig. 8a). Qualitatively similar results were observed for the D15 weighted average nAb titer (Fig. 8b) and for D15 titer against other antigens.

A causal mediation analysis done by the method of Benkeser et al. [35] (see the SAP) shows that D15 titer against BA.1 mediated an estimated 27.3% (95% CI: 8.5%–46.0%) of the total Omicron-

containing vs. Prototype relative vaccine efficacy among Stage 2 participants, and the result remained largely the same when restricted to naïve participants (proportion mediated: 28.4%; 95% CI: 12.4%–44.3%). On the other hand, D15 ID50 against D614G mediated an estimated 8.6% (95% CI: −1.2%–18.3%) of the total relative vaccine efficacy among Stage 2 participants and 15.7% (95% CI: 8.5%–22.8%) among naïve participants. Mediation for non-naïve participants could not be assessed based on too few COVID-19 endpoints.

## Discussion

The COVAIL trial was designed to define the immunogenicity landscape for different vaccine platforms and inserts among SARS-CoV-2 naïve and non-naïve volunteers who had received one booster dose following a primary immunization series. Sufficient cases of largely

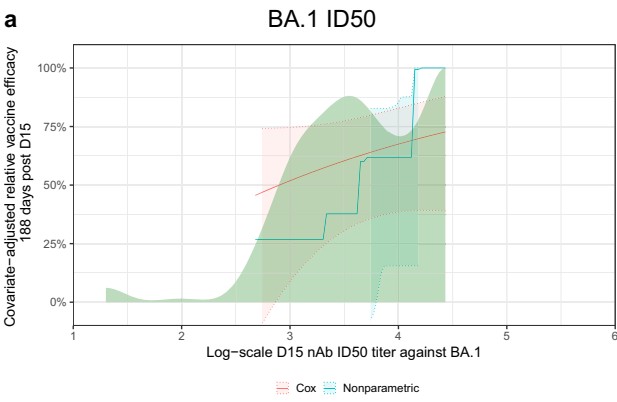

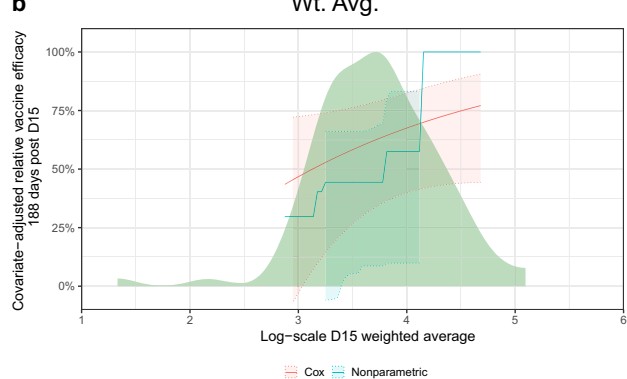

**Fig. 8 | Controlled relative vaccine efficacy curves.** The curves show the contrast in the estimated COVID-19 cumulative incidence at 188 days post-D15 for the hypothetical assignment of all participants to an Omicron-containing vaccine and the **a** D15 BA.1 titer or **b** D15 weighted average to a given fixed value with the overall estimated cumulative incidence of Prototype vaccine recipients [Stage 2 Pfizer-BioNTech vaccine arms 7 (Prototype), 8 (Beta + Omicron), 9 (Omicron), 12 (Prototype + Omicron)]. Estimation of the curve was restricted to the middle 95% of the marker distribution. Shaded regions represent 95% confidence intervals. The green shaded region is a smoothed histogram of log10 **a** D15 BA.1 titer (AU/ml) or **b** D15 weighted average titer. Analyses adjusted for the force of infection score, risk score, and baseline naïve/non-naïve status. Wt. Avg. = Maximum diversity-weighted geometric mean of the five neutralizing antibody titers D614G, Beta, Delta, BA.1, and BA.4/BA.5. AU, arbitrary units; nAb-ID50, 50% inhibitory dilution neutralizing antibody titer. Source data are provided as a Source Data file.

BA.2, BA.4, and BA.5 COVID-19 accrued among recipients of a second booster such that extensive immune correlates analyses contrasting inserts and assays to different lineages could be performed. In this analysis, we focused on the Pfizer-BioNTech and Moderna mRNA vaccines. Because prior infection substantially modified the association of neutralization titer with COVID-19 risk, results were reported separately for naïve and non-naïve participants.

COVID-19 cumulative incidence for participants in the High D15 BA.1 neutralizing antibody titer tertile subgroup remained much lower over time compared to participants in the Low or Middle D15 BA.1 neutralizing antibody tertile subgroups, whereas in contrast there was very little separation between the Low and Middle tertile cumulative incidence curves. This finding, which was seen both in naïve and non-naïve participants, suggests there may be a D15 BA.1 neutralizing antibody threshold associated with protection.

In analyses of naïve participants combined over all vaccine arms, the hazard ratios of COVID-19 (over ~6 months following the second booster) per 10-fold increase in D15 neutralizing antibody titer ranged from 0.78 to 0.85 across the six markers − suggestive of weak inverse correlates of COVID-19 risk − but with all 95% CI upper limits slightly higher than 1. Previous correlate studies in naïve participants following the primary immunization series with Prototype lineages have consistently shown a relationship between D614G neutralization titer and incidence of COVID-19, with hazard ratios of around 0.40[36]. This was also shown in population-level correlates meta-analyses[37,38]. A correlate analysis following a first mRNA booster dose also showed a significant association, with a hazard ratio for BA.1 titer of 0.31[29]. In the earlier correlate studies of the Moderna mRNA-1273 (Prototype) vaccine, post-dose two D614G neutralization titers ranged from about 19 AU/ml to 153,000 AU/ml in Monogram assay units (The IU50/ml values in Gilbert et al. can be back-converted to AU/ml values by dividing by 0.0653, as described in the Supplementary Material of Gilbert et al.[27]). In COVAIL, D15 BA.1 titer ranged from about 20 AU/ml to 27,000 AU/ml in Monogram assay units, demonstrating substantial overlap with the earlier studies. Further, there were over 150 breakthrough COVID-19 endpoints in naïve participants, so the analysis had ample precision to characterize a relationship in naïve participants, with a clear conclusion of weaker inverse correlates than previously found. While the reason for the weaker relationship in COVAIL is not clear, a potential reason may be related to immune imprinting. The primary immunization for naïve participants in previous studies and for COVAIL was from Prototype vaccines. In previous studies, the

circulating strains of SARS-CoV-2 were more similar to the vaccine strain than for COVAIL which had a preponderance of BA.2 and BA.5 strains. The immune correlate relationship may weaken with greater discordance between imprinted strain and circulating strain.

In contrast, significant hazard ratios below one ranging from 0.39 to 0.52 for the six neutralizing titer markers at D15 were observed in non-naïve participants, where these correlates appeared strongest for COVID-19 over the first three months of follow-up. These inverse correlates results were robust to other methods of analysis and assay strains. While non-naïve and naïve participants at the same D1 titer had the same predicted D15 titer following boost, non-naïve participants differ in multiple other ways from naïve participants. These include higher baseline and D15 neutralization titers (middle 90% D15 BA.1 titer ranging from about 1850 to 27,104 AU/ml for non-naïve participants compared to 451–22,834 AU/ml for naïve participants) and potentially also other factors not measured in this study. Possible reasons for the difference in the correlates between naïve and non-naïve participants include: (1) a stronger relationship between antibody and risk at higher antibody levels, and little relationship at lower levels for which the quantity of antibody was too low to generate or mark protection; (2) B cells, T cells, and other factors following natural infection have greater synergy with antibody at higher titers, compared to B and T cells from vaccination alone; and/or (3) boosted higher titer antibody following natural infection is qualitatively better even at the same neutralization titer. Memory B cells that are reactivated, e.g. via SARS-CoV-2 infection that occurs post-vaccination, are a source of anamnestic antibody responses[39,40]. In an exploratory analysis, we saw a significantly lower anamnestic response to SARS-CoV-2 in non-naïve participants compared to naïve participants after controlling for pre-COVID-19 neutralizing antibody levels. The factors responsible for clearing the virus faster and reducing the anamnestic response in non-naïve participants during COVID-19 are present at exposure and might have a similar role in preventing COVID-19, thus supporting hypotheses (2) and (3) above. Additional planned correlate analyses involving T-cell responses as well as systems serology, a computational approach that uses high-throughput data on non-neutralizing antibody functions and antibody biophysical properties to identify properties of humoral responses linked with risk[41], may help clarify the difference in protection between naïve and non-naïve participants and whether/how D15 titer predicts or impacts COVID-19 risk.

In addition to assessing antibody at D15, we correlated predicted neutralizing titer at each day of follow-up with COVID-19 risk for that

day. For both naïve and non-naïve participants, the predicted-at-exposure inverse correlates were stronger than for the D15 inverse correlates. These results suggest that the current level of antibody at exposure may be a better measure of protective ability over follow-up than peak antibody, which seems intuitive. Additional studies with longer follow-ups would help further test this hypothesis. Moreover, we compared antibody titer at D15 vs. fold-rise in titer from the second boost to D15 as correlates, and absolute titer – not fold-rise – was the clear correlate. This may have implications for defining success criteria for authorization/approval of boosters.

An important question is whether, in the context of a variant boost, matching the variant of the COVID-19 outcome with neutralization titer provides a stronger correlate relationship than an unmatched titer. To our knowledge, this question has not been addressed in previous analyses of individual-level data. We addressed this question by comparing the hazard ratios for an unmatched analysis (D614G titer correlated with BA.4/BA.5 Omicron COVID-19) with a matched analysis (BA.4/BA.5 Omicron titer correlated with BA.4/BA.5 Omicron COVID-19). Generally, the hazard ratios and covariate-adjusted cumulative incidence functions were quite similar for naïve participants and nearly identical for non-naïve participants, though with slightly better precision with variant matching. This result is not surprising given the high correlation between the different assays (Supplementary Figs. 4 and 5).

COVAIL was conceived to explore how variant-based vaccines might modify the post-boost immunological landscape. While our study had sufficient statistical power to look at correlates of risk, it was underpowered for correlates of protection (CoP) analyses that entail comparisons of COVID-19 incidence between arms, and we assessed groupings of study arms to check for large relative efficacy between groups and, where found, to assess CoPs of the relative efficacy. There was a large reduction in COVID-19 incidence for the Pfizer-BioNTech Omicron-containing versus Prototype vaccines allowing a CoP analysis (0.36 per 100 person-years vs. 0.92 per 100 person-years; p = 0.001). Controlled relative vaccine efficacy CoP analysis that pooled over naïve and non-naïve participants yielded relative efficacy estimates ranging from 51.7% at D15 BA.1 titer of 1000 AU/ml to 67.4% at 10,000 AU/ml and between 26% at 1000 AU/ml to 100% at 10,000 AU/ml depending on the method used, supporting a CoP although with limited precision. Assessment of D15 titers as mediators of the relative efficacy supported a modest fraction of the relative efficacy mediated through titer, 27% for D15 BA.1 titer.

While not randomized to the COVAIL stage, we observed a much lower COVID-19 incidence rate among non-naïve compared to naïve participants after adjustment for D15 antibody and baseline risk score. While this comparison may be biased due to residual confounding, one might expect the bias to understate the benefit of natural infection as non-naïve participants have demonstrated by their past infection that they had a greater likelihood of exposure. For seasonal coronaviruses, previous infection increases the risk of subsequent infection[42]. Further, in an exploratory analysis that adjusted for pre-COVID-19 titer, we observed a markedly reduced anamnestic response to COVID-19 in non-naïve participants suggesting that unmeasured factors are more quickly clearing the virus and likely have a similar beneficial effect in preventing COVID-19. A deeper examination of the immunological profile in naïve compared to non-naïve participants may identify specific factors responsible for the lower COVID-19 rate in non-naïve participants.

A strength of our study is that it is a prospective randomized trial studying a second booster dose during the BA.2/BA.4/BA.5 period with extensive characterization of neutralization titers over time against multiple variants for all participants. To our knowledge, this is the first randomized trial with an individual-level immune correlates analysis conducted of a booster variant dose. Moreover, the study was well powered with a large number of COVID-19 breakthrough cases and over 100-fold inter-vaccinee variation in antibody levels at D1 and at D15. Additionally, we studied mRNA vaccines that are widely used and thus most relevant to public policy.

A limitation of our study is that it only assessed neutralization titers as correlates. This limitation will be addressed in future work, including T-cell and systems serology analyses. A second limitation is that our study used a COVID-19 endpoint for some cases (59/213, 28%) that was defined as a self-reported positive viral test, differing from the endpoint definition in the phase 3 trials that required central lab virologic confirmation and meeting pre-specified symptoms criteria. However, the majority of COVID-19 endpoints (154/213, 72%) were virologically confirmed. Additional limitations are that it assessed titers against SARS-CoV-2 variants that are no longer circulating and that there is the potential for misclassification between naïve and non-naïve cases, given that anti-N antibodies are known to wane over time and may yield negative results even if the participant was previously infected. Additionally, some participants were categorized as non-naïve based solely on self-reported data. Finally, because this is an observational study and antibody level and naïve/non-naïve status were not randomly assigned, it is possible that confounding partly drives the difference in slopes between the naïve and non-naïve groups. To drive a difference in slopes, however, this confounding would need to be complex, where say testing and risk behavior depend on the underlying and unknown antibody level in naïve participants in a different way than for non-naïve participants.

The current vaccination policy for COVID-19 is moving toward booster dosing with updated vaccines matched to recently circulating variants. Licensure of variant-based vaccines using neutralization titer as a surrogate endpoint has been predicated on the strong and consistent relationship between neutralization titer and COVID-19 mostly from studies of primary immunization or first booster doses and with earlier variants. As SARS-CoV-2 develops into an endemic virus, population-level immunity reflects multiple bouts of vaccination and infection with successively different variants and the relationship between neutralization titer and COVID-19 risk may continue to change. We show this relationship continues within the non-naïve population, which supports the continued use of neutralization titer as a surrogate endpoint guiding approval and use. Moreover, the relevance of this finding increases as the naïve population continues to decrease in size. We also show that this relationship has changed within the naïve population underscoring the need for continued evaluation of the relationship between neutralization titer and variant COVID-19 risk.

## Methods
All analyses were pre-specified in the SAP, except as noted when results are presented and in Section 10 of the SAP.

### COVAIL trial
All participants in the COVAIL trial (NCT05289037) provided written informed consent before enrollment. A stipend was provided for participation in the study, which was determined by each enrolling site. The trial was reviewed and approved by the Advarra Central Institutional Review Board on 22 March 2022 and overseen by an independent Data and Safety Monitoring Board. The trial complied with all relevant ethical regulations. For further details on trial design, safety, and immunogenicity in Stages 1–3[18] and Stage 4[19], see Branche et al [18,19].

### Baseline risk score
A baseline risk score for best predicting the occurrence of COVID-19 starting 7 days after the D15 visit was built using cross-validated super learning pooling over all study participants, based on all demographic and vaccination history input variables, using a similar approach as taken for assessing immune correlates in the phase 3 trials through the

US Government's COVID-19 Vaccine Correlates of Protection Program[27,31,32,43–45]. The baseline risk score developed using the Superlearner[46,47] model (CV-AUC = 0.604) was adjusted for as a covariate in all correlates analyses and cumulative incidence analyses. See the SAP for further details.

## Force of infection score

To address the fact that the four stages enrolled participants in different calendar periods and that incidence of COVID-19 can have temporal trends, the peak correlates analyses also adjusted for an FOI score calculated with data from the Coronavirus Resource Center's database [previously hosted by Johns Hopkins University at https://coronavirus.jhu.edu/); that site is now inactive and the data can be accessed on Github at https://github.com/CSSEGISandData/COVID-19 and https://github.com/govex/COVID-19/tree/master/data_tables/vaccine_data. See the SAP for further details.

## D15 and fold-rise correlates of risk analyses

Univariable correlates of risk analyses were performed in the correlates analysis per-protocol cohort (Supplementary Fig. 2), pooling across the 13 one-dose mRNA arms (Supplementary Fig. 1) and in pre-specified subgroups determined by vaccine received. Analyses were conducted separately for baseline naïve and non-naïve participants. The analyses were based on Cox regression models adjusting for two covariates: the FOI score and baseline risk score and conducted separately for each D15 and fold-rise nAb-ID50 marker. Sensitivity analyses were also conducted to examine each component of the covariate adjustment strategy [(1) adjustment only for the FOI score and for the ≥65 age indicator; (2) adjustment only for the ≥65 age indicator; (3) no covariate adjustment]. The covariate-adjusted hazard ratios of COVID-19 per 10-fold increase in each marker were estimated and reported, along with 95% CIs and Wald-type, two-sided p-values. For further details, see the SAP.

Controlled risk curves were also reported in addition to HRs. A controlled risk curve describes the COVID-19 cumulative incidence had all participants received a specified vaccine under investigation and had their marker set to a specified level under consideration. Controlled risk curves were estimated using: 1. A semiparametric Cox-regression-based approach; and 2. A nonparametric approach developed in refs. 48, 49 and were implemented using the R package *vaccine*[50]. In addition to plotting the controlled risk curve by a full range of marker levels, we also estimated the controlled risk by D15 BA.1 ID50 tertiles using a Cox-regression-based approach. For further details, see the SAP. Code for conducting the controlled risk curve analyses is provided in Supplementary Software 1.

## Antibody decay model for supporting exposure-proximal correlates analysis

A linear mixed effects (LME) model was formulated for longitudinal measurements of log10 antibody titers and fit to measurements at visit days 14 (D15), 28 (D29), 90 (D91), and 180 (D181) post-vaccination. The model had fixed effects for days since D15, prior infection assessed by N-positivity at D1, and the interaction of prior infection and time since D15. The model has an individual-level random intercept. This model was used to calculate the antibody half-lives shown in Supplementary Table 10. Half-lives were calculated based on an LME model to facilitate comparison with previously reported results.

For the correlates analyses, the antibody decay model was piecewise linear with a bend at day 76, separate (fixed effects) terms for naïve and non-naïve participants, and a random intercept as in Eq. (1).

$$
\begin{aligned}
X_i(t) = \beta_0 &+ \beta_1(t - t_{i,d15}) + \beta_2(I(NposD1 = 1)) + \beta_3(t - t_{i,d15})I(NposD1 = 1) \\
&+ \beta_4(t - t_{i,d15} - 76)(I(t - t_{i,d15} \geq 76)) \\
&+ \beta_5(t - t_{i,d15} - 76)(I(t - t_{i,d15} \geq 76))I(NposD1 = 1) + b_i + \epsilon_{i,t}
\end{aligned}
\tag{1}
$$

where $t_{i,d15}$ is the calendar date of the D15 measurement for individual $i$, $X_i(t)$ is the time-varying log10 antibody titer on calendar day $t$ for individual $i$, $t - t_{i,d15}$ is the number of days since the D15 measurement for individual $i$ on calendar day $t$, $I(NposD1 = 1)$ is an indicator of being anti-n positive on D1, $(t - t_{i,d15} - 76)$ is the number of days since 76 days after the D15 measurement for individual $i$ on calendar day $t$, $I(t - t_{i,d15} \geq 76)$ is an indicator of whether calendar day $t$ is at least 76 days after the D15 measurement for individual $i$, $b_i \sim N(0, \tau^2)$ is an individual-level random intercept, and $\epsilon_{i,t} \sim N(0, \sigma^2)$. Values below the LOD are interval-censored below LOD, i.e. follow a tobit model. The model is fit to the D15, D29, D91, and D181 measurements.

Model fitting only used each participant's data prior to a possible breakthrough infection (regardless of symptoms) or out-of-study vaccination. A participant is determined to have had an asymptomatic infection between D22 and D91 if the following three conditions held: (1) anti-N seropositive at D29 or D91 visit, (2) anti-N seronegative at the D15 visit; (3) No COVID-19 endpoint during follow-up through 14 days post D29 visit (if first anti-N seropositive at Day 29) or through 14 days post D91 visit (if first anti-N seropositive at D91 visit). A participant is determined to have had an asymptomatic infection between D91 and D181 if the following three conditions held: (1) anti-N seropositive at the D181 visit, (2) anti-N seronegative at the D15, D29, and D91 visits, (2) No COVID-19 endpoint during follow-up through 14 days post D181 visit. Participants who were determined to have had an asymptomatic infection between the D22 and D91 visits only contributed their D15 measurement. Participants who were determined to have been asymptomatically infected between D91 and D181 only contributed measurements through the earlier of the date of D91 or the date of their symptomatic infection. This restriction was evaluated in simulations to assess the risk of bias prior to data analysis. The model was fit separately for each antibody marker.

## Exposure-proximal correlates of risk analysis

In an exposure-proximal correlates analysis, the predicted-at-exposure marker level was associated with the instantaneous risk of COVID-19 via a two-step procedure[26]. First, each participant's log10 antibody titer trajectory was predicted based on the antibody decay model. In the second step, a proportional hazards regression model was formulated. These analyses were conducted separately in naïve and non-naïve participants, and adjusted for the baseline risk score as a covariate. Because exposure-proximal analyses operated on the calendar-time scale, the external FOI was accounted for automatically and no FOI score adjustment was done. Bootstrap percentile CIs and p-values are reported. Code for conducting all exposure-proximal correlates of risk analyses is provided in Supplementary Software 1. See the SAP for further details.

## Controlled relative vaccine efficacy D15 correlates of protection analysis

Controlled relative vaccine efficacy curves describe the cumulative incidence of COVID-19 had all participants received a specified vaccine under investigation and had their D15 marker set to a specified level under consideration, compared to the cumulative incidence of COVID-19 had all participants received another specified vaccine. Controlled relative vaccine efficacy curves were estimated using: 1. A semiparametric Cox-regression-based approach; and 2. A nonparametric approach developed in refs. 48,49. Both approaches were applied to all participants and separately to baseline naïve participants and adjusted for the FOI score and the risk score (and naïve status for all participants analysis). The methods were applied to compare the Stage-2 Pfizer-BioNTech Omicron-containing vaccines to the Stage-2 Pfizer-BioNTech Prototype vaccine and were implemented using the R package *vaccine*[50]. Code for conducting the controlled relative vaccine efficacy analyses is provided in Supplementary Software 1.

### Mediation D15 marker correlates of protection analysis

Mediation analysis using the method previously used for the US Government COVID-19 Vaccine CoP Program Phase 3 trials[35] was applied to the Stage-2 Pfizer-BioNTech Omicron-containing vaccines to Stage-2 Pfizer-BioNTech Prototype vaccine. Point and 95% confidence interval estimates were generated for the natural indirect effect, i.e., marker-mediated VE (vaccine efficacy mediated through a given D15 marker), the natural direct effect, i.e., non-marker-mediated VE (vaccine efficacy through all other pathways not through the D15 marker), and the proportion of vaccine efficacy mediated through the D15 marker.

### Neutralizing antibody assay

A central lab (Monogram) measured neutralizing antibody titers using the PhenoSense SARS-CoV-2 Assay (Monogram Biosciences)[51]. This assay has been validated to CLIA/CAP standards and was also used for US Government COVID-19 Vaccine CoP Program Phase 3 trials[43–45].

In the assay, HEK 293 cells (Monogram Biosciences Master Cell Bank LC0027490) were co-transfected with a plasmid driving the expression of Spike-D614G and a lentiviral backbone plasmid (harboring a firefly luciferase reporter gene). Pseudovirus stocks were collected 2 days post-transfection, filtered, and stored <−70 °C.

At 24 h prior to assay day, HEK 293 cells were transiently transfected with plasmids driving the expression of the ACE2 receptor and of the TMPRSS2 protease. On assay day, the pseudovirus was incubated for one hour at 37 °C with the test samples. Serum samples were run in singlicate titrations using 10 serial three-fold dilutions. A suspension (10,000 cells per well) of the transiently transfected HEK 293 cells was then added to the serum-virus mixtures. Plates were incubated for 3 days at 37 °C in 7% $CO_2$, after which Steady Glo reagent (Promega) was added to each well. Luminescence, measured in relative light units, is directly proportional to virus inoculum infectivity and was measured using a Luminoskan luminometer.

Neutralization titers express the inhibitory dilution of serum samples at which relative light units were reduced by 50% (ID50) compared to virus control (no serum) wells. Data analysis (inhibition curve fitting and ID50 determinations) was done using Monogram proprietary analysis software. ID50 readouts are in arbitrary units/ml (AU/ml). For the D614G strain, multiplying AU/ml by 0.0653 calibrates these titers to those in the 20/136 WHO International Standard for anti-SARS-CoV-2 immunoglobulin (described in the Supplementary Materials of Gilbert et al.[27]), enabling the titers to be expressed in International Units (IU50/ml). Variant-specific neutralizing antibodies in the 20/136 International Standard, however, have not been measured in the PhenoSense SARS-CoV-2 Assay, and thus only D614G titers can be expressed in IU50/ml. For this reason, we always report AU/ml in the present work. The limit of detection (LOD) of the Monogram assay was 40 AU/ml (the same for all lineages of Spike-pseudotyped virus: D614G, Beta, Delta, BA.1, and BA.4/BA.5); all values below the LOD were assigned value ID50 = 20 AU/ml.

### Reporting summary

Further information on research design is available in the Nature Portfolio Reporting Summary linked to this article.

## Data availability

The trial dataset will be available to appropriate academic parties on request from the corresponding author, in accordance with the data sharing policies of the Division of Microbiology and Infectious Diseases of NIAID with input from the investigator group subject to submission of a suitable study protocol and analysis plan. Requests should be directed to the corresponding author, Dr. Dean Follmann (dfollmann@niaid.nih.gov), and will be responded to within a month. Source data are provided with this paper.

## Code availability

All analyses were done reproducibly on the basis of publicly available R scripts hosted on the GitHub collaborative programming platform (https://github.com/CoVPN/correlates_reporting2)[52] as well as in Supplementary Software 1. The analyses described in the correlates_reporting2 module (Cox proportional hazards modeling of risk) were conducted in R[53] (version 4.0.4) and used over 100 R packages (https://github.com/CoVPN/correlates_reporting2/blob/master/renv.lock). The analyses described in the "README.txt" file (cumulative incidence analysis, controlled risk curve analysis, and controlled relative vaccine efficacy analysis) were conducted in R[53] (version 4.2.1) and used the R packages vaccine (version 1.2.1)[54] and CFsurvival (version 0.1.0)[55]. The analyses described in the "README.md" file (all exposure-proximal correlates of risk analyses) were conducted in R[53] (version 4.3.1) and used nearly 100 R packages, listed in the README.md file.

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

## Acknowledgements

We thank the participants and site staff of the COVAIL trial and Lindsay Carpp for scientific writing and technical editing. This work was funded by the National Institute of Allergy and Infectious Diseases (NIAID) of the National Institutes of Health (NIH) "COVAIL Correlates of Protection Project" under Leidos Biomedical Inc. contract 75N910D00024, task order 75N91022F00007 and by NIAID award R37AI054165 (PBG). The COVAIL trial was funded in part by federal funds from NIAID and the National Cancer Institute, NIH, under contract 75N910D00024, task order no. 75N91022F00007, and in part by the Administration for Strategic Preparedness and Response, Biomedical Advanced Research and Development Authority, under Government Contract no. 75A50122C00008 with Monogram Biosciences, LabCorp. The contracts and federal funding are

not an endorsement of the study results, product or company. This work was also supported in part with federal funds from the NIAID, NIH, under contract no. 75N93021C00012, and by the Infectious Diseases Clinical Research Consortium (IDCRC) through the NIAID under award no. UM1AI148684. Testing of neutralizing antibody titers by Monogram Biosciences, LabCorp has been funded in part with federal funds from the Department of Health and Human Services, Administration for Strategic Preparedness and Response, Biomedical Advanced Research and Development Authority, under contract no. 75A50122C00008. Testing for anti-N-specific antibody was conducted by Cerba Research under contract no. 75N93021D00021. D.J.S. and A.N. are supported by the NIH–NIAID Centers of Excellence for Influenza Research and Response (CEIRR) contract 75N93021C00014 as part of the SAVE program (D.J.S.). A.N. was supported by the Gates Cambridge Trust. The content of this paper is solely the responsibility of the authors and does not necessarily represent the official views of the NIH–NIAID. NIH authors contributed to the work as described below and NIH reviewed the draft and approved its submission.

## Author contributions

Conceptualization: B.Z., Y.F., L.D., P.C.R., P.B.G., D.F.; Methodology: B.Z., Y.F., L.D., J.F., B.B., S.B., D.B., P.B.G., D.F.; Software: B.Z., Y.F., L.D., J.F., S.C., J.W., C.Y., C.A.M., C.M., B.B., S.B., D.B., J.M., M.M., P.B.G., D.F.; Validation: B.Z., Y.F., L.D., J.M., M.M., P.B.G., D.F.; Formal analysis: B.Z., Y.F., L.D., J.F., S.C., J.W., C.Y., C.A.M., C.M., B.B., S.B., D.B., A.N., D.J.S., J.M., M.M., P.B.G., D.F.; Investigation: N.G.R., A.R.B., D.J.D., A.R.F., D.S.G., L.R.B., S.E.F., J.A.W., S.J.L., S.K., E.B.W., R.M.N., R.R., L.A.J., T.M.B., A.C.K., A.F.L., L.C.I., R.M.P., M.B., P.L.W., S.M.M., P.A.G., D.N.F., R.L.A., C.M.P.; Resources: N.G.R., A.R.B., D.J.D., A.R.F., D.S.G., L.R.B., S.E.G., J.A.W., S.J.L., S.K., E.B.W., R.M.N., R.R., L.A.J., T.M.B., A.C.K., A.F.L., L.C.I., R.M.P., M.B., P.L.W., S.M.M., P.A.G., D.N.F., R.L.A., C.M.P., M.K.M., S.U.N., P.C.R.; Data curation: B.Z., Y.F., L.D., J.F., S.C., J.W., N.G.R., C.Y., C.A.M., C.M., B.B., S.B., D.B., A.N., D.J.S., J.M., M.M., P.B.G., D.F.; Writing – original draft: B.Z., Y.F., P.B.G., D.F.; Writing – review and editing: all authors; Visualization: B.Z., Y.F., L.D., J.F., S.C., J.W., C.Y., C.A.M., C.M., B.B., S.B., D.B., P.B.G., D.F.; Supervision: P.C.R., P.B.G., D.F.; Project administration: M.K.M., S.U.N., P.C.R., P.B.G., D.F.; Funding acquisition: D.J.S., P.B.G.

## Funding

## Competing interests

N.G.R. is a paid safety consultant for ICON, CyanVac, Imunon, and EMMES, and has served on selected advisory boards for Sanofi, Seqirus, Pfizer-BioNTech and Moderna. Emory receives funds for N.G.R. to conduct research from Sanofi, Lilly, Merck, Quidel, Immorna, Vaccine Company, and Pfizer-BioNTech. S.K. reports research support to his institution from Pfizer-BioNTech, Moderna, Bavarian Nordic, Meissa, Centers for Disease Control and Prevention, and National Institutes of Health. E.B.W. has served as an investigator for clinical trials or studies sponsored by Pfizer-BioNTech, Moderna, Seqirus, Najit, and Clinetic; on advisory boards for Pfizer-BioNTech and Vaxcyte; as a consultant to IliAD Biotechnologies; and as a DSMB member for Shionogi. T.M.B. has served on an Advisory Board for Sanofi. D.N.F. has served on an Advisory Board for Gilead Sciences and for AXCELLA, and as site PI for clinical trials sponsored by Gilead Sciences, Regeneron, MetroBiotech LLC, and the NIH (DMID COVAIL). The remaining authors declare no competing interests.

## Additional information

Bo Zhang [1], Youyi Fong [1,2,3], Lauren Dang [4], Jonathan Fintzi [4], Shiyu Chen [1], Jing Wang [5], Nadine G. Rouphael [6], Angela R. Branche [7], David J. Diemert [8], Ann R. Falsey [7], Daniel S. Graciaa [6], Lindsey R. Baden [9], Sharon E. Frey [10], Jennifer A. Whitaker [11], Susan J. Little [12], Satoshi Kamidani [13,14], Emmanuel B. Walter [15], Richard M. Novak [16], Richard Rupp [17], Lisa A. Jackson [18], Chenchen Yu [1], Craig A. Magaret [1], Cindy Molitor [1], Bhavesh Borate [1], Sydney Busch [19], David Benkeser [19], Antonia Netzl [20], Derek J. Smith [20], Tara M. Babu [21], Angelica C. Kottkamp [22], Anne F. Luetkemeyer [23], Lilly C. Immergluck [24,35], Rachel M. Presti [25], Martín Bäcker [26], Patricia L. Winokur [27], Siham M. Mahgoub [28], Paul A. Goepfert [29], Dahlene N. Fusco [30], Robert L. Atmar [11], Christine M. Posavad [31,32], Jinjian Mu [33], Mat Makowski [33], Mamodikoe K. Makhene [34], Seema U. Nayak [34], Paul C. Roberts [34], Peter B. Gilbert [1,2,3], Dean Follmann [4] ✉, Coronavirus Variant Immunologic Landscape Trial (COVAIL) Study Team

[1]Vaccine and Infectious Disease Division, Fred Hutchinson Cancer Center, Seattle, WA, USA. [2]Public Health Sciences Division, Fred Hutchinson Cancer Center, Seattle, WA, USA. [3]Department of Biostatistics, School of Public Health, University of Washington, Seattle, WA, USA. [4]Biostatistics Research Branch, National

Institute of Allergy and Infectious Diseases, National Institutes of Health, Bethesda, MD, USA. [5]Clinical Monitoring Research Program Directorate, Frederick National Laboratory for Cancer Research, Frederick, MD, USA. [6]Hope Clinic, Emory University, Decatur, GA, USA. [7]Vaccine and Treatment Evaluation Unit, University of Rochester, Rochester, NY, USA. [8]George Washington Vaccine Research Unit, George Washington University, Washington, DC, USA. [9]Department of Medicine, Brigham and Women's Hospital, Harvard Medical School, Boston, MA, USA. [10]Center for Vaccine Development, Saint Louis University, St Louis, MO, USA. [11]Department of Molecular Virology and Microbiology and Department of Medicine, Baylor College of Medicine, Houston, TX, USA. [12]Division of Infectious Diseases and Global Public Health, Department of Medicine, University of California, San Diego, La Jolla, CA, USA. [13]Center for Childhood Infections and Vaccines, Children's Healthcare of Atlanta, Atlanta, GA, USA. [14]Department of Pediatrics, Emory University, Atlanta, GA, USA. [15]Duke Human Vaccine Institute, Duke University School of Medicine, Durham, NC, USA. [16]Project WISH, University of Illinois at Chicago, Chicago, IL, USA. [17]Department of Pediatrics, University of Texas Medical Branch, Galveston, TX, USA. [18]Kaiser Permanente Washington Health Research Institute, Seattle, WA, USA. [19]Department of Biostatistics and Bioinformatics, Rollins School of Public Health, Emory University, Atlanta, GA, USA. [20]Center for Pathogen Evolution, Department of Zoology, University of Cambridge, Cambridge, UK. [21]Division of Allergy and Infectious Diseases, Department of Medicine, University of Washington, Seattle, WA, USA. [22]Vaccine and Treatment Evaluation Unit, Manhattan Research Clinic, New York University Grossman School of Medicine, New York, NY, USA. [23]Division of HIV, Infectious Diseases and Global Medicine, Zuckerberg San Francisco General Hospital, University of California, San Francisco, CA, USA. [24]Clinical Research Center, Department of Microbiology, Biochemistry, and Immunology, Morehouse School of Medicine, Atlanta, GA, USA. [25]Department of Medicine, Washington University School of Medicine, St Louis, MO, USA. [26]Department of Internal Medicine, University of Utah Schoole of Medicine, Salt Lake City, Utah, USA. [27]Department of Medicine, University of Iowa College of Medicine, Iowa City, IA, USA. [28]Howard University College of Medicine, Howard University Hospital, Washington, DC, USA. [29]Department of Medicine, University of Alabama at Birmingham, Birmingham, AL, USA. [30]Department of Medicine, Tulane University School of Medicine, New Orleans, LA, USA. [31]Infectious Diseases Clinical Research Consortium (IDCRC) Laboratory Operations Unit, Fred Hutchinson Cancer Center, Seattle, WA, USA. [32]Department of Laboratory Medicine and Pathology, University of Washington, Seattle, WA, USA. [33]The Emmes Company LLC, Rockville, MD, USA. [34]Division of Microbiology and Infectious Diseases, National Institute of Allergy and Infectious Diseases, National Institutes of Health, Bethesda, MD, USA. [35]Present address: Biological Sciences Division, The University of Chicago, Chicago, IL, USA.
✉e-mail: dfollmann@niaid.nih.gov

## Coronavirus Variant Immunologic Landscape Trial (COVAIL) Study Team

Nadine G. Rouphael [6], Angela R. Branche [7], David J. Diemert [8], Ann R. Falsey[7], Daniel S. Graciaa [6], Lindsey R. Baden [9], Sharon E. Frey[10], Jennifer A. Whitaker[11], Susan J. Little [12], Satoshi Kamidani[13,14], Emmanuel B. Walter[15], Richard M. Novak[16], Richard Rupp[17], Lisa A. Jackson[18], Antonia Netzl [20], Derek J. Smith [20], Tara M. Babu[21], Angelica C. Kottkamp [22], Anne F. Luetkemeyer [23], Lilly C. Immergluck[24,35], Rachel M. Presti[25], Martín Bäcker[26], Patricia L. Winokur[27], Siham M. Mahgoub[28], Paul A. Goepfert [29], Dahlene N. Fusco [30], Robert L. Atmar [11], Christine M. Posavad[31,32], Jinjian Mu[33], Mat Makowski[33], Mamodikoe K. Makhene[34], Seema U. Nayak[34] & Paul C. Roberts [34]

