## [Peer Review file · Nature Communications]

Neutralizing antibody immune correlates in COVAIL trial recipients of an mRNA second COVID-19 vaccine boost

Corresponding Author: Professor Dean Follmann

Version 0:

Reviewer comments:

Reviewer #1

(Remarks to the Author)

Zhang et al. present an interesting and important analysis of COVID-19 breakthrough infections and correlates of risk from an RCT of second booster dose, randomised to receive different variant updated vaccines (that is, a 4th dose for those who had a two dose primary regimen).

The study finds an important, albeit surprising, observation that the neutralising antibody titres are more strongly correlated with protection in previously infected individuals, compared with those who have not had previous infection. The study is thorough and well conducted, including well-thought through sensitivity analyses to stress-test their result.

However, there remains some critical issues with the presentation of the work and discussion of the results. In particular, given that one of the primary results contradicts existing evidence, in that the study shows only a weak correlation between neutralisation titre and risk for naïve individuals (the same authors have shown a stronger correlation in numerous studies previously) – it seems important to consider, more comprehensively, the potential confounding that might be driving this observation. In addition to this additional discussion of confounding, a dominant concern for this reviewer is the massive complexity of the correlates analysis. It raises questions about whether the analysis choices may be unintentionally driving (confounding) their conclusions.

Main concerns:

1. Likely sources of confounding in the study design may be responsible for the conclusions, but are not addressed or discussed.

The authors find that neutralisation correlates with protection in previously infected people, but less so in naïve people (people without previous infection). This is very surprising since existing correlates studies have shown that it is in naïve vaccinated individuals that neutralisation correlates with protection. It seems to this reviewer that this result may entirely arise due to confounding that have not been considered in the analysis and are not discussed as caveats in the discussion. Specifically, (a) the infection outcome being self-reported and potentially ill-defined, (b) individual testing-behavioural differences based on previous infection status (propensity to get tested may differ), and (c) demographic differences in the two groups potentially indicating differences not only in risk of infection, but vaccination history (eg. Intervals) and testing behavioural differences.

Specifically:

a. Study infection outcome (not primary outcome) is vaguely defined and depends on behaviour.

The outcome of SARS-CoV-2 infection in this study is based on self-reporting and doesn't have any requirement for symptoms. Thus, the outcome of the study is highly variable and may be asymptomatic or symptomatic infection depending on the differences in behaviour of people in the study. If people have a low threshold for testing or test routinely for vocational purposes, then an infection outcome is more likely and more likely to be asymptomatic than for those who only test when symptoms are severe. This imprecise outcome likely also interacts other potential sources of confounding below.

b. Behavioural factors are likely confounders between individuals with and without previous infection.

Both people's status as 'previously infected' and their COVID-19 infection outcome in this study are based on self-reported

positive test results (at least in part). However, people's testing behaviours are almost certainly not random with respect to status of previous infection. E.g. It is possible:

- i) People change their testing behaviour if they have had COVID-19 (and are less likely to test for COVID-19 – or the symptom threshold at which people seek testing is changed).
- ii) People who are still naïve for COVID-19 in March–Oct 2022 in the USA are not a typical cohort. This is evidenced by 64% of the enrolled people were naïve for COVID-19 at a time when seroprevalence in the US was >50-60%#. This may be a cohort that is more likely to test at a lower symptom threshold.

Jones JM, Manrique IM, Stone MS, et al. Estimates of SARS-CoV-2 Seroprevalence and Incidence of Primary SARS-CoV-2 Infections Among Blood Donors, by COVID-19 Vaccination Status — United States, April 2021–September 2022. *MMWR Morb Mortal Wkly Rep* 2023;72:601–605. DOI: <http://dx.doi.org/10.15585/mmwr.mm7222a3>.

c. Demonstratable demographic differences in the two groups that could cause confounding.

The two groups in the study (previously infected, not previously infected) are demographically distinct (different ages/ethnicities). This supports the notions above that there is potential behavioural differences in these cohorts, especially around testing behaviour and thresholds for testing. Older people were less likely to have been infected by 2022 in the pandemic than younger people (according to seroprevalence data), but are also more likely to engage with health care interventions and testing. They were likely to have more frequent booster vaccines, and also to test more frequently for COVID-19. The authors attempt to account for demographic differences in the two populations, but the authors have not discussed the limitations of these adjustments. They have also not discussed whether these can account for both risk of infection differences as well as testing propensity differences (almost certainly not because there is an implicit confounding on these two aspects).

Together these potential sources of confounding in the study could drive the primary observed conclusion of the study and explain the contradictory result of this study with the authors previous studies. Therefore, they must be (ideally) considered in the analysis, or these limitations raised in the discussion as potential reasons for the conclusions of the study.

2. A high degree of complexity in analysis and low transparency (black box pipeline).

The authors approach to analysing a correlate of risk here involved numerous complex steps in a pipeline, including:

- risk adjustments based on a machine learning approach
- risk adjustment based on calendar time using a database
- (in the case of proximal correlates) antibody decay modelling with a mixed effects model
- cox model of risk adjusted data, sometimes with and sometimes without risk-adjustment.

These methods were all pre-specified and there is no risk of cherry-picking analysis tools to get a particular result, which is a strength of the work. None-the-less it is not possible in its current form to really understand whether any of the steps in the pipeline (especially around risk adjustments and covariates in the modelling) are correctly eliminating (or introducing) confounding. All methodologies have assumptions and limitations. I am not across the details of the risk adjustment methodology used by the authors, but it will certainly have assumptions and limitations. Could these be introducing unintentional confounding in the analysis and/or driving the apparent contradictory result observed here of a weak correlation between neutralisation and risk in naïve individuals? In addition, the authors should demonstrate how each element of the analysis pipeline is influencing the results. If the super learner risk adjustments are not performed and a simple cox-regression analysis is performed (perhaps with or without covariates) how are the correlates of risk altered? The authors can achieve this by re-running the analysis pipeline with and without various aspects of the pipeline.

Also, without a naïve control population in this RCT it is hard to see how the risk adjustments are performed with the super learner? In the authors previous work my understanding is that these risk adjustments were informed by the naïve placebo control groups? How is risk adjustment made with only intervention arms?

The authors must show the impact of analysis choices on the outcome of the analysis (by reducing model complexity and showing how results change). Further, the authors discuss some limitations around the data but do not discuss the potential assumptions and limitations of the methods adopted.

3. Data not available.

Finally, because of the model complexity it is absolutely critical that the full underlying data be made available in this study for readers to critically assess the correlates analysis. The original study does not make the Full data available either. Of note: There are no ethical concerns with releasing the data, because all of this data can be de-identified. E.g. If the authors felt age-buckets instead of exact age would ensure no identifiability concerns that would also be sufficient. As it stands, there is no way to assess the analytic pipeline or reproduce (or critique) the result.

Other issues:

Reporting:

1. The authors say previous infection was defined as either self-reported or anti-N positive. What proportion of previous

infection was characterised by either of these criteria?

2. The authors report p-values for interaction analysis, but not effect size (e.g. lines 282-283). It would be good to see effect sizes in the sup. table S5.
3. Covariates adjusted for is not clear. It is very hard to work out what covariates are adjusted for in covariate adjusted analysis. Was it the same set of covariates in all analyses. Did this include previous infection status? If so, this seems like it may unintentionally introduce a confounder (via collider?) in the regression? Could the authors spell out more clearly what covariates are included? It is apparently in the SAP, but I could not easily find it. On page 59 I see a list of "relevant participant co-variables" but this includes previous infection status.

Ab Decay modelling:

4. It is not possible to assess how well the decay model is fitting the data since the data and fitted model is nowhere clearly presented. There is a random set of 10 individuals given in figure S21, but this is not adequate.
5. It is important to understand over time how many neutralisation titres dropped below the detection limit, and whether the decay generally looks exponential over the time period being modelled (or is it biphasic etc? – and this might differ by group)?
6. It is unclear whether the authors account for (or need to account for) values below the limit of detection? The supplementary methods say they will do this (with imputation) if more than 10% are below the limit of detection? But it doesn't say if they had to do this or not? Also, what if the 10% that are below LOD are all at later time points or in one group? Could this not introduce a bias? This seems important since it seems more likely that the low titres in individuals without previous infection would be more likely to fall below the LOD over time?
7. The authors do not consider a random effect on the slope between individuals. It seems possible/likely there is random variation in the decay of neutralising antibodies between individuals.

Proximal Correlates:

8. If I am understanding correctly, the uncertainty in the predicted titres for each individual do not seem to be taken into account in the correlates analysis. i.e. there is clearly significant uncertainty in the predicted Ab titre for each individual, and this uncertainty doesn't seem to be taken into account – rather it is used as though it is an errorless estimate? Consider including this uncertainty in the correlates analysis?
9. In the BA.5 analysis it is unclear that the authors do not now have more values below LOD, especially over time, and so the issues with proximal correlates model and decay described above are perhaps more of a concern here. Can the authors show the BA.5 titres over time, and the model fit to this data?

Discussion:

10. The reasons the authors provide for the surprising conclusion of a different association between neutralisation and risk between groups seems to contradict the data here and in previous studies.

The authors say that the different correlation for naïve and non-naïve may be the result of:

- “(1) a stronger relationship between antibody and risk at higher antibody levels, and little relationship at lower levels for which the quantity of antibody was too low to generate or mark protection;
- (2) B cells, T cells, and other factors following natural infection have greater synergy with antibody at higher titers, compared to B and T cells from vaccination alone; and/or
- (3) boosted higher titer antibody following natural infection is qualitatively better even at the same neutralization titer.”
- But, all of these explanations assume that the difference between the correlation in naïve and non-naïve populations (i.e. steeper for non-naïve), is because the association gets stronger in non-naïve people. But the authors themselves note that the association for naïve people is weaker than previously observed. So it seems like it's the naïve cohort in this study that is the outlier here? At similar titres the authors have previously shown that naïve people have a stronger correlation of neutralisation and protection than observed here (e.g. correlates in JnJ vaccinees, Fong et al., Nat Micro 2022).

The authors must spell out more clearly in the discussion that it seems the reason for their primary conclusion is a change in the relationship between neutralisation and protection in naïve people compared to previous observations. This in itself may point to either something being different about a 4th booster dose, though this seems difficult to interpret, or something different about this RCT in itself. The latter seems more likely to this reviewer – though the former cannot be excluded.

(Remarks on code availability)

Reviewer #2

(Remarks to the Author)

In the manuscript “Neutralizing antibody immune correlates by prior SARS-CoV-2 infection status in COVAIL trial recipients of an mRNA second COVID-19 vaccine boost” by Bo Zhang et al., the authors present an analysis of correlates of risks and correlates of protection for COVID-19 infection in the context of the COVAIL trial. The trial included 985 participants, who received COVID-19 vaccination boosters (meaning they had been vaccinated previously) that were either monovalent or bivalent vaccines. The authors measured neutralizing antibody (nAb) titers against five different strains of COVID-19 up to 6 months after administration of the booster vaccine. They observed that higher nAb titers reduced the risk of infection with the SARS-CoV-2 Omicron strain significantly and found that the risk reduction was stronger in individuals previously infected with SARS-CoV-2.

While the study is in principle interesting, the description of the results should be improved. For example, in the abstract the authors mention that they assess “titers [...] as correlates of COVID-19” where they could have meant to say “COVID-19 disease outcome” or “COVID-19 infection risk”. The same can be said for the introduction where they use a similar wording. Furthermore, in the introduction they use a couple of terms, such as “exposure-proximal” or “COVID-19 endpoint”, which are only explained in the results section. This makes a lot of the text incomprehensible at first. While these might be minor language issues, the frequency at which this occurred was striking.

Apart from the language and structural issues of the text, the description of the statistical methods is in my opinion insufficient. The major issue is the insufficient description of the baseline risk of infection. The authors mention the use of a “superlearner”, which is described in more detail in Section 6.3 of the SAP. The superlearner learns the baseline risk of infection, which is used in adjusting their “correlates analyses”. This superlearner adjusts for many different “baseline covariates” (such as age and sex) and the text gives the impression that the output of the superlearner is equal to in their description of the Cox-model. However, it is unclear whether it is, because it is not individual specific and does not depend on the covariates in the equation. It is also unclear where the force of infection score that they calculate is included in the model. Additionally, the Cox-model adjusts for the same covariates again in, which would make the interpretability of the effect of the covariate more difficult.

In addition to this, it is in my opinion crucial that:

- 1) the authors make the datasets available (see first comment below),
- 2) reveal already in the instruction that the dataset has been studied before and has not been extended in this manuscript, and
- 3) discuss the what's novel insights have been obtained here compared to the previous studies.

Further major and minor issues are also listed below.

Major issues:

0. Data

The authors state that “All data were previously included with Branche et al.18,19”. I had a detailed look at these papers. While they say that “All data are included in the paper.”, I was unable to find it. Furthermore, the code refers to a very specific data file “covail_data_processed_20240313.csv”, which given that data seems to be newer than the two cited publications. In my opinion it is crucial to share that data to ensure reproducibility. If indeed data is used from a previous publication and I failed to find it, a direct link should be provided and the data from the original study should be used directly in the code. If this is not done, the data reformatting needs to be explained in details.

1. Results

- a. D15 (Peak) titer inverse correlates of risk of COVID-19 stronger for non-naïve than naïve participants
 - i. “Cox model interaction $P = 0.002-0.009$ for the six markers for boost-proximal cases”: It is unclear what is meant by “Cox model interaction”. According to the equations in Section 6.3 in the SAP, there are no interactions between two antibody titers in the sense that the Cox model includes. It is especially unclear, because in section 6.3 of the SAP, they do include an interaction term between naïve status and the antibody titer, but I do not think that it is used here. Furthermore, since the authors perform univariate Cox regression, the authors should be careful with expressions such as “associations [...] are generally stronger” or “all six D15 titer markers were inverse correlates of risk” without multiple-testing correction. The last minor issue is the usage of the capital “P” for reporting p-values, as the convention is to use a lower case “p”.
 - ii. “Post-hoc sensitivity analyses”: Missing or unclear description of the sensitivity analysis.
- b. The authors state that “A participant was determined to be SARS-CoV-2 non-naïve (hereafter, “non-naïve”) if they self-reported a previous SARS-CoV-2 infection or had detectable anti-N antibodies”. The latter seems to rely on a specification of a threshold. It would be beneficial if this threshold is reported.

2. Discussion

- a. “with clear conclusion of weaker inverse correlates than previously found”: No reasoning for this discrepancy is given.
- b. “non-naïve and naïve participants at the same D1 titer had the same predicted D15”: Prediction of D15 titers was not discussed in the text.

3. Methods

- a. Antibody decay model for supporting exposure-proximal correlates analysis
 - i. Clear description of the mixed-effects model is missing.
 - ii. The model does not seem to have any random effects, according to the authors’ description. This would be a clear limitation.
- b. About the Cox models:
 - i. Insufficient description of the use of the baseline risk of infection and force of infection scores. The supplement provides additional details, but it remains for instance unclear what’s meant by the statement: “the peak correlates analyses also adjusted for an FOI score”.
 - ii. Throughout the text it is unclear whether the Cox models were separately estimated for naïve and non-naïve participants or whether the Cox model simply included a covariate with the naïve status information.

Minor issues:

1. Abstract

- a. “the increasing number of vaccine boosters raise questions as to this surrogate’s contemporary performance”: Unclear how the number of vaccine boosters raise questions to the performance.

- b. “correlates of COVID-19”: Unclear what exactly they want to assess here.
- c. “exposure-proximal”: Meaning unclear
- d. “Omicron COVID-19 risk”: Unclear if risk of death, severe disease course or infection risk.

2. Introduction

- a. “In this work we studied how antibody level correlates with COVID-19”: See abstract issue
- b. Distinction between CoR and CoP could be made clearer. In the latter one is trying to find causality while the first is just a study of associations. A source as given by Gilbert et al. (Reference 23 in this manuscript) would be good.
- c. D614G, Ancestral, and Prototype variants: I assume all variants are the same, so one term is enough.
- d. “Inverse CoR”: A term that is more confusing than just “reduces risk of COVID-19 infection”.
- e. “D1 to D15 fold-rise nAb titer”: Odd wording. Suggestion: Fold-change of antibody levels between D1 and D15.
- f. “exposure-proximal”: Meaning unclear
- g. “COVID-19 endpoints”: Unclear what endpoints they mean at this point of the text.
- h. “this study affords substantial precision”: What does this mean?

3. Results:

- a. Some headings are sentences and are missing the verb.
- b. Trial schema and participant demographics
 - i. Figure S1 also shows trial Stage 3. It is not described here.
 - ii. “participants who developed COVID-19 prior to 7 days post D15 were excluded”: Unclear why they were excluded and confusing wording.
 - iii. No information about prior vaccination status reported here. Gives the impression that individuals did not receive any previous vaccinations.
- c. Available neutralizing antibody titer data
 - i. “BA.4/BA.5 titers at D15 were only measured from a random subset”: Why?
 - ii. “D15 BA.4/BA.5 titers for participants with missing values were imputed based on D29 BA.4/BA.5 ID50 and D15 BA.1 ID50”: Reference to part of methods where imputation is described is missing.
- d. Neutralizing antibody titers lower in COVID-19 cases than non-cases
 - i. “Naïve participants had lower D1 titers than non-naïve participants, with 26% vs. 3% undetectable”: Are these percentages medians of the study population? What is the meaning of “undetectable”?
 - ii. “are expected to be boosted 22-fold”: Is that their prediction or something they measured? Again, are these median numbers?
 - iii. “with geometric mean (GM) ID50 in non-cases, booster-proximal cases, and booster-distal cases 203, 155, 134 AU/ml”: Missing confidence intervals for the means. Same in the next sentence.
 - iv. Why use AU/ml? According to the methods, there was also some conversion to the WHO International Standard, which would result in IU/ml.
- e. D15 (Peak) titer inverse correlates of risk of COVID-19 stronger for non-naïve than naïve participants
 - i. “The associations between D15 titers with subsequent COVID-19 were generally stronger in non-naïve than naïve participants”: What associations? Is “correlations” meant?
 - ii. “intersection of the middle 90% of titer values”: Would this be 5% to 95% quantile? Why restrict the analysis to these values?
 - iii. “We further conducted a sensitivity analysis to explore the influence of a different covariate adjustment strategy on the results”: What is the “different covariate adjustment strategy”?
 - iv. “The HR point estimates were largely consistent across the subgroups, with that among Prototype insert boost recipients closest to unity [1.00 (0.56, 1.78); P = 0.999] and that among Monovalent Omicron insert boost recipients the smallest [0.57 (0.30, 1.07)]: Unclear what the authors mean by “given that larger fold-rises tend to occur among those with lower baseline antibody levels.”: This is not reported in any figure or supported by another reference.
- f. Exploratory Analysis of pre and post COVID-19 titer by previous infection status
 - i. The slopes should probably be weighted by the distance to the infection, because dynamics may vary strongly.
 - ii. “Figure S20A”: Figure S22A is probably meant.
 - iii. “Adjusting for pre COVID-19 titer being above or below the median in the non-naïve arm”: Why is this adjustment done?
- g. Lineage-specific correlates of risk: Similar results when restricting to BA.4/BA.5 COVID-19
 - i. “However, the lineage-matched analyses displayed marginally improved precision as the lineage-matched-based CIs were in general narrower than those obtained based on D614G titer.”: This is not a valid conclusion. It may just mean that the D614G titer does not have an effect on the risk of infection.

4. Discussion

- a. “weak inverse correlates with COVID-19”: See above.
- b. “with all 95% CI upper limits slightly higher than 1”: See above about multiple testing correction and a different formulation would just be that these markers had no significant effect on the risk of infection with SARS-CoV-2.
- c. “anamnesic response to COVID-19”: What is an anamnesic response?
- d. “The hazard ratios for both naïve and non-naïve participants were stronger than for the D15 correlates and estimated with more precision.”: See above.
- e. “These results suggest that pre-existing antibody at exposure may be a better measure of protective ability over follow-up than peak antibody”: I do not see how this follows from the previous sentence.
- f. “the study was well powered”: The study had sufficient statistical power?

5. Methods

- a. Methods are spread out across the main article, the supplementary methods section, and the SAP. For details the reader

should always only be referred to the SAP, as it contains the most detailed information.

b. "secular trends": What does secular mean in this context?

(Remarks on code availability)

I had a brief look at the code. It contains a README file and a few routines for basic analysis. In my opinion it would be very beneficial if the dataset could also be included.

Version 1:

Reviewer comments:

Reviewer #1

(Remarks to the Author)

The authors have for the most part addressed the concerns of this reviewer - and there responses are thoughtful and comprehensive.

However, I note that the authors have still not made the data publicly available. This was a concern raised by both reviewers.

The data sharing statement added by the authors declares that people are able to contact the author and request the data and only if its approved by an NIH department will the data be shared. This is not making the data available, and in my teams experience the vast majority of such requests are ignored or rejected.

The only way to ensure that data is actually available, and able to be used to ensure reproducibility, is for the data to be made publicly available as a supplement or in a repository.

I note of course that this is a more broad issue in the field than being particular to this manuscript or author group.

(Remarks on code availability)

Reviewer #2

(Remarks to the Author)

We would like to thank the authors for addressing every comment with such attention to detail. The authors have answered most of the comments clearly and resolved most issues of the original manuscript. However, there were a couple of responses that warrant further discussion:

1) Response #2 – "The risk score is a function of baseline covariates and takes on a different value [...]": To begin with, it seems that parts of my comments have been lost in the response letter. These parts mostly appear to be equations. There may have been some misunderstanding due to this. To clarify the situation, I will try to explain my issue with the superlearner and the Cox model again. It was unclear to me where the superlearner enters the Cox model. As it was called a baseline risk, it could have entered the model as λ_0 . Then X_i would have been covariates such as age and sex, as they X_i was called "baseline covariates". In this case, the covariates would have been included in the model twice (once in the superlearner and once in X_i). However, after carefully reading Section 6 of the SAP again, it has become clear that the output of the superlearner enters the model as a "baseline covariate" in $X_i * \beta$. In my opinion, this connection is not very clear from the main text or the beginning of the supplementary methods and should be made clearer earlier.

Apart from this issue, the authors have made a good effort to evaluate their results by comparing different model formulations and it is interesting to see that the results remain very similar. This obviously raises the question whether it is necessary to adjust for the covariates using the superlearner.

3) Response #5 – "We have interrogated the interaction test more thoroughly [...]": The interaction part has become clearer, but it is unclear what the authors mean with a hazard ratio ratio ("HR Ratio"). I believe, what the authors mean to show is just the hazard ratio (per one standard deviation increase of antibody titer), as it is described by the proportional hazards model. This is not what was written though and should be clarified

4) Response #9 – "We have added the following (lines 524-530)[...]": Simply a hint that the revised manuscript says "Prototype vaccines" but the response text says "ancestral strain vaccines"

5) Response #13 – "Please see the response to reviewer #1, "Other issue" #3.": Is related to response #2 and my comment to this response.

6) Response #46 – "We have softened the language, please see above.": Since the 95% CI of the hazard ratio includes 1, it does suggest that there is no relationship between titer and risk of COVID-19. I would be very careful about interpreting things which are not significant or only barely significant. It is well known that effect size estimates are in these cases biased and mostly overestimated.

As soon as the above mentioned issues have been resolved, we can recommend the current work for publication.

(Remarks on code availability)

REVIEWER COMMENTS

Reviewer #1 (Remarks to the Author):

Zhang et al. present an interesting and important analysis of COVID-19 breakthrough infections and correlates of risk from an RCT of second booster dose, randomised to receive different variant updated vaccines (that is, a 4th dose for those who had a two dose primary regimen).

The study finds an important, albeit surprising, observation that the neutralising antibody titres are more strongly correlated with protection in previously infected individuals, compared with those who have not had previous infection. The study is thorough and well conducted, including well-thought through sensitivity analyses to stress-test their result.

However, there remains some critical issues with the presentation of the work and discussion of the results. In particular, given that one of the primary results contradicts existing evidence, in that the study shows only a weak correlation between neutralisation titre and risk for naïve individuals (the same authors have shown a stronger correlation in numerous studies previously) – it seems important to consider, more comprehensively, the potential confounding that might be driving this observation. In addition to this additional discussion of confounding, a dominant concern for this reviewer is the massive complexity of the correlates analysis. It raises questions about whether the analysis choices may be unintentionally driving (confounding) their conclusions.

Main concerns:

1. Likely sources of confounding in the study design may be responsible for the conclusions, but are not addressed or discussed.

The authors find that neutralisation correlates with protection in previously infected people, but less so in naïve people (people without previous infection). This is very surprising since existing correlates studies have shown that it is in naïve vaccinated individuals that neutralisation correlates with protection. It seems to this reviewer that this result may entirely arise due to confounding that have not been considered in the analysis and are not discussed as caveats in the discussion. Specifically, (a) the infection outcome being self-reported and potentially ill-defined, (b) individual testing-behavioural differences based on previous infection status (propensity to get tested may differ), and (c) demographic differences in the two groups potentially indicating differences not only in risk of infection, but vaccination history (eg. Intervals) and testing behavioural differences.

Specifically:

a. Study infection outcome (not primary outcome) is vaguely defined and depends on behaviour. The outcome of SARS-CoV-2 infection in this study is based on self-reporting and doesn't have any requirement for symptoms. Thus, the outcome of the study is highly variable and may be asymptomatic or symptomatic infection depending on the differences in behaviour of people in the study. If people have a low threshold for testing or test routinely for vocational purposes, then an infection outcome is more likely and more likely to be asymptomatic than for those who only test when symptoms are severe. This imprecise outcome likely also interacts other potential sources of confounding below.

Response: Thank you for the opportunity to clarify. The study outcome of interest was symptomatic SARS-CoV-2 infection, which is a subset of self-reported/study conducted infections. We have confirmed that 98% (208/213) endpoints in our analysis met the CDC clinical criteria and supportive laboratory criteria, and have added the following sentence to the “COVID-19 endpoint definition, case and non-case definitions” (lines 224-225):

“Almost all (208/213) COVID-19 endpoints met the CDC clinical criteria and supportive laboratory criteria for a COVID-19 surveillance case¹ (Supplementary Table 4).”

We also now present detailed information on symptoms in the revision (lines 246-249):

“...Supplementary Table 6 provides a breakdown of the numbers of participants, by participant naïve/non-naïve status, with each of 22 different symptoms. There was no statistical difference in the distribution of symptoms between naïve and non-naïve participants (chi-squared $p = 0.12$).”

Moreover, to enable comparison to the results of the immune correlates analysis of a third dose of mRNA-1273 in the COVE trial,² we defined a second endpoint (“COVE COVID-19”) based on the clinical criteria for the COVID-19 endpoint used in the COVE primary efficacy analyses^{3,4} as well as previous COVE immune correlates analyses.^{2,5-8} Of the 213 COVID-19 endpoints included in the present analysis, 191 (89.7%) met the COVE clinical criteria (listed below) and supportive laboratory criteria for a COVE endpoint [in naïve participants: 162/181 (89.5%); in non-naïve participants: 29/32 (90.6%)].

Definition of the COVE COVID-19 endpoint:

- Self-reported positive SARS-CoV-2 test (RT-PCR or antigen test) OR study-conducted positive SARS-CoV-2 test (nasal swab and subsequent nucleic acid amplification test at an unscheduled illness visit) AND
- At least TWO of the following systemic symptoms: fever ($\geq 38^{\circ}\text{C}$), chills, myalgia, headache, sore throat, new loss of taste or smell, OR
- At least ONE of the following respiratory signs/symptoms: cough, shortness of breath or difficulty breathing, OR clinical or radiographical evidence of pneumonia.

Bullets (2) and (3) above are the clinical criteria used for a COVID-19 case definition in the primary COVE efficacy analyses.^{3,4}

We repeated the D15 and exposure-proximal correlates analyses using the COVE endpoint and found that the results (Supplementary Figs. 24 and 28, respectively) were virtually the same as for the COVID-19 endpoint (see lines 366-373 and 405-408 in the revision).

b. Behavioural factors are likely confounders between individuals with and without previous infection. Both people’s status as ‘previously infected’ and their COVID-19 infection outcome in this study are based on self-reported positive test results (at least in part). However, people’s testing behaviours are almost certainly not random with respect to status of previous infection. E.g. It is possible:

- i) People change their testing behaviour if they have had COVID-19 (and are less likely to test for COVID-19 – or the symptom threshold at which people seek testing is changed).
- ii) People who are still naïve for COVID-19 in March-Oct 2022 in the USA are not a typical cohort. This is evidence by 64% of the enrolled people were naïve for COVID-19 at a time when seroprevalence in the US was $>50\text{-}60\%$. This may be a cohort that is more likely to test at a lower symptom threshold.

Response: These are good points. To examine this issue, we reasoned that if naïve participants had a lower threshold for COVID-19 testing they should have fewer reported symptoms as captured by our case report forms. In a post hoc analysis we compared the reported symptoms between naïve and

non-naïve participants. Overall, naïve and non-naïve cases reported similar numbers of symptoms (naïve: average 6.3, non-naïve: average 5.9), and the symptoms were comparable between naïve and non-naïve participants (chi-squared $p = 0.122$). These data have been added to the revision as Supplementary Table 6.

To further alleviate the concern about outcome misclassification, e.g., people did not test and/or did not report mild COVID-19, we conducted an additional analysis focusing on COVID-19 of at least moderate severity (“moderate-to-severe COVID-19”), which was defined based on the CDC clinical criteria but with the modification that OR was changed to AND, as follows:

Clinical Criteria

In the absence of a more likely diagnosis:

- Acute onset or worsening of at least two of the following symptoms or signs:
 - fever (measured or subjective),
 - chills,
 - rigors,
 - myalgia,
 - headache,
 - sore throat,
 - nausea or vomiting,
 - diarrhea,
 - fatigue,
 - congestion or runny nose.

AND

- Acute onset or worsening of any one of the following symptoms or signs:
 - cough,
 - shortness of breath,
 - difficulty breathing,
 - olfactory disorder,
 - taste disorder,
 - confusion or change in mental status,
 - persistent pain or pressure in the chest,
 - pale, gray, or blue-colored skin, lips, or nail beds, depending on skin tone,
 - inability to wake or stay awake.

We found qualitatively similar results (Figure R1, below). Peak nAb-ID50 appeared to be a stronger inverse correlate of risk of moderate-to-severe COVID-19 for non-naïve participants compared to naïve participants. Moreover, among non-naïve participants, the inverse correlation between nAb-ID50 and moderate-to-severe COVID-19 appeared to be even stronger compared to that between nAb-ID50 and the symptomatic (any severity) COVID-19 endpoint analyzed in the rest of the manuscript, though the sample size is not large enough to draw a firm conclusion on this point.

Figure R1. Cox model covariate-adjusted hazard ratios of moderate-to-severe COVID-19 per 10-fold increase in D15 neutralizing antibody titer, shown separately in SARS-CoV-2 (A, B) naïve and (C, D) non-naïve participants. (A) and (C) show hazard ratios per 10-fold increase in D15 titer (AU/ml) for each marker BA.1, Beta, Delta, BA.4/BA.5, D614G, or weighted average (Wt. Avg.). (B) and (D) show hazard ratios per 10-fold increase in D15 BA.1 titer (AU/ml) in the designated subgroups. Analyses adjusted for force of infection score and risk score. Point estimates, 95% confidence intervals (CIs), and 2-sided p-values are shown. Wt. Avg. = Maximum diversity weighted geometric mean of the five nAb titers D614G reference, Beta, Delta, Omicron BA.1, and Omicron BA.4/BA.5. AU, arbitrary units; nAb-ID50, 50% inhibitory dilution neutralizing antibody titer.

c. Demonstratable demographic differences in the two groups that could cause confounding. The two groups in the study (previously infected, not previously infected) are demographically distinct (different ages/ethnicities). This supports the notions above that these is potential behavioural differences in these cohorts, especially around testing behaviour and thresholds for testing. Older people were less likely to have been infected by 2022 in the pandemic than younger people (according to seroprevalence data), but are also more likely to engage with health care interventions and testing.

They were likely to have more frequent booster vaccines, and also to test more frequently for COVID-19. The authors attempt to account for demographic differences in the two populations, but the authors have not discussed the limitations of these adjustments. They have also not discussed whether these can account for both risk of infection differences as well as testing propensity differences (almost certainly not because there is an implicit confounding on these two aspects).

Together these potential sources of confounding in the study could drive the primary observed conclusion of the study and explain the contradictory result of this study with the authors previous studies. Therefore, they must be (ideally) considered in the analysis, or these limitations raised in the discussion as potential reasons for the conclusions of the study.

Response: We agree there is a potential for residual confounding. However the potential unobserved confounders (testing and risk behavior) or poorly adjusted for observed confounders (age, ethnicity) are more likely to cause a difference in the *absolute* COVID-19 rates between naïve and non-naïve participants. For these confounders to change the Cox models *slopes* for the antibody covariate for naïve and non-naïve participants, we would need the confounders to also depend on the underlying (and, to the participant, unknown) antibody level – and, further, that this dependence would be different for naïve and non-naïve participants. A possible example would be if non-naïve participants have a very steep dropoff in testing proclivity/risk behavior as antibody increases, while naïve participants have little dropoff in testing proclivity/risk behavior as antibody increases. While this could happen, it is a more complex phenomena than testing/risk behavior differing by naïve/non-naïve status. We now write in the discussion (lines 625-630).

“Finally, because this is an observational study and antibody level and naïve/non-naïve status were not randomly assigned, it is possible that confounding partly drives the difference in slopes between the naïve and non-naïve groups. To drive a difference in slopes, however, this confounding would need to be complex, where say testing and risk behavior depends on the underlying and unknown antibody level in naïve participants in a different way than for non-naïve participants”.

Please also see our response to the reviewer’s main concern (a), i.e. the correlates results using the COVE endpoint. These results not only enable comparison to the previous COVE booster correlates analysis, but also further guard against outcome misclassification, i.e., misclassifying a participant as a non-case, due to health seeking behaviors (testing frequency, tendency to report, etc).

2. A high degree of complexity in analysis and low transparency (black box pipeline).

The authors approach to analysing a correlate of risk here involved numerous complex steps in a pipeline, including:

- risk adjustments based on a machine learning approach
- risk adjustment based on calendar time using a database
- (in the case of proximal correlates) antibody decay modelling with a mixed effects model
- cox model of risk adjusted data, sometimes with and sometimes without risk-adjustment.

These methods were all pre-specified and there is no risk of cherry-picking analysis tools to get a particular result, which is a strength of the work. None-the-less it is not possible in its current form to really understand whether any of the steps in the pipeline (especially around risk adjustments and covariates in the modelling) are correctly eliminating (or introducing) confounding. All methodologies have assumptions and limitations. I am not across the details of the risk adjustment methodology used by the authors, but it will certainly have assumptions and limitations. Could these be introducing

unintentional confounding in the analysis and/or driving the apparent contradictory result observed here of a weak correlation between neutralisation and risk in naïve individuals? In addition, the authors should demonstrate how each element of the analysis pipeline is influencing the results. If the super learner risk adjustments are not performed and a simple cox-regression analysis is performed (perhaps with or without covariates) how are the correlates of risk altered? The authors can achieve this by re-running the analysis pipeline with and without various aspects of the pipeline.

Also, without a naïve control population in this RCT it is hard to see how the risk adjustments are performed with the super learner? In the authors previous work my understanding is that these risk adjustments were informed by the naïve placebo control groups? How is risk adjustment made with only intervention arms?

Response: Super Learner is applied to the entire group, without treatment indicator or antibody titer in the model. The output of the superlearner is effectively a scalar summary of a vector of risk factors, blinded to the antibody level or treatment indicator. This is akin to a prognostic score often used in clinical research. Using such a scalar prognostic variable in lieu of a vector of risk factors is motivated by the relatively small sample size and number of endpoints in these survival analyses. In sensitivity analyses (see the response below), we further dropped the risk score, and the results were virtually unchanged.

The authors must show the impact of analysis choices on the outcome of the analysis (by reducing model complexity and showing how results change). Further, the authors discuss some limitations around the data but do not discuss the potential assumptions and limitations of the methods adopted.

Response: The reviewer raises two points – model complexity and methods assumptions/limitations – which we address in turn.

To reduce model complexity, we adopted the advice from the reviewer and repeated our analyses with and without various aspects of the pipeline.

In our original submission we (1) repeated the peak analyses replacing the risk score with an indicator of age ≥ 65 (so the model is fit separately for naïve and non-naïve participants and only adjusts for the FOI score + age ≥ 65 indicator (Supplementary Figures 15 and 16 in the originally submitted supplement). In the revision, we have added: (2) further excluding the FOI score (so the model is fit separately for naïve and non-naïve participants and only adjusts for the age indicator); (3) further excluding the age indicator (so the model is fit separately for naïve and non-naïve participants and does not adjust for anything). These plots have been added to Supplementary Figures 16 and 17 (panels C-F of each figure), such that each of these supplementary figures shows the results of these three different sensitivity analyses in naïve and in non-naïve participants (analogues of Figure 4B, 4D and Figure 5A, 5C). In addition, we have added a sensitivity analysis for the exposure-proximal correlates analyses with no covariate adjustment. Forest plots summarizing this sensitivity analysis have been added as Supplementary Figure 27.

As we have now demonstrated via these comprehensive stress tests, the observed differential HRs are robust to covariate adjustment (see e.g. a table below providing the D15 BA.1 nAb results, where the source HRs are shown in Figures 5, S16, and S17).

Group (Naïve/non-naïve)	HR of COVID-19 per 10-fold increase in D15 BA.1 nAb titer, with adjustment for:				Source Figs
	FOI score and risk score ("standard" adjustment)	FOI score and the ≥ 65 age indicator (Sensitivity #1)	≥ 65 age indicator only (Sensitivity #2)	No covariate adjustment (Sensitivity #3)	
All one-dose mRNA, naïve	0.85 (0.67, 1.09)	0.86 (0.68, 1.11)	0.87 (0.68, 1.11)	0.87 (0.68, 1.11)	5a; S16a, S16c, S16e
All one-dose mRNA, non-naïve	0.52 (0.33, 0.82)	0.51 (0.32, 0.81)	0.48 (0.29, 0.79)	0.50 (0.31, 0.81)	5c; S16b, S16d, S16f
All mRNA Moderna, naïve	0.71 (0.52, 0.97)	0.67 (0.49, 0.94)	0.67 (0.48, 0.93)	0.74 (0.54, 1.01)	4b; S17a, S17c, S17e
All mRNA Pfizer, naïve	0.65 (0.33, 1.26)	0.65 (0.34, 1.24)	0.68 (0.35, 1.31)	0.67 (0.35, 1.30)	4b; S17a, S17c, S17e

Regarding the assumptions and limitations, see response to main point 1.

3. Data not available.

Finally, because of the model complexity it is absolutely critical that the full underlying data be made available in this study for readers to critically assess the correlates analysis. The original study does not make the Full data available either. Of note: There are no ethical concerns with releasing the data, because all of this data can be de-identified. E.g. If the authors felt age-buckets instead of exact age would ensure no identifiability concerns that would also be sufficient. As it stands, there is no way to assess the analytic pipeline or reproduce (or critique) the result.

Response: In our revised Data Availability statement, we provide details on how the dataset can be requested. In sum, interested parties can direct an inquiry to the corresponding author, and after evaluation by the NIH Division of Microbiology and Infectious Diseases, the data will be shared via data transfer agreement if the request is approved.

Other issues:

Reporting:

1. The authors say previous infection was defined as either self-reported or anti-N positive. What proportion of previous infection was characterised by either of these criteria?

Response: Among the N = 356 non-naïve participants included in the correlates analysis per-protocol cohort (Supplementary Fig. 2), 341 tested anti-N positive at baseline; the other 15 tested anti-N negative and self-reported a previous SARS-CoV-2 infection.

We have added (underlined) similar information in the revision for the cumulative incidence analysis cohort:

“Of the 1006 participants eligible for cumulative incidence analyses (Supplementary Fig. 2), 648 (64.4%) were naïve. The remaining 358 (35.6%) were non-naïve, almost all of whom (342/358, 95.5%) had a positive anti-N test result at baseline. The remaining 16 non-naïve participants (4.5%) tested anti-N negative at baseline and were classified as non-naïve due to self-reporting a previous SARS-CoV-2 infection.” (lines 198-203)

2. The authors report p-values for interaction analysis, but not effect size (e.g. lines 282-283). It would be good to see effect sizes in the sup. table S5.

Response: We have now added the exp(coefficient), interpreted as a ratio of HRs, to Supplementary Tables 7, 8 and 9. In addition, in this version, we also reported results with and without covariate adjustment in the interaction test.

3. Covariates adjusted for is not clear. It is very hard to work out what covariates are adjusted for in covariate adjusted analysis. Was it the same set of covariates in all analyses. Did this include previous infection status? If so, this seems like it may unintentionally introduce a confounder (via collider?) in the regression? Could the authors spell out more clearly what covariates are included? It is apparently in the SAP, but I could not easily find it. On page 59 I see a list of “relevant participant co-variates” but this includes previous infection status.

Response: The peak correlates analyses were all conducted separately in naïve and in non-naïve participants. Because the analysis was conducted separately in naïve and non-naïve participants, we adjusted for the baseline risk score and the FOI score but not baseline naïve/non-naïve status. For the peak correlates analyses, we also conducted 3 sensitivity analyses examining each component of the covariate adjustment strategy.

The exposure-proximal correlates analyses were all conducted separately in the naïve and non-naïve participants and adjusted for the baseline risk score. The FOI score was not adjusted in the exposure-proximal analyses because exposure-proximal analyses were calendar-time-based and took into account the external force of infection automatically.

We have made the following revisions (underlined) in Methods:

Lines 673-681:

“Correlates of risk analyses

Univariable correlates of risk analyses were performed in the correlates analysis per-protocol cohort (Supplementary Fig. 2), pooling across the 13 one-dose mRNA arms (Supplementary Fig. 1) and in pre-specified subgroups determined by vaccine received, separately for baseline naïve and non-naïve participants. The analyses were based on Cox regression models adjusting for the FOI score and baseline risk score and conducted separately for each D15 and fold-rise nAb ID50 marker. Sensitivity analyses were also conducted to examine each component of the covariate adjustment strategy [1] adjustment only for the FOI score and for the ≥ 65 age indicator; 2) adjustment only for the ≥ 65 age indicator; 3) no covariate adjustment].”

Lines 738-740:

“Exposure-proximal correlates of risk analysis

... These analyses were conducted separately in naïve and in non-naïve participants, and adjusted for the baseline risk score. Because exposure-proximal analyses operated on the calendar-time-scale, the external force of infection was automatically accounted for and no FOI score adjustment was done.”

Ab Decay modelling:

4. It is not possible to assess how well the decay model is fitting the data since the data and fitted model

is nowhere clearly presented. There is a random set of 10 individuals given in figure S21, but this is not adequate.

Response: In addition to Figure S21 (renumbered to Supplementary Figure 25 in the revision) we now provide scatterplots of the predicted (based on a biphasic antibody decay model) and measured log₁₀ titers at D15, D29, D91, and D181 for the different assays in Supplementary Figure 26 (also shown below). There is high concordance between the observed and predicted values, with correlations ranging from 0.96 to 0.98. Another way to think of this is via the reliability ratio, which is 1 minus (the variance of the observed minus predicted values over the variance of the observed values). This ranges from 90% to 95%, with values higher than 90% considered excellent.

We have added the following text to the revision (lines 385-390):

“Supplementary Fig. 26 additionally plots measured versus predicted (based on a biphasic antibody decay model) log₁₀ ID₅₀ titers at D15, D29, D91, and D181, separately for BA.1, Beta, Delta, BA.4/BA.5, D614G, and the weighted average marker. High concordance was seen between the observed and predicted values for each variant/marker, with Pearson correlations ranging from 0.95 to 0.98 and reliability ratios ranging from 90% to 95%.”

Caption: Pearson Cor.: Pearson Correlation. Var(Obs. log titer): variance of the observed log₁₀ titer. Var(Obs. - Pred.): Variance of the difference between the observed log₁₀ titer and the predicted log₁₀ titer.

5. It is important to understand over time how many neutralisation titres dropped below the detection limit, and whether the decay generally looks exponential over the time period being modelled (or is it biphasic etc? – and this might differ by group)?

Response: We investigated and found that for some of the assays a biphasic decay model did fit better, and have thus replaced the results from the previous linear mixed effects model with those from a biphasic model with a bendpoint at day 76, different (fixed effects) slopes and intercepts for the naïve and non-naïve groups, and a random intercept. The estimated hazard ratios were virtually unchanged with the new model, for example, the hazard ratios per 10-fold increase in predicted-at-exposure BA.1 titer obtained using the previous LME model were: naïve participants: 0.75 (0.61, 0.93), $p = 0.008$; non-naïve participants: 0.43 (0.28, 0.65), $p < 0.001$. By comparison, the same hazard ratios obtained using the new biphasic model were: naïve participants 0.74 (0.59, 0.94), $p = 0.016$; non-naïve participants: 0.41 (0.23, 0.64), $p < 0.001$.

The following items have been updated or added in the revision:

Methods (lines 698-716) (updated)

Figure 5, panels b and d (updated)

Figure 6, all panels (updated)

Supplementary Fig. 25 (updated)

Supplementary Fig. 26 (added)

Supplementary Fig. 27 (added)

Supplementary Fig. 28 (added)

Supplementary Fig. 31, all panels (updated)

6. It is unclear whether the authors account for (or need to account for) values below the limit of detection? The supplementary methods say they will do this (with imputation) if more than 10% are below the limit of detection? But it doesn't say if they had to do this or not? Also, what if the 10% that are below LOD are all at later time points or in one group? Could this not introduce a bias? This seems important since it seems more likely that the low titres in individuals without previous infection would be more likely to fall below the LOD over time?

Response: We did not perform imputation because the % of values below LOD for all assays was less than 10%.

Marker	# below LOD	% below LOD
BA.1	84	2.5%
Beta	33	0.9%
Delta	16	0.5%
BA.4.5	137	5.0%
D614G	10	0.3%
MDW	16	0.5%

To further investigate the reviewer's concern, we focused on the BA.1 assay, looking at % LOD by time and naïve/non-naïve status. We see that indeed there are more below LOD at later time points and for the naïve group. In our analysis we set values at LOD to LOD/2. To address the reviewer's concern we switched to a random effects model with tobit errors. This model assumes that values below the limit of detection are interval censored below the limit of detection i.e. follow a truncated Gaussian distribution. Given the steady decay of antibodies, this model is the natural way to address the limit of detection and we now use this approach to model antibody decay in our paper. Interestingly, the results are virtually unchanged from using LOD/2, for example for BA.1 the hazard ratio for naïve participants was 0.74 (95% CI 0.59, 0.94) using the tobit model and 0.74 (95% CI 0.59, 0.93) using LOD/2 when both models had a bend point at 76 days after day 15. For non-naïves, the analogous estimates were 0.41 (95% CI 0.23, 0.64) using the tobit model and 0.42 (95% CI 0.23, 0.64) using LOD/2 when both models had a bend point at 76 days after day 15.

Visit day	Number and % below LOD Naïve	Number and % below LOD Non-naïve
15	8/629 (1.3%)	3/356 (0.8%)
29	7/595 (1.3%)	3/343 (0.9%)
91	19/480 (4.0%)	3/328 (0.9%)
181	36/308 (11.7%)	5/288 (1.7%)

7. The authors do not consider a random effect on the slope between individuals. It seems possible/likely there is random variation in the decay of neutralising antibodies between individuals.

Response: This is a fair point, and we tried to fit models with a random slope as well as a random intercept. However, the random slope models had problems converging. Nonetheless, we did find good agreement between observed and predicted titer with the random intercept model (see our response to point 4 above).

Proximal Correlates:

8. If I am understanding correctly, the uncertainty in the predicted titres for each individual do not seem to be taken into account in the correlates analysis. i.e. there is clearly significant uncertainty in the predicted Ab titre for each individual, and this uncertainty doesn't seem to be taken into account – rather it is used as though it is an errorless estimate? Consider including this uncertainty in the correlates analysis?

Response: We now incorporate this uncertainty by bootstrapping the entire procedure. Specifically, we use a Bayesian bootstrap which gives a random weight for each participant. For n participants the weight vector is given by w_1, \dots, w_n where the vector w_1, \dots, w_n is a random draw from a Dirichlet distribution with parameters $(n, 1/n, \dots, 1/n)$. Thus the weights sum to n , just as the sample size remains n for the regular bootstrap where participants are sampled with replacement. The weight for each participant was applied to both the antibody decay modeling and the Cox modeling. The Bayesian bootstrap was applied separately for naive and non-naive participants. There was a slight increase in the standard errors with this bootstrap.

9. In the BA.5 analysis it is unclear that the authors do not now have more values below LOD, especially over time, and so the issues with proximal correlates model and decay described above are perhaps more of a concern here. Can the authors show the BA.5 titres over time, and the model fit to this data?

Response:

Below is a spaghetti plot of the BA.4/BA.5 log₁₀ titers over time. The smooth black curve is the mean trajectory.

To assess model fit, the scatterplot of observed to predicted BA.4/BA.5 titer (Panel D in Supplementary Fig. 26) is shown below, where the correlation was 0.97 between observed and predicted titers.

The table below shows the numbers and percentages of naïve and non-naïve participants with BA.4/BA.5 titers below the LOD at D15, D29, D91, and D181. As the reviewer surmised, these numbers were indeed higher than those for BA.1 (see the BA.1 table in the response to point [6] above).

Visit day	Number and % below LOD Naïve	Number and % below LOD Non-naïve
15	3/153 (2.0%)	0/166 (0%)
29	22/594 (3.7%)	4/344 (2.1%)
91	47/480 (9.8%)	3/328 (0.9%)
181	53/308 (17.2%)	5/288 (1.7%)

We also now use the tobit model for antibody decay throughout the manuscript. For BA.4/BA.5, the hazard ratio for naïve participants across all mRNA groups was 0.74 (95% CI 0.58, 0.94) using the tobit model and 0.74 (95% CI 0.59, 0.93) using LOD/2 when both models had a bend point at 76 days after day 15. For non-naïve participants across all mRNA groups, the hazard ratio for BA.4/BA.5 was 0.43 (95% CI 0.25, 0.71) for the tobit model and 0.44 (95% CI 0.24, 0.71) using LOD/2 when both models had a bend point at 76 days after day 15.

Discussion:

10. The reasons the authors provide for the surprising conclusion of a different association between neutralisation and risk between groups seems to contradict the data here and in previous studies. The authors say that the different correlation for naïve and non-naïve may be the result of:

“(1) a stronger relationship between antibody and risk at higher antibody levels, and little relationship at lower levels for which the quantity of antibody was too low to generate or mark protection; (2) B cells, T cells, and other factors following natural infection have greater synergy with antibody at higher titers, compared to B and T cells from vaccination alone; and/or (3) boosted higher titer antibody following natural infection is qualitatively better even at the same neutralization titer.”

But, all of these explanations assume that the difference between the correlation in naïve and non-naïve populations (i.e. steeper for non-naïve), is because the association gets stronger in non-naïve people. But the authors themselves note that the association for naïve people is weaker than previously observed. So it seems like it’s the naïve cohort in this study that is the outlier here? At similar titres the authors have previously shown that naïve people have a stronger correlation of neutralisation and protection than observed here (e.g. correlates in JnJ vaccinees, Fong et al., Nat Micro 2022).

Response: Thank you for this very good point. This study was conducted in Spring 2022 after the initial (BA.1) Omicron wave and when the majority of the population had been infected. Most other correlates analyses included in the paper you cite and a more recent one from Zhang et al. 2024 were conducted in 2020 and 2021 when a minority of the population had been infected. Thus the COVAIL naïve population is presumably lower risk than the naïve populations of previous studies. Furthermore, in COVAIL, COVID-19 was mostly BA.2 and BA.5 while earlier studies were mostly ancestral era strains of COVID-19. As the COVAIL naïve group had been primed with ancestral strain

vaccines and their first booster dose included ancestral strain, COVAIL naïve participants have more of a mismatch between immune priming and infecting strain than previous naïve groups.

Of these two reasons (lower risk, greater discordance between imprinting and infecting virus), we think the latter seems more reasonable as an explanation for why the naïve group has a weaker relationship in COVAIL compared to earlier studies.

The authors must spell out more clearly in the discussion that it seems the reason for their primary conclusion is a change in the relationship between neutralisation and protection in naïve people compared to previous observations. This in itself may point to either something being different about a 4th booster dose, though this seems difficult to interpret, or something different about this RCT in itself. The latter seems more likely to this reviewer – though the former cannot be excluded.

Response.

We think there are two issues: 1) The relationship in naïve participants for COVAIL differs from before; and 2) The relationship between naïve and non-naïve participants differs. We address both in the discussion. Regarding point 1, we now write

“While the reason for the weaker relationship in COVAIL is not clear. A potential reason may be related to immune imprinting. The primary immunization for naïves in previous studies and for COVAIL was from ancestral strain vaccines. In previous studies the circulating strains of SARS-CoV-2 were more similar to the vaccine strain than for COVAIL which had a preponderance of BA.2 and BA.5 strains. The immune correlate relationship may weaken with greater discordance between imprinted strain and circulating strain.” (lines 523-529)

Reviewer #2 (Remarks to the Author):

In the manuscript “Neutralizing antibody immune correlates by prior SARS-CoV-2 infection status in COVAIL trial recipients of an mRNA second COVID-19 vaccine boost” by Bo Zhang et al., the authors present an analysis of correlates of risks and correlates of protection for COVID-19 infection in the context of the COVAIL trial. The trial included 985 participants, who received COVID-19 vaccination boosters (meaning they had been vaccinated previously) that were either monovalent or bivalent vaccines. The authors measured neutralizing antibody (nAb) titers against five different strains of COVID-19 up to 6 months after administration of the booster vaccine. They observed that higher nAb titers reduced the risk of infection with the SARS-CoV-2 Omicron strain significantly and found that the risk reduction was stronger in individuals previously infected with SARS-CoV-2.

While the study is in principle interesting, the description of the results should be improved. For example, in the abstract the authors mention that they assess “titers [...] as correlates of COVID-19” where they could have meant to say “COVID-19 disease outcome” or “COVID-19 infection risk”. The same can be said for the introduction where they use a similar wording. Furthermore, in the introduction they use a couple of terms, such as “exposure-proximal” or “COVID-19 endpoint”, which are only explained in the results section. This makes a lot of the text incomprehensible at first. While these might be minor language issues, the frequency at which this occurred was striking.

Response: Due to the relatively tight word limit of the Abstract, we now state “symptomatic COVID-19 (“COVID-19”)” and use “COVID-19” thereafter. The revised portion now reads “correlates of risk of symptomatic COVID-19 (“COVID-19”) and as correlates of relative (Pfizer-BioNTech Omicron vs. Prototype) booster protection against COVID-19”. We also have added (“predicted-at-exposure”) at first instance after “exposure-proximal” in the Abstract.

The same approach is applied in the Introduction through the following additions:

- “In this work we studied how antibody level correlates with symptomatic COVID-19 (hereafter, “COVID-19”) (lines 133-134)”
- “In the present analysis, the COVID-19 endpoint was the symptomatic subset of self-reported or study-conducted positive SARS-CoV-2 tests (details below).” (lines 157-159)
- “exposure-proximal²⁶ (i.e., predicted level at the time of exposure leading to a COVID-19 endpoint)” (lines 150-151)

Apart from the language and structural issues of the text, the description of the statistical methods is in my opinion insufficient. The major issue is the insufficient description of the baseline risk of infection. The authors mention the use of a “superlearner”, which is described in more detail in Section 6.3 of the SAP. The superlearner learns the baseline risk of infection, which is used in adjusting their “correlates analyses”. This superlearner adjusts for many different “baseline covariates” (such as age and sex) and the text gives the impression that the output of the superlearner is equal to in their description of the Cox-model. However, it is unclear whether it is, because it is not individual specific and does not depend on the covariates in the equation. It is also unclear where the force of infection score that they calculate is included in the model. Additionally, the Cox-model adjusts for the same covariates again in, which would make the interpretability of the effect of the covariate more difficult.

Response: The risk score is a function of baseline covariates and takes on a different value for each participant. We would like to clarify that statistical analyses prespecified in the SAP adjusted for the risk score (in lieu of the covariates the risk score depends on) and the FOI score. Adjusting for a risk score is due to practical consideration about that the number of endpoints is relatively small compared to the number and level of baseline covariates; hence, it is more practical to adjust for a summary of these covariates, which is the risk score.

We have adopted the advice from the other reviewer and repeated our analyses with and without various aspects of the pipeline. In our original submission we (1) repeated the analyses replacing the risk score with an indicator of age ≥ 65 (so the model is fit separately for N and NN and only adjusts for the FOI score + age ≥ 65 indicator (Supplementary Figures 15 and 16 in the originally submitted supplement). In the revision, we have added: (2) further excluding the FOI score (so the model is fit separately for N and NN and only adjusts for the age indicator); (3) further excluding the age indicator (so the model is fit separately for N and NN and does not adjust for anything). These plots have been added to Supplementary Figures 16 and 17 (panels C-F of each figure), such that each of these supplementary figures shows the results of these three different sensitivity analyses in naïve and in non-naïve participants. A similar sensitivity analysis with no covariate adjustment was also performed for the exposure-proximal markers and has been added as Supplementary Figure 27. As we have now demonstrated via these comprehensive stress tests, the observed differential HRs are robust to covariate adjustment.

We have also conducted additional investigation of the interaction test. Please see our response to Results 1a.

In addition to this, it is in my opinion crucial that:

- 1) the authors make the datasets available (see first comment below),

- 2) reveal already in the instruction that the dataset has been studied before and has not been extended in this manuscript, and
- 3) discuss the what's novel insights have been obtained here compared to the previous studies.

Response:

- 1) **Please see the revised data availability statement: “Requests for the minimum dataset necessary to reproduce the results reported in this manuscript should be directed to the corresponding author, Dr. Dean Follmann (dfollmann@niaid.nih.gov). The request will be evaluated by the NIH Division of Microbiology and Infectious Diseases (DMID) and the data shared via data transfer agreement, if approved.”**
- 2) **We have added: “All data analyzed in the present work have been previously studied in immunogenicity analyses.^{18,19” (lines 213-241)}**
- 3) **The COVAIL study was unique in that sufficient cases of largely BA.2, BA.4, and BA.5 COVID-19 accrued among recipients of a second booster variant dose such that extensive immune correlates analyses contrasting inserts and assays to different lineages, including Omicron, could be performed. Most previous immune correlates studies of COVID-19 vaccines using individual-level data were conducted pre-Omicron, and in SARS-CoV-2 naïve individuals. In Zhang et al.’s Omicron-era immune correlates analysis of a third dose of mRNA-1273 using individual-level data in the COVE trial, the boost was ancestral (i.e. no variant boosters were used). As we state in lines 609-611: “To our knowledge, this is the first randomized trial with an individual-level immune correlates analysis conducted of a booster variant dose.” Therefore, one novel insight of the present study was that it could assess – in the Omicron era – whether, in the context of a variant boost, matching the variant of the COVID-19 outcome with neutralization titer provides a stronger correlate relationship than an unmatched titer. We have revised the Discussion (underlined) to emphasize this (lines 570-578): “An important question is whether, in the context of a variant boost, matching the variant of the COVID-19 outcome with neutralization titer provides a stronger correlate relationship than an unmatched titer. To our knowledge, this question has not been addressed in previous analyses of individual-level data. We addressed this question by comparing the hazard ratios for an unmatched analysis (D614G titer correlated with BA.4/BA.5 Omicron COVID-19) with a matched analysis (BA.4/BA.5 Omicron titer correlated with BA.4/BA.5 Omicron COVID-19). Generally, the hazard ratios and covariate-adjusted cumulative incidence functions were quite similar for naïve participants and nearly identical for non-naïve participants, though with slightly better precision with variant matching.”**

Another novel feature of our study is the ability to assess Omicron era correlates of risk in both naive and non-naive participants following a 4th dose, which we thoroughly describe in the discussion section.

Further major and minor issues are also listed below.

Major issues:

0. Data

The authors state that "All data were previously included with Branche et al.^{18,19}". I had a detailed look at these papers. While they say that "All data are included in the paper.", I was unable to find it.

Furthermore, the code refers to a very specific data file "covail_data_processed_20240313.csv", which

given that data seems to the newer that the two cited publications.

In my opinion it is crucial to share that data to ensure reproducibility. If indeed data is used from a previous publication and I failed to find it, a direct link should be provided and the data from the original study should be used directly in the code. If this is not done, the data reformatting needs to be explained in details.

Response: Please see our revised Data Availability statement.

1. Results

a. D15 (Peak) titer inverse correlates of risk of COVID-19 stronger for non-naïve than naïve participants
 i. “Cox model interaction P = 0.002-0.009 for the six markers for boost-proximal cases”: It is unclear what is meant by “Cox model interaction”. According to the equations in Section 6.3 in the SAP, there are no interactions between two antibody titers in the sense that the Cox model includes . It is especially unclear, because in section 6.3 of the SAP, they do include an interaction term between naïve status and the antibody titer, but I do not think that it is used here. Furthermore, since the authors perform univariate Cox regression, the authors should be careful with expressions such as “associations [...] are generally stronger” or “all six D15 titer markers were inverse correlates of risk” without multiple-testing correction.

Response: We have interrogated the interaction test more thoroughly, and report the results in the updated Supplementary Tables 7-9 in the revision.

In Supplementary Table 7 (also copied below) we show the results of interaction test (~ naïve + peak marker + naïve * peak marker), with and without adjusting for the risk score or the FOI score, in the boost proximal period and the entire period. I scale all the markers. The column header “HR Ratio” is the ratio of HRs, so 1.632 means the HR (per 1 SD increase) for naïves is 1.632 times the HR (per 1 SD increase) for non-naïves.

		Boost Proximal					Entire Period					
		Sample size	Endpoints	Without Cov. Adjust.		With Cov. Adj.		Endpoints	Without Cov. Adjust.		With Cov. Adj.	
				HR Ratio	p	HR Ratio	p		HR Ratio	p	HR Ratio	p
BA.1	985	144	1.632	0.004	1.632	0.004	213	1.414	0.028	1.402	0.028	
Beta	985	144	1.555	0.009	1.538	0.01	213	1.343	0.05	1.336	0.052	
Delta	985	144	1.544	0.006	1.543	0.006	213	1.328	0.042	1.341	0.033	
D614G	985	144	1.506	0.005	1.506	0.005	213	1.266	0.071	1.278	0.058	
BA.4/BA.5	985	144	1.818	0.002	1.836	0.002	213	1.532	0.014	1.511	0.018	
MDW	985	144	1.691	0.002	1.679	0.002	213	1.418	0.025	1.413	0.025	

Because there is a concern that the differentiate HR between naïves and non-naïves may be an artifact of the limited overlap of the peak titer between naïves and non-naïves, we then repeated the same interaction test (~ naïve + peak marker + naïve * peak marker, with and without adjusting for the risk score or the FOI score) but restricted to the overlapped peak titer region (defined as the intersection of the middle 90% quantiles of naïves and that of non-naïves). The results are shown in Supplementary Table 8 (also copied below). For instance, for BA.1, this range is from 3.3 to 4.4 in the log₁₀-scale, which is in the high peak titer territory. The table below summarizes the results. The point estimates of the coefficient of the interaction term (ratio of HRs) are relatively stable, though covariate adjustment appears to further increase the precision. Anyway, with or without covariate

adjustment, there is evidence that the HRs differed even in the relatively high peak level territory (e.g., between 3.3 and 4.4 for BA.1).

		Boost Proximal					Entire Period					
		Sample size	Endpoints	Without Cov. Adjust.		With Cov. Adj.		Endpoints	Without Cov. Adjust.		With Cov. Adj.	
				HR Ratio	p	HR Ratio	p		HR Ratio	p	HR Ratio	p
BA.1	765	103	2.211	0.005	1.632	0.004	150	1.511	0.057	1.402	0.028	
Beta	789	106	1.350	0.256	1.538	0.010	154	1.163	0.479	1.336	0.052	
Delta	780	107	1.662	0.068	1.543	0.006	157	1.399	0.139	1.341	0.033	
D614G	781	109	1.674	0.071	1.506	0.005	160	1.239	0.347	1.278	0.058	
BA.4/BA.5	741	97	2.427	0.004	1.836	0.002	144	1.525	0.076	1.511	0.018	
MDW	753	99	2.113	0.011	1.679	0.002	148	1.504	0.081	1.413	0.025	

Finally, a second sensitivity analysis where the same interaction test (~ naïve + peak marker + naïve * peak marker, with and without adjusting for the risk score or the FOI score) but restricted to those with detectable D1 titer. The results are shown in Supplementary Table 9 (also copied below).

		Boost Proximal					Entire Period					
		Sample size	Endpoints	Without Cov. Adjust.		With Cov. Adj.		Endpoints	Without Cov. Adjust.		With Cov. Adj.	
				HR Ratio	p	HR Ratio	p		HR Ratio	p	HR Ratio	p
BA.1	813	104	1.759	0.001	1.632	0.004	156	1.429	0.021	1.402	0.028	
Beta	933	135	2.083	0.003	1.538	0.010	200	1.607	0.023	1.336	0.052	
Delta	963	138	2.770	0.001	1.543	0.006	207	1.904	0.007	1.341	0.033	
D614G	973	142	3.725	<.001	1.506	0.005	211	2.132	0.002	1.278	0.058	
BA.4/BA.5	699	80	1.850	0.004	1.836	0.002	117	1.461	0.040	1.511	0.018	
MDW	813	104	1.759	0.001	1.632	0.004	156	1.429	0.021	1.402	0.028	

The last minor issue is the usage of the capital “P” for reporting p-values, as the convention is to use a lower case “p”.

Response: We have changed all capital “P”s to lower-case “p”s in the text and tables.

ii. “Post-hoc sensitivity analyses”: Missing or unclear description of the sensitivity analysis.

Response: These refer to the sensitivity analyses described above. We have revised to “In addition, we repeated the interaction tests restricted to the intersection of the middle 90% of titer values (Supplementary Table 8) and to participants with detectable D1 titers (Supplementary Table 9) in two post hoc sensitivity analyses. The results also supported interactions (Supplementary Tables 8, 9).” (lines 304-307)

b. The authors state that “A participant was determined to be SARS-CoV-2 non-naïve (hereafter, “non-naïve”) if they self-reported a previous SARS-CoV-2 infection or had detectable anti-N antibodies”. The latter seems to rely on a specification of a threshold. It would be beneficial if this threshold is reported.

Response: Anti-N seropositivity was assessed using the Elecsys Anti-SARS-CoV-2 assay, which has a cutoff index threshold of 1.0. We now write

“A participant was determined to be SARS-CoV-2 non-naïve (hereafter, “non-naïve”) if they self-reported a previous SARS-CoV-2 infection or had detectable anti-N antibodies [defined as Elecsys Anti-SARS-CoV-2 assay (Roche) cutoff index ≥ 1.0] at D1.” (lines 195-197)

2. Discussion

a. “with clear conclusion of weaker inverse correlates than previously found”: No reasoning for this discrepancy is given.

Response: We have added the following (lines 524-530):

“While the reason for the weaker relationship in COVAIL is not clear, a potential reason may be related to immune imprinting. The primary immunization for naïve participants in previous studies and for COVAIL was from ancestral strain vaccines. In previous studies the circulating strains of SARS-CoV-2 were more similar to the vaccine strain than for COVAIL which had a preponderance of BA.2 and BA.5 strains. The immune correlate relationship may weaken with greater discordance between imprinted strain and circulating strain.”

b. “non-naïve and naïve participants at the same D1 titer had the same predicted D15”: Prediction of D15 titers was not discussed in the text.

Response: This was implicit from Figures 2A and 2B where we state the linear regressions were virtually identical for naïve and non-naïve participants. We have made this clear now by adding

“, i.e. leading to the same predicted D15 titer for a given D1 titer whether naïve or non-naïve. ”

when we discuss Figure 2 (lines 274-275).

3. Methods

a. Antibody decay model for supporting exposure-proximal correlates analysis

i. Clear description of the mixed-effects model is missing.

Response: We have added the paragraph below to the Methods section (lines 701-716).

“For the correlates analyses, the antibody decay model was piecewise linear with a bend at day 76, separate (fixed effects) terms for naïve and non-naïve participants, and a random intercept.

Specifically,

$$\begin{aligned} X_i(t) = & \beta_0 + \beta_1(t - t_{i,d15}) + \beta_2(I(NposD1 = 1)) + \beta_3(t - t_{i,d15})I(NposD1 = 1) \\ & + \beta_4(t - t_{i,d15} - 76) \left(I(t - t_{i,d15} \geq 76) \right) \\ & + \beta_5(t - t_{i,d15} - 76) \left(I(t - t_{i,d15} \geq 76) \right) I(NposD1 = 1) + b_i + \epsilon_{i,t} \end{aligned}$$

where $t_{i,d15}$ is the calendar date of the D15 measurement for individual i , $X_i(t)$ is the time-varying log10 antibody titer on calendar day t for individual i , $t - t_{i,d15}$ is the number of days since the D15 measurement for individual i on calendar day t , $I(NposD1 = 1)$ is an indicator of being anti-n positive on D1, $(t - t_{i,d15} - 76)$ is the number of days since 76 days after the D15 measurement for individual i on calendar day t , $I(t - t_{i,d15} \geq 76)$ is an indicator of

whether calendar day t is at least 76 days after the day 15 measurement for individual i , $b_i \sim N(0, \tau^2)$ is an individual-level random intercept, and $\epsilon_{i,t} \sim N(0, \sigma^2)$. Values below the LOD are interval censored below LOD, i.e. follow a tobit model. The model is fit to the D15, D29, D91, and D181 measurements.”

ii. The model does not seem to have any random effects, according to the authors’ description. This would be a clear limitation.

Response: We do have a random intercept, see above. We did try a random slope but some of the models had problems converging so we just used a random intercept model. The model gives good predictions as described in the response to comment #9 of reviewer #1.

b. About the Cox models:

i. Insufficient description of the use of the baseline risk of infection and force of infection scores. The supplement provides additional details, but it remains for instance unclear what’s meant by the statement: “the peak correlates analyses also adjusted for an FOI score”.

Response: Please see the response to reviewer #1, “Other issue” #3.

ii. Throughout the text it is unclear whether the Cox models were separately estimated for naïve and non-naïve participants or whether the Cox model simply included a covariate with the naïve status information.

Response: The models were fit separately in naïve and non-naïve participants, except for models that tested for an interaction between naïve/non-naïve status and Antibody.

CLARIFICATION

Minor issues:

1. Abstract

a. “the increasing number of vaccine boosters raise questions as to this surrogate’s contemporary performance”: Unclear how the number of vaccine boosters raise questions to the performance.

Response: Revised to “the pandemic’s evolution and the introduction of variant-adapted vaccine boosters raise questions as to this surrogate’s contemporary performance”.

b. “correlates of COVID-19”: Unclear what exactly they want to assess here.

Response: Revised to “...assessed as correlates of risk of symptomatic COVID-19 (“COVID-19”) and as correlates of relative (Pfizer-BioNTech Omicron vs. Prototype) booster protection against COVID-19...”

c. “exposure-proximal”: Meaning unclear

Response: Added “..exposure-proximal (“predicted-at-exposure”) titers...”

d. “Omicron COVID-19 risk”: Unclear if risk of death, severe disease course or infection risk.

Response: We have defined “COVID-19” above as “symptomatic COVID-19”. Space limitations in the Abstract preclude listing of the entire endpoint definition.

2. Introduction

a. “In this work we studied how antibody level correlates with COVID-19”: See abstract issue

Response: We have revised to: “...correlates with symptomatic COVID-19 (hereafter, “COVID-19”)...” (lines 133-134)

b. Distinction between CoR and CoP could be made clearer. In the latter one is trying to find causality while the first is just a study of associations. A source as given by Gilbert et al. (Reference 23 in this manuscript) would be good.

Response: After “...correlate of risk (CoR) and correlate of protection (CoP) analyses”, we have added three additional references:

Plotkin, S. A. Correlates of protection induced by vaccination. *Clin Vaccine Immunol* 17, 1055-1065, doi:10.1128/CVI.00131-10 (2010).

Plotkin, S. A. & Gilbert, P. B. Nomenclature for immune correlates of protection after vaccination. *Clin Infect Dis* 54, 1615-1617, doi:10.1093/cid/cis238 (2012).

Plotkin, S. A. & Gilbert, P. B. Correlates of Protection. In: *Vaccines*, Eighth Edition, Editors Walter Orenstein, Paul Offit, Kathryn Edwards, Stanley Plotkin. Pages 35-40. Elsevier Inc., New York. (2022).

Moreover, we have cited the Gilbert et al. reference on CoP analyses within the following sentence: “Formally, a CoR seeks to understand how the immune marker is associated with the clinical endpoint in a defined cohort such as a particular vaccine arm. CoP analyses²³ seek to understand how the immune marker potentially modifies and explains the vaccine efficacy of a vaccine arm vs placebo, or the relative protective efficacy of one vaccine arm vs a second vaccine arm.” (lines 135-139)

Note though that as laid out by Plotkin and Gilbert,⁹ a correlate of protection may be a mechanistic correlate of protection (i.e. a CoP that is “that is mechanistically and causally responsible for protection”) or a nonmechanistic correlate of protection (a CoP that is “not a mechanistic causal agent of protection”).

c. D614G, Ancestral, and Prototype variants: I assume all variants are the same, so one term is enough.

Response: They are not all the same. “Prototype” and “ancestral” each refer to the mRNA-1273 vaccine insert Spike sequence from the Wuhan-Hu-1 isolate (preferably referred to as the index strain per WHO guidelines to avoid geographical stigma). At the first mention of “Prototype” in the Abstract and in the main text, we now provide “(Ancestral)” directly after, and we have revised any subsequent “Ancestral” mentions to “Prototype”. The choice of the use “Prototype” is based on the precedent of Branche et al.^{10,11}, as it is helpful to the reader to have terms harmonized across this manuscript and the primary COVAIL papers.

D614G refers to the index strain (Prototype) harboring the D614G mutation.¹² Neutralizing antibody titers were assessed against Prototype Spike harboring the D614G mutation. We have clarified this in the text (underlined) at line 141:

“We previously reported that 50% inhibitory dilution (ID50) neutralizing antibody (nAb) titers against D614G (“Reference”, corresponding to Prototype harboring the D614G mutation)...”

d. “Inverse CoR”: A term that is more confusing than just “reduces risk of COVID-19 infection”.

Response: Since “reduces risk” implies causality, which is an unsupported implication, we prefer the term “inverse CoR”. As suggested in 2b above, we have added additional literature references on CoRs and CoPs.

e. “D1 to D15 fold-rise nAb titer”: Odd wording. Suggestion: Fold-change of antibody levels between D1 and D15.

Response: Revised to “...markers corresponding to fold-change of antibody levels between D1 and D15 were not inverse CoRs.”

f. “exposure-proximal”: Meaning unclear

Response: Revised to “exposure-proximal (i.e., predicted level at the time of exposure leading to a COVID-19 endpoint)”.

g. “COVID-19 endpoints”: Unclear what endpoints they mean at this point of the text.

Response: We now add in the Introduction “In the present analysis, the COVID-19 endpoint was the symptomatic subset of self-reported or study-conducted positive SARS-CoV-2 tests (details below).” (lines 157-159)

h. “this study affords substantial precision”: What does this mean?

Response: Revised to “substantial precision could be achieved in the analyses”.

3. Results:

a. Some headings are sentences and are missing the verb.

Response: Subheadings have been revised to meet the 60-character limit and all have similar format now (brief phrase)

b. Trial schema and participant demographics

i. Figure S1 also shows trial Stage 3. It is not described here.

Response: We have added a footnote to Supplementary Figure 1: “*Note that data from Stage 3 of the study is not analyzed in the present study, but we include this stage in the overall schematic for completeness. The Stage 3 correlates results were described by Fong et al.¹ “

ii. “participants who developed COVID-19 prior to 7 days post D15 were excluded”: Unclear why they were excluded and confusing wording.

Response: Revised to “Further, participants with a COVID-19 endpoint prior to 7 days post D15 were excluded from the correlates analysis per-protocol cohort for the reason that the D15 antibody

marker measurements in these individuals might have been influenced by the preceding SARS-CoV-2 infection causing this COVID-19 endpoint.” (lines 190-194)

iii. No information about prior vaccination status reported here. Gives the impression that individuals did not receive any previous vaccinations.

Response: We have added the following sentence: “The COVAIL trial enrolled healthy adults (≥ 18 years old), irrespective of prior SARS-CoV-2 infection status, who had received a primary vaccination series and a single boost (homologous or heterologous) with an approved or emergency use authorized COVID-19 vaccine.^{18,19}” (lines 170-173)

We have also clarified “as a second vaccine boost” in the description of each vaccine regimen for each stage.

c. Available neutralizing antibody titer data

i. “BA.4/BA.5 titers at D15 were only measured from a random subset”: Why?

Response: At the time assays were chosen, D15 BA.4/5 was not a priority, and so BA.4/5 assays were not run with the other assays. However, a concordance study was later done using a random subset of stage 1 and stage 2 D15 samples to compare with the Duke pseudovirus assay.

ii. “D15 BA.4/BA.5 titers for participants with missing values were imputed based on D29 BA.4/BA.5 ID50 and D15 BA.1 ID50”: Reference to part of methods where imputation is described is missing.

Response: We have added a reference to the SAP (as suggested above to refer to SAP instead of Methods).

d. Neutralizing antibody titers lower in COVID-19 cases than non-cases

i. “Naïve participants had lower D1 titers than non-naïve participants, with 26% vs. 3% undetectable”: Are these percentages medians of the study population? What is the meaning of “undetectable”?

Response: We have revised to “Naïve participants had lower D1 BA.1 titers as well as lower D1 BA.1 response rates compared to non-naïve participants, with 26% (N=163 of 629) of naïve participants vs. 3% (N=9 of 356) of non-naïve participants having undetectable BA.1 titer [defined as D1 BA.1 titer \leq limit of detection (LoD), 40 AU/ml].” (lines 268-272)

ii. “are expected to be boosted 22-fold”: Is that their prediction or something they measured? Again, are these median numbers?

Response: Please see the response to Discussion comment 2b above on prediction of titers and linear regression. We have made a minor edit (underlined) to clarify that the boost refers to the level on D15:

“D1 BA.1 titers of 100 and 1000 AU/ml for both naïve and non-naïve participants are predicted to be boosted 22-fold and 7-fold, respectively, by D15.” (lines 275-277)

iii. “with geometric mean (GM) ID50 in non-cases, booster-proximal cases, and booster-distal cases 203, 155, 134 AU/ml”: Missing confidence intervals for the means. Same in the next sentence.

Response: Added in both sentences.

iv. Why use AU/ml? According to the methods, there was also some conversion to the WHO International Standard, which would result in IU/ml.

Response: Thank you for the question. Only D614G titers can be expressed in IU50/ml and therefore all results are reported in AU/ml. We have added the following to Methods:

“For the D614G strain, multiplying AU/ml by 0.0653 calibrates these titers to those in the 20/136 WHO International Standard for anti-SARS-CoV-2 immunoglobulin (described in the Supplementary Materials of Gilbert et al.²⁶), enabling the titers to be expressed in International Units (IU50/ml). Variant-specific neutralizing antibodies in the 20/136 International Standard, however, have not been measured in the PhenoSense SARS CoV-2 Assay, and thus only D614G titers can be expressed in IU50/ml. For this reason, we always report AU/ml in the present work.” (lines 785-792)

e. D15 (Peak) titer inverse correlates of risk of COVID-19 stronger for non-naïve than naïve participants
i. “The associations between D15 titers with subsequent COVID-19 were generally stronger in non-naïve than naïve participants”: What associations? Is “correlations” meant?

Response: Revised to “The inverse correlations of D15 titers with COVID-19 risk...” (line 301)

ii. “intersection of the middle 90% of titer values”: Would this be 5% to 95% quantile? Why restrict the analysis to these values?

Response: The quoted phrase corresponds to the intersection of the middle 90% of titer values in naïve participants with the middle 90% of titer values in non-naïve participants. The driving rationale is that the peak titer of naïve participants could be so low that few, if any, non-naïve participants in the dataset had such a titer; conversely, non-naïve participants could have high peak titers, such that they exceed the level seen in any naïve participants. Restricting to a range of peak titers with both naïve and non-naïve participants helps improve the robustness of such an interaction test.

iii. “We further conducted a sensitivity analysis to explore the influence of a different covariate adjustment strategy on the results”: What is the “different covariate adjustment strategy”?

Response: We have revised to clarify the original and alternative covariate adjustment strategies, as follows (lines 329-335):

“In the analyses shown in Fig. 5, adjustment was done for FOI score and risk score. We further conducted multiple sensitivity analyses to explore the influence of three alternative covariate adjustment strategies [adjustment for: 1) only the FOI score and the ≥ 65 age indicator; 2) adjustment only for the ≥ 65 age indicator; 3) no covariate adjustment] on the results; hazard ratios of each D15 ID50 marker remained virtually unchanged across all three strategies (Supplementary Fig. 16).”

iv. “The HR point estimates were largely consistent across the subgroups, with that among Prototype insert boost recipients closest to unity [1.00 (0.56, 1.78); $P = 0.999$] and that among Monovalent Omicron insert boost recipients the smallest [0.57 (0.30, 1.07)”: Unclear what the authors mean by “given that larger fold-rises tend to occur among those with lower baseline antibody levels.”: This is not reported in any figure or supported by another reference.

Response: We have added a new supplementary figure to support this statement and made the following revision in the text: “These fold-rise correlates are challenging to interpret compared to D15

absolute-level correlates, given that larger post-boost fold-rises in antibody titer were seen in the data analyzed here (as shown in Supplementary Fig. 23 for BA.1).” (lines 362-365)

f. Exploratory Analysis of pre and post COVID-19 titer by previous infection status

i. The slopes should probably be weighted by the distance to the infection, because dynamics may vary strongly.

Response: We agree that accounting for the genetic/proteomic sequence and/or antigenic distance of the infecting virus to the neutralizing antibody variant could in theory impact the rise in neutralizing antibody, for example a BA.4 infection might have a different rise of BA.1 neutralization titer compared to a BA.1 infection, all things being equal. However, it would be a major research endeavor to conduct these analyses, requiring new/additional statistical methods that would consider what types of viral distances are most appropriate, and how to account for the missing data on viral sequences.

More importantly, for distance to change the main conclusions of this exploratory analysis, there would need to have been a large imbalance in the infecting strains between naïve and non-naïve participants. In fact the infecting strains were similar between naïve and non-naïve participants (chi-squared $p = 0.21$), as shown in the table below.

Lineage	Non-Naïve	Naïve
BA.2	7	58
BA.4	2	15
BA.5	34	157
Missing	3	17

ii. “Figure S20A”: Figure S22A is probably meant.

Response: Thank you for the catch; we have corrected (Supplementary Fig. 29a in the revision).

iii. “Adjusting for pre COVID-19 titer being above or below the median in the non-naïve arm”: Why is this adjustment done?

Response: As pre COVID-19 titers tend to be higher in non-naïve participants, there is potential confounding of the non-naïve effect with pre COVID-19 titer, hence the adjusted analysis.

g. Lineage-specific correlates of risk: Similar results when restricting to BA.4/BA.5 COVID-19

i. “However, the lineage-matched analyses displayed marginally improved precision as the lineage-matched-based CIs were in general narrower than those obtained based on D614G titer.”: This is not a valid conclusion. It may just mean that the D614G titer does not have an effect on the risk of infection.

Response: The sentence was meant to comment on an empirical observation rather than to make a conclusion. Regardless of if the CIs cover 1 or not, BA.4/5-titer-based CIs are in general narrower than their counterparts based on D614G titer. If one believes the inverse correlates conclusion, e.g., among non-naïves, then one may conclude that lineage-matched nAb could be slightly advantageous over non-matched titer.

4. Discussion

a. “weak inverse correlates with COVID-19”: See above.

Response: Revised to “In analyses of naïve participants combined over all vaccine arms, the hazard ratios of COVID-19 (over ~6 months following the second booster) per 10-fold increase in D15 neutralizing antibody titer ranged from 0.78 to 0.85 across the six markers — suggestive of weak inverse correlates of COVID-19 risk — but with all 95% CI upper limits slightly higher than 1.” (lines 506-510)

b. “with all 95% CI upper limits slightly higher than 1”: See above about multiple testing correction and a different formulation would just be that these markers had no significant effect on the risk of infection with SARS-CoV-2.

Response: We have softened the language, please see above.

c. “anamnestic response to COVID-19”: What is an anamnestic response?

Response: An anamnestic reaction is defined as an “immunological response in which a second or subsequent exposure to an antigen causes a greater and more rapid reaction than that elicited by the initial exposure. It is a manifestation of immunological memory.” (Oxford Dictionary of Biochemistry and Molecular Biology, 2nd edition, 2008, DOI: 10.1093/acref/9780198529170.001.0001).

We have added the following sentence to the revision, as well as two references that discuss anamnestic immune responses in COVID-19:

“Memory B cells that are reactivated, e.g. via SARS-CoV-2 infection that occurs post-vaccination, are a source of anamnestic antibody responses.^{33,34}” (lines 547-548)

33 Sette, A. & Crotty, S. Immunological memory to SARS-CoV-2 infection and COVID-19 vaccines. *Immunol Rev* 310, 27-46, doi:10.1111/imr.13089 (2022).

34 Goldblatt, D., Alter, G., Crotty, S. & Plotkin, S. A. Correlates of protection against SARS-CoV-2 infection and COVID-19 disease. *Immunol Rev* 310, 6-26, doi:10.1111/imr.13091 (2022).

d. “The hazard ratios for both naïve and non-naïve participants were stronger than for the D15 correlates and estimated with more precision.”: See above.

Response: Revised to “For both naïve and non-naïve participants, the predicted-at-exposure inverse correlates were stronger than for the D15 correlates”. (lines 561-563)

e. “These results suggest that pre-existing antibody at exposure may be a better measure of protective ability over follow-up than peak antibody”: I do not see how this follows from the previous sentence.

Response: We have rephrased to “These results suggest that the current level of antibody at exposure may be a better measure of protective ability over follow-up than peak antibody, which seems intuitive.” (lines 563-565)

f. “the study was well powered”: The study had sufficient statistical power?

Response: Made this revision. (line 582)

5. Methods

a. Methods are spread out across the main article, the supplementary methods section, and the SAP. For details the reader should always only be referred to the SAP, as it contains the most detailed information.

Response: Done.

b. “secular trends”: What does secular mean in this context?

Response: We have revised to “temporal”.

Reviewer #2 (Remarks on code availability):

I had a brief look at the code. It contains a README file and a few routines for basic analysis. In my opinion it would be very beneficial if the dataset could also be included.

Response: Please see our revised data availability statement.

** See Nature Portfolio’s author and referees’ website at www.nature.com/authors for information about policies, services and author benefits.

References

- 1 Centers for Disease Control and Prevention. National Notifiable Diseases Surveillance System (NNDSS): Coronavirus Disease 2019 (COVID-19) 2021 Case Definition. Last reviewed August 24, 2021. Access date October 3, 2024. <https://ndc.services.cdc.gov/case-definitions/coronavirus-disease-2019-2021/>.
- 2 Zhang, B. *et al.* Omicron COVID-19 Immune Correlates Analysis of a Third Dose of mRNA-1273 in the COVE Trial. doi: 10.1101/2023.10.15.23295628. Posted 15 Oct, 2023. Access date 20 May, 2024. *medRxiv* (2023).
- 3 Baden, L. R. *et al.* Efficacy and Safety of the mRNA-1273 SARS-CoV-2 Vaccine. *N Engl J Med* **384**, 403-416, doi:10.1056/NEJMoa2035389 (2021).
- 4 El Sahly, H. M. *et al.* Efficacy of the mRNA-1273 SARS-CoV-2 Vaccine at Completion of Blinded Phase. *N Engl J Med* **385**, 1774-1785, doi:10.1056/NEJMoa2113017 (2021).
- 5 Benkeser, D. *et al.* Comparing antibody assays as correlates of protection against COVID-19 in the COVE mRNA-1273 vaccine efficacy trial *Science Translational Medicine* **15**, eade9078 (2023).
- 6 Gilbert, P. B. *et al.* Immune correlates analysis of the mRNA-1273 COVID-19 vaccine efficacy clinical trial. *Science* **375**, 43-50, doi:10.1126/science.abm3425 (2022).
- 7 Hejazi, N. S. *et al.* Stochastic interventional approach to assessing immune correlates of protection: Application to the COVE messenger RNA-1273 vaccine trial. *Int J Infect Dis* **137**, 28-39, doi:10.1016/j.ijid.2023.09.012 (2023).
- 8 Huang, Y. *et al.* Stochastic Interventional Vaccine Efficacy and Principal Surrogate Analyses of Antibody Markers as Correlates of Protection against Symptomatic COVID-19 in the COVE mRNA-1273 Trial. *Viruses* **15**, doi:10.3390/v15102029 (2023).
- 9 Plotkin, S. A. & Gilbert, P. B. Nomenclature for immune correlates of protection after vaccination. *Clin Infect Dis* **54**, 1615-1617, doi:10.1093/cid/cis238 (2012).

- 10 Branche, A. R. *et al.* Comparison of bivalent and monovalent SARS-CoV-2 variant vaccines: the phase 2 randomized open-label COVAIL trial. *Nat Med* **29**, 2334-2346, doi:10.1038/s41591-023-02503-4 (2023).
- 11 Branche, A. R. *et al.* Immunogenicity of the BA.1 and BA.4/BA.5 Severe Acute Respiratory Syndrome Coronavirus 2 Bivalent Boosts: Preliminary Results From the COVAIL Randomized Clinical Trial. *Clin Infect Dis* **77**, 560-564, doi:10.1093/cid/ciad209 (2023).
- 12 Korber, B. *et al.* Tracking Changes in SARS-CoV-2 Spike: Evidence that D614G Increases Infectivity of the COVID-19 Virus. *Cell* **182**, 812-827 e819, doi:10.1016/j.cell.2020.06.043 (2020).
- 13 Srivastava, K. *et al.* SARS-CoV-2-infection- and vaccine-induced antibody responses are long lasting with an initial waning phase followed by a stabilization phase. *Immunity* **57**, 587-599 e584, doi:10.1016/j.immuni.2024.01.017 (2024).

REVIEWERS' COMMENTS

Reviewer #1 (Remarks to the Author):

The authors have for the most part addressed the concerns of this reviewer - and there responses are thoughtful and comprehensive.

However, I note that the authors have still not made the data publicly available. This was a concern raised by both reviewers.

The data sharing statement added by the authors declares that people are able to contact the author and request the data and only if its approved by an NIH department will the data be shared. This is not making the data available, and in my teams experience the vast majority of such requests are ignored or rejected.

The only way to ensure that data is actually available, and able to be used to ensure reproducibility, is for the data to be made publicly available as a supplement or in a repository.

I note of course that this is a more broad issue in the field than being particular to this manuscript or author group.

Response: We agree with this view and tried to gain permission to publicly share the data. This study was funded by the Division of Microbiology and Infectious Diseases of NIAID who control access to the data. After extensive discussion their position was that data could be shared after vetting but not made publicly available and that requests should follow the standard procedure they have in place.

Reviewer #2 (Remarks to the Author):

We would like to thank the authors for addressing every comment with such attention to detail. The authors have answered most of the comments clearly and resolved most issues of the original manuscript. However, there were a couple of responses that warrant further discussion:

1) Response #2 – “The risk score is a function of baseline covariates and takes on a

different value [...]”: To begin with, it seems that parts of my comments have been lost in the response letter. These parts mostly appear to be equations. There may have been some misunderstanding due to this. To clarify the situation, I will try to explain my issue with the superlearner and the Cox model again. It was unclear to me where the superlearner enters the Cox model. As it was called a baseline risk, it could have entered the model as λ_0 . Then X_i would have been covariates such as age and sex, as they X_i was called “baseline covariates”. In this case, the covariates would have been included in the model twice (once in the superlearner and once in X_i). However, after carefully reading Section 6 of the SAP again, it has become clear that the output of the superlearner enters the model as a “baseline covariate” in $X_i * \beta$. In my opinion, this connection is not very clear from the main text or the beginning of the supplementary methods and should be made clearer earlier. Apart from this issue, the authors have made a good effort to evaluate their results by comparing different model formulations and it is interesting to see that the results remain very similar. This obviously raises the question whether it is necessary to adjust for the covariates using the superlearner.

Response: The ‘baseline risk score’ developed in this article is a covariate X_i used in the Cox regression adjustment, but not the baseline hazard. We have clarified this through the following revision (underlined) in the “Baseline risk score” section of Methods:

“The baseline risk score developed using the Superlearner^{46,47} model (CV-AUC = 0.604) was adjusted for as a covariate in all correlates analyses and cumulative incidence analyses. See the SAP for further details.”

Indeed, we showed that the association between the biomarker and the outcome is robust to inclusion or exclusion of the risk score. The SAP prespecified including the risk score in the primary analysis. We have also further clarified that the risk score was adjusted as a baseline covariate in the peak, fold-rise and exposure-proximal correlates analyses. Please see our response to your comment (5).

3) Response #5 – “We have interrogated the interaction test more thoroughly [...]”: The interaction part has become clearer, but it is unclear what the authors mean with a hazard ratio ratio (“HR Ratio”). I believe, what the authors mean to show is just the hazard ratio (per one standard deviation increase of antibody titer), as it is described by the proportional hazards model. This is not what was written though and should be clarified

Response: Sorry for the confusion. HR ratio refers to a ratio of two hazard ratios, one HR describing the association between the marker and the outcome among the baseline naïve study participants and the other HR describing the marker-outcome association among the baseline non-naïve study participants.

4) Response #9 – “We have added the following (lines 524-530)[...]”: Simply a hint that the revised manuscript says “Prototype vaccines” but the response text says “ancestral strain vaccines”

Response: Thanks for catching this. We meant to say ‘Prototype vaccines’ as in the revised manuscript.

5) Response #13 – “Please see the response to reviewer #1, “Other issue” #3.”: Is related to response #2 and my comment to this response.

Response: We found the specific request of this comment somewhat difficult to identify, but our best interpretation is that the reviewer is asking for further clarification on covariate adjustment. We have now further clarified that analyses were conducted separately for naïve and non-naïve participants and adjusted for two covariates: the FOI score and the risk score in the Methods section under ‘D15 and fold-rise correlates of risk analyses:’

“Analyses were conducted separately for baseline naïve and non-naïve participants. The analyses were based on Cox regression models adjusting for two covariates: the FOI score and baseline risk score and conducted separately for each D15 and fold-rise nAb ID50 marker.”

We also further clarify it in the Methods section under “*Exposure-proximal correlates of risk analysis:*”

“These analyses were conducted separately in naïve and in non-naïve participants, and adjusted for the baseline risk score as a covariate.”

6) Response #46 – “We have softened the language, please see above.”: Since the 95% CI of the hazard ratio includes 1, it does suggest that there is no relationship between titer and risk of COVID-19. I would be very careful about interpreting things which are not significant or only barely significant. It is well known that effect size estimates are in these cases biased and mostly overestimated.

Response: Thank you. We agree that the findings that are barely significant need be interpreted with great caution. We believe that our current revised version: 'suggestive of weak inverse correlates of COVID-19 risk — but with all 95% CI upper limits slightly higher than 1' has called for readers' attention to interpret these results carefully.

As soon as the above mentioned issues have been resolved, we can recommend the current work for publication.